# Integrated plasma and vegetation proteomic characterization of infective endocarditis for early diagnosis and treatment

Shiman He[1,7], Xuejiao Hu[2,7], Jiajun Zhu[1,7], Weiteng Wang ●[3,4,7], Chi Ma[1,7], Peng Ran[1,7], Oudi Chen[3], Fanyu Chen ●[3], Hongkun Qing[3], Jianhong Ma[5], Danni Zeng[2], Yunzhi Wang ●[1], Weijiang Liu[2], Jinwen Feng ●[1], Lixi Gan[3], Zhaoyu Qin ●[1], Subei Tan ●[1], Sha Tian ●[1], Chen Ding ●[1,6] ✉, Xuhua Jian ●[3] ✉ & Bing Gu ●[2] ✉

Infective endocarditis, a life-threatening condition, poses challenges for early diagnosis and personalized treatment due to insufficient biomarkers and limited understanding of its pathophysiology. Here, we performed proteomic profiling of plasma and vegetation samples from 238 patients with infective endocarditis and 100 controls, with validation in two external plasma cohorts (n = 328). We developed machine learning-based diagnostic and prognostic models for infective endocarditis, with area under the curve values of 0.98 and 0.87, respectively. Leucine-rich alpha-2-glycoprotein 1 and NADH:ubiquinone oxidoreductase subunit B4 are potential biomarkers associated with infection severity. Pathologically, protein networks characterized by glycometabolism, amino acid metabolism, and adhesion are linked to adverse events. Liver dysfunction may exacerbate the condition in patients with severe heart failure. Neutrophil extracellular traps emerge as promising therapeutic targets in *Streptococcus* or *Staphylococcus aureus* infections. Our findings provide insights into biomarker discovery and pathophysiological mechanisms in infective endocarditis, advancing early diagnosis and personalized medicine.

Infective endocarditis (IE) is a life-threatening disease characterized by infection, usually bacterial, of the endocardial surface of the heart. Endocardial-cell damage exposes subendothelial collagen and other matrix molecules, allowing platelets, fibrin, and bacteria circulating in the bloodstream to adhere and colonize, ultimately leading to germ-laden vegetation formation[1]. The annual incidence rate of IE is rising worldwide, estimated to be 3–15 per 100,000 people[2]. IE can trigger multiple complications, including heart failure, stroke, intracardiac abscess, and systemic embolization[3], and the 1-year mortality rate of IE is up to 30%, which is worse than that of many cancers[4].

The diagnosis of IE is based on a combination of major and minor criteria rooted in microbiologic, echocardiographic, and clinical

[1]Clinical Research Center for Cell-based Immunotherapy of Shanghai Pudong Hospital, State Key Laboratory of Genetics and Development of Complex Phenotypes, School of Life Sciences, Human Phenome Institute, Fudan University, Shanghai, China. [2]Department of Laboratory Medicine, Guangdong Provincial People's Hospital (Guangdong Academy of Medical Sciences), Southern Medical University, Guangzhou, China. [3]Department of Cardiovascular Surgery, Guangdong Cardiovascular Institute, Guangdong Provincial People's Hospital (Guangdong Academy of Medical Sciences), Southern Medical University, Guangzhou, China. [4]Department of Cardiac Surgery, Fuwai Hospital, National Center for Cardiovascular Diseases, Chinese Academy of Medical Sciences and Peking Union Medical College, Beijing, China. [5]Department of Laboratory Medicine, Heyuan People's Hospital, Heyuan, China. [6]Departments of Cancer Research Institute, Affiliated Cancer Hospital of Xinjiang Medical University, Xinjiang Key Laboratory of Translational Biomedical Engineering, Urumqi, China. [7]These authors contributed equally: Shiman He, Xuejiao Hu, Jiajun Zhu, Weiteng Wang, Chi Ma, Peng Ran. ✉e-mail: chend@fudan.edu.cn; jianxuhua@gdph.org.cn; gubing@gdph.org.cn

metrics, as outlined in the 2023 Duke-ISCVID Criteria[5]. However, the complexity and inaccuracy of pathogen identification and echocardiographic technologies pose challenges to achieving an early and accurate diagnosis of IE. These issues cause difficulties in clinical decision-making for up to 30% suspected IE patients[6]. Plasma biomarkers have important application value in disease diagnosis. Although proteins, such as procalcitonin (PCT), apolipoprotein A-I (APOA1), cystatin C (CST3), and complement C3, have been identified as potential biomarkers for IE, few stable and reliable biomarkers of IE have been discovered thus far because of either a low specificity or insufficient evidence[7–9].

The successful treatment of IE hinges on eradicating in vivo pathogens and repairing heart valves, typically necessitating bactericidal antibiotics and surgery[6]. Prognostic assessment and risk stratification are crucial for clinical decision-making in IE, particularly for native valve IE (NVIE)[10], guiding prophylactic and therapeutic strategies. Prior studies have revealed independent predictors of IE mortality, including age, left-sided valve vegetation, and complications such as heart failure and embolic events[11,12]. Rapid identification of patients with these risk factors and subsequent surgical treatment improves survival outcomes in IE patients. However, preoperative risk assessment for IE remains a challenge in clinical decision-making due to the lack of robust and efficient preoperative risk stratification models[13], which primarily focus on clinical manifestations and may be further refined by incorporating biomarkers from blood tests. Furthermore, antibiotic resistance and operative risk further exacerbate the already limited treatment options for IE[4,14]. Thus, developing advanced therapeutic regimens to enhance the safety and efficacy of treatments is essential. Yet, this effort is hindered by a limited understanding of the complex pathological mechanisms involved in IE.

Exploring the underlying biomarkers and pathological mechanisms related to disease severity and prognosis is crucial for effective risk prediction and stratified therapy in IE, where severity and prognosis are linked to valvular damage, infection severity, and complications. The microbial etiology of IE varies across regions and socioeconomic contexts. IE patients resulting from prior rheumatic heart disease remain prevalent in developing countries, accounting for up to two-thirds of cases[4]. In this context, IE is commonly caused by infections with viridans group streptococci or group D streptococci, predominantly affecting the mitral and aortic valves. Other common causative pathogens of IE include *Staphylococcus aureus* (*S. aureus*), coagulase-negative staphylococci (CoNS), and *Coxiella burnetii* (*C. burnetii*)[14]. Nevertheless, the pathogen-specific host response patterns for these organisms are poorly understood, limiting our insights into the mechanisms by which they contribute to developing further intracardiac lesions in IE. Furthermore, structural valvular damage from vegetation and related lesions often leads to heart failure, worsening the condition of IE patients. Heart failure complicating IE is independently associated with poor outcomes and the main indication for surgery for IE[13]. Yet, there is still a gap in our understanding of the molecular mechanisms by which heart failure exacerbates IE. Enhancing our understanding of disease progression mechanisms is crucial for advancing the therapeutic management of IE.

To this end, our study aims to identify early diagnostic and prognostic biomarkers for IE and to elucidate the underlying pathological mechanisms of IE onset and progression, providing valuable insights to enhance clinical management. We performed a large-scale characterization of plasma and vegetation proteomes using mass spectrometry (MS) technology in a discovery cohort comprising 238 IE and 100 non-IE cases (Cohort 1). In addition, we included two external validation cohorts, Cohort 2 ($n = 184$) and Cohort 3 ($n = 144$), for validation. Our study offers a comprehensive resource that enhances our understanding of the molecular landscape of IE and helps in early and accurate diagnosis and stratified therapy.

## Results

### The plasma and vegetation proteomic landscape of IE
Reaching an early and accurate diagnosis in cases of suspected IE is a central challenge of the disease. We prospectively collected a discovery cohort (Cohort 1) comprising 238 IE patients and 100 non-IE individuals (control group) and conducted an integrated plasma and vegetation proteomic analysis to provide insights for the development of early and accurate diagnostic and therapeutic strategies for IE (Fig. 1A and Supplementary Fig. 1A). The non-IE individuals were selected from patients with heart valve disease who exhibited clinical features suggestive of IE but were ultimately diagnosed ruled out for IE ("Methods"). Plasma proteomic analysis was performed on 202 preoperative (collected at admission) and 23 postoperative (collected within three to seven days postoperatively) plasma samples from the 238 IE patients, along with 100 preoperative plasma samples from each non-IE individual. In this study, the plasma proteomic analysis was primarily based on proteomic data obtained from preoperative samples, except for those clearly indicated as derived from postoperative plasma samples. Concurrently, vegetation proteomic analysis was conducted on 158 fresh-frozen vegetation samples from Cohort 1. In addition, we recruited two external validation cohorts, Cohort 2 ($n = 184$) and Cohort 3 ($n = 144$), consisting of plasma samples collected at admission for validation (Fig. 1A and Supplementary Fig. 1A). All participants in this study received surgical treatment. The majority of IE patients in our cohorts had NVIE, with only eight cases of prosthetic valve IE (PVIE)−four in Cohort 1, two in Cohort 2, and two in Cohort 3. Patients with implanted cardiac devices were not included. The baseline clinical characteristics of all participants are summarized in Supplementary Data 1.

The proteomic quality control analysis demonstrated high correlations among the proteomes of HEK293T cell samples and pooled plasma samples across different batches, suggesting the robustness and consistency of the MS data without batch effects (Supplementary Fig. 2A, B). A total of 9524 proteins were identified in plasma samples and 10,624 proteins in vegetation samples, with average counts of 2023 and 3384 proteins per sample, respectively (Supplementary Fig. 2C). The abundance of plasma proteins in IE was consistent with that in non-IE, and both were distributed at approximately 10 orders of magnitude (Supplementary Fig. 2D). Ultimately, 2314 plasma proteins and 3694 vegetation proteins were selected for downstream analysis after data filtering ("Methods").

### Plasma proteome-based machine learning classifier accurately distinguishes IE from non-IE cases
Uniform manifold approximation and projection (UMAP) for dimension reduction analysis distinctly discriminated between IE and non-IE cases among all three cohorts (Fig. 1B and Supplementary Fig. 2E, F). Differential analysis of the plasma proteome in Cohort 1 (FC > 1.5; FC < 0.6; FDR < 0.05) identified 273 differentially expressed proteins (DEPs) between the IE and non-IE cases. Proteins involved in the acute phase (e.g., CRP, SAA1, and SAA2), complement (e.g., C9 and CFHR5), and actin (e.g., ACTB, ACTG1, and POTEF) were upregulated in IE, while high-density lipoprotein (HDL) proteins, such as APOA2 and APOC1, and Ras GTPase-activating proteins, such as RASA4 and RASA4B, were downregulated (Fig. 1C). Epidemiological evidence shows an inverse association between plasma HDL levels and atherosclerotic cardiovascular disease risk[15]. Furthermore, pathway enrichment analysis revealed the upregulation of pathways involved in the inflammatory response and cell motility in IE compared to non-IE, while pathways associated with plasma lipoprotein clearance and protein ubiquitination were downregulated (Fig. 1D).

To identify early diagnostic biomarkers for IE, we next built a machine learning (ML) model using the extreme gradient boosting (XGBoost) algorithm[16] between IE and non-IE cases utilizing plasma proteomic data from the discovery cohort (Cohort 1) and two external

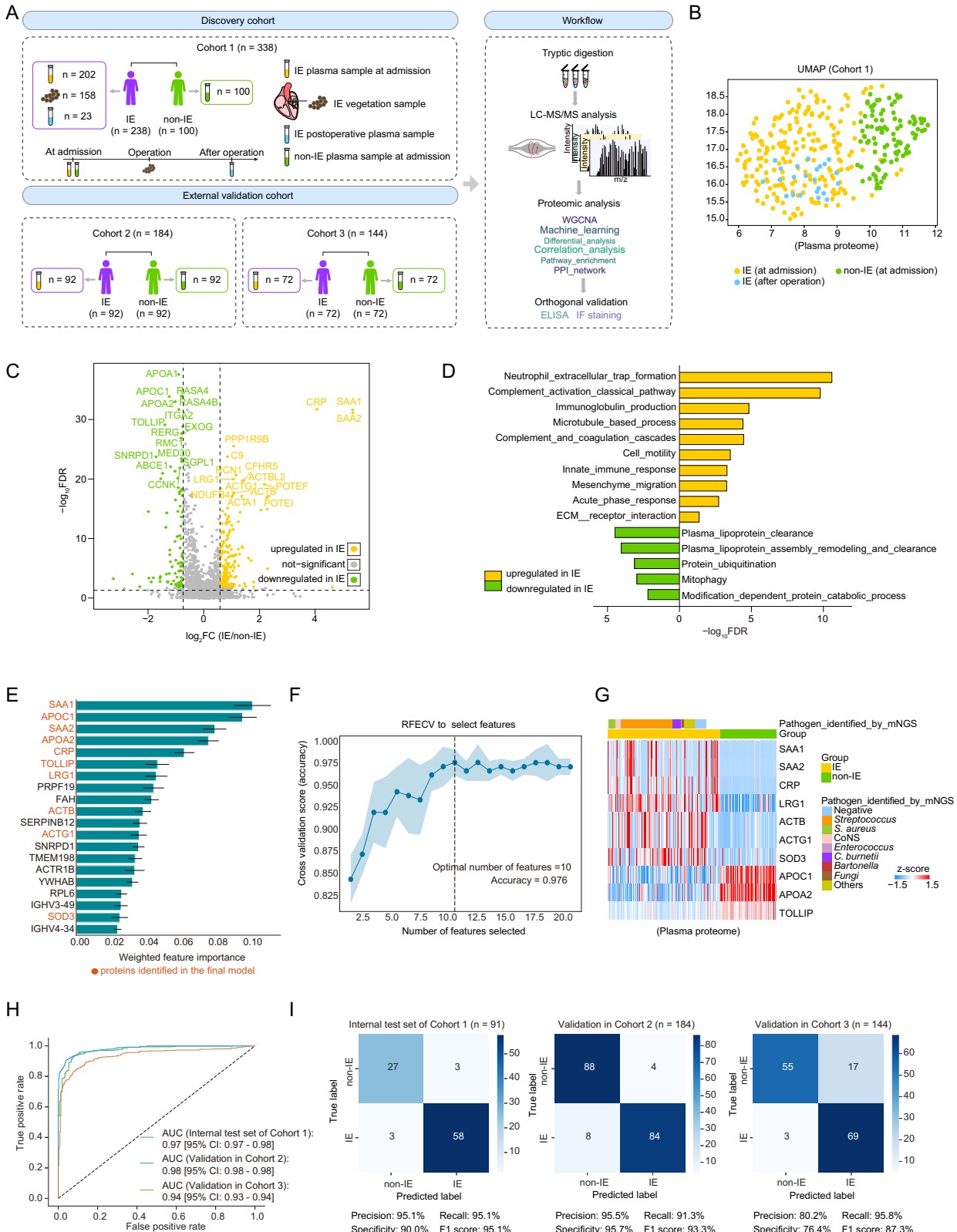

validation cohorts (Cohort 2 and Cohort 3) (Fig. 1E–I and Supplementary Fig. 1B; "Methods"). This diagnostic prediction model for IE contained a panel of 10 plasma proteins, including 7 that were upregulated in IE, comprising CRP, SAA1, SAA2, leucine rich alpha-2-glycoprotein 1 (LRG1), actin beta (ACTB), actin gamma 1 (ACTG1), and superoxide dismutase 3 (SOD3), while 3 that were downregulated, including APOA2, APOC1, and toll interacting protein (TOLLIP) (Fig. 1G

and Supplementary Fig. 2G, H). This model could discriminate IE from non-IE with an overall area under the receiver operating characteristic (ROC) curve (AUC) of 0.97 in the hold-out test set of the discovery cohort (Cohort 1), which was further independently validated in the two external validation cohorts with overall AUC values of 0.98 (Cohort 2) and 0.94 (Cohort 3) (Fig. 1H). In addition, we utilized a confusion matrix to evaluate the model's predictive capabilities, which

**Fig. 1 | Study design and comparative plasma proteomic analysis between IE and non-IE groups. A** Overview of the cohort populations and research strategy. **B** UMAP of the plasma samples in Cohort 1. **C** Volcano plot displaying dysregulated plasma proteins in IE at admission. The unpaired two-sided Wilcoxon rank-sum test was used for differential analysis, with p-values adjusted for FDR. The x-axis represents log2fold-change (FC), and the y-axis represents -log10false discovery rate (FDR). **D** Bar chart illustrating dysregulated pathways in IE. The x-axis represents -log10FDR. **E** Bar plot illustrating the weighted feature importance of the top 20 plasma proteins identified by the ensemble model to distinguish between IE and non-IE samples. The error bars represent the standard error of the mean (SEM) of the weighted feature importance calculated from five-fold cross-validation across six algorithms. Data are presented as mean ± SEM ($n = 6$). **F** Scatter line plot illustrating optimal feature combinations yielding the highest accuracy, determined

through recursive feature elimination cross-validation (RFECV). The error bands represent the 95% confidence interval (CI) from five-fold cross-validation. **G** Heatmap showing the expression profiles of the 10 proteins identified in the model. Each column represents a patient sample, and rows indicate proteins. The color range in the heatmap represents the row z-score of the normalized protein expression values, ranging from +1.5 (red) to −1.5 (blue). The annotation of the pathogen identified by metagenomic next-generation sequencing (mNGS) for each sample is displayed above the heatmap, with blanks indicating missing records. **H, I** Receiver Operating Characteristic (ROC) curves (**H**) and confusion matrices (**I**) illustrating model performance on the hold-out test set of the discovery cohort (Cohort 1) and two external validation cohorts, Cohort 2 and Cohort 3. Source data are provided as Source Data files.

showed a specificity of 90% and an F1 score of 95.1% in the hold-out test set of Cohort 1, and Cohort 2 and Cohort 3 exhibited consistent results (Fig. 1I).

Furthermore, the microbial etiology of IE has significant geographical differences, particularly between developed countries and developing countries[17]. More than half of the IE patients in our Cohort 1 had streptococcal infections (Fig. 1G). To further evaluate the model's robustness and generalization, we randomly selected patients from Cohort 1 based on pathogen proportions typical of developed countries and formed a reorganized test cohort for validation. Our findings showed that the proteins identified in the model maintained significant discriminative efficacy in the reorganized test cohort compared to Cohort 1 and were largely unaffected by changes in pathogen proportions (Supplementary Fig. 3A, B). Overall, these results demonstrate the model's robust performance and reliability in distinguishing between IE and non-IE individuals, indicating that the panel of 10 proteins identified in this model are promising early diagnostic biomarkers that could aid in clinical decision-making.

## Identification of prognostic protein co-expression networks of the plasma and vegetation proteomes

We then investigated the proteomic characteristics associated with various clinical features among IE patients by analyzing matched vegetation and plasma proteomes of Cohort 1. Weighted gene co-expression network analysis (WGCNA)[18] was performed to identify key protein networks (modules) associated with clinical traits in IE ("Methods"), and 15 modules in the plasma proteome and 16 modules in the vegetation proteome were detected (Supplementary Fig. 4). We isolated modules that were significantly correlated with clinical features and delineated the signaling pathways associated with the modules (Fig. 2A–D). In the plasma proteome (Fig. 2A, B), the positive results of blood culture, vegetation detected by echocardiography (Echo), and pathogen identified by metagenomics next-generation sequencing (mNGS) were significantly positively correlated with the MEbrown module, which was characterized by immunoglobulin production pathway, while exhibited a significant negative correlation with the MEturquoise module featured by negative regulation of endopeptidase activity pathway. This result demonstrated that immunoglobulin and endopeptidase protein networks were closely associated with infections within the IE patients. In addition, the MEred and MEgreenyellow modules, which were characterized by pathways involving glycometabolism, cell adhesion, antigen presentation, and mRNA splicing, were significantly positively associated with multiple risk factors and/or adverse events after cardiac surgery, including coronary artery disease (CAD), previous cardiac surgery, intra-aortic balloon pump (IABP), and ICU stays.

Moreover, in the vegetation proteome (Fig. 2C, D), mitral annular calcification (MAC), aortic valve calcification (AVC), and bicuspid aortic valve (BAV) had significantly negative correlations with the MEturquoise, MEgreen, and MEgreenyellow modules, which were enriched in the translation, vesicle-mediated transport, and platelet

degranulation pathways, respectively, indicating that the development of valve diseases may be related to abnormalities in these functions. Notably, we observed a remarkable positive association between reoperation and the MEpurple module characterized by collagen formation. The MEmagenta module, characterized by mitochondrial tRNA aminoacylation and antigen presentation, was significantly correlated with poor prognosis, such as death within 30 days after discharge (D30) and postoperative cerebral embolism (POCE). The MEmidnightblue module, characterized by metabolic functions, exhibited significant positive associations with postoperative adverse events, including extracorporeal membrane oxygenation (ECMO), low cardiac output syndrome (LCOS), and ICU stays.

As the ICU stay-associated modules were identified in both the plasma (MEred) and vegetation proteomes (MEmidnightblue), we further filtered hub proteins with high within-module connectivity to identify biomarkers related to ICU stays (Methods). Sixteen hub plasma proteins, including collagen type I alpha 2 chain (COL1A2) (MMred = 0.73, GS.ICU = 0.30) and nucleoporin 160 (NUP160) (MMred = 0.72, GS.ICU = 0.35), were identified (Fig. 2E) and enriched in pathways involving extracellular matrix (ECM) receptor interaction and glucose metabolism with protein-protein interactions (PPIs) (Fig. 2F, G). Fourteen hub vegetation proteins, such as glycine amidinotransferase (GATM) (MMmidnightblue = 0.89, GS.ICU = 0.28) and mannose phosphate isomerase (MPI) (MMmidnightblue = 0.89, GS.ICU = 0.23), were identified (Fig. 2H) and enriched in pathways including amino acid metabolism as well as fructose and mannose metabolism with PPIs (Fig. 2I, J). In summary, these findings suggest that the enhancement of glycometabolism, amino acid metabolism, and adhesion is associated with the exacerbation of IE patients' condition.

## The association between liver dysfunction and severe heart failure in IE progression suggests stratified therapy

Improving our understanding of the pathological mechanisms related to disease progression is crucial for developing stratified therapy in IE. Notably, the above-mentioned MEturquoise module in the plasma proteome exhibited significant negative correlations with multiple clinical features associated with IE progression, including positive microbial findings, detection of vegetation or abscess, mitral or tricuspid valve surgery, and the New York Heart Association (NYHA) Functional Classification, with the correlation with NYHA classification being the most significant (Fig. 2A). The NYHA functional classification is widely used to assess heart failure severity in clinical practice[19]. NYHA class III or IV is associated with poor prognosis of IE patients[11].

We indeed found significant enrichment of many risk factors and adverse events, such as ICU stay group ($p = 1.42e-05$) and D30 ($p = 0.016$), in NYHA class IV patients (Fig. 3A). Next, we identified 21 plasma proteins that were significantly negatively correlated with the NYHA functional classification and involved in the pathways of the MEturquoise module (Fig. 3B, C). Notably, we observed significant differences in the expression profiles of the 21 plasma proteins

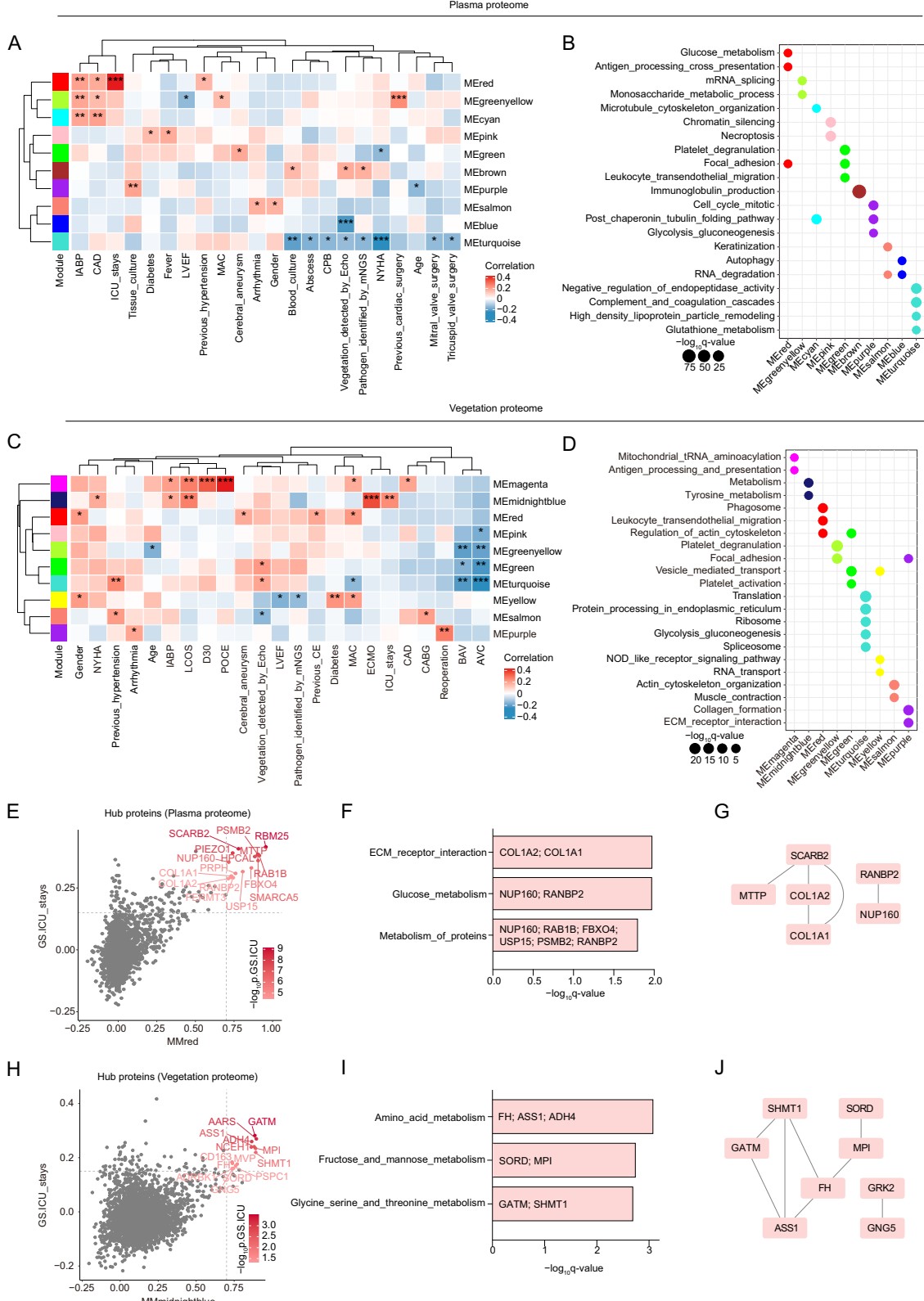

between NYHA class IV patients and those in NYHA class II and III (Fig. 3D), including multiple common liver-derived proteins, such as HDL (e.g., APOA1), complement (e.g., C4A), and serine protease inhibitor (e.g., SERPINA6), which were downregulated in NYHA class IV patients. Tissue- and cell-specific enrichment analysis showed that the 21 plasma proteins were markedly enriched in liver tissue (Fig. 3E). To further investigate whether patients with NYHA class IV had liver

dysfunction, we subsequently assessed the levels of six clinical blood markers (CBMs) to characterize liver function in patients. We found that the levels of aspartate aminotransferase (AST; $p = 0.0052$) and direct bilirubin (DBIL; $p = 2e-05$) were significantly higher in the NYHA class IV patients than those in the NYHA class II and III, whereas the level of albumin (ALB; $p = 0.031$) was lower (Fig. 3F), suggesting liver dysfunction in NYHA class IV patients of IE. In addition, we found a

**Fig. 2 | Plasma and vegetation proteomic modules associated with clinical traits of IE. A–D** Module–trait associations and characterization in the plasma and vegetation proteomes. Heatmap (**A**, **C**) showing the correlation between modules and clinical traits. Each row corresponds to a module. Each column corresponds to a clinical trait. The color range represents correlation coefficients. Bubble chart (**B**, **D**) outlining the pathways significantly enriched for the proteins in different modules. The bubble size corresponds to the significance of the pathways enriched in each module, and the colors correspond to the modules. **E-J** Hub proteins associated with ICU stays in the plasma (**E–G**) and vegetation (**H-J**) proteomic modules. Scatter plots (**E**, **H**) displaying the hub proteins associated with the ICU stays in the plasma and vegetation proteomes, respectively. The *x*-axis represents

the correlation between proteins and their corresponding modules, and the y-axis represents the correlation between proteins and the ICU stays. The color range represents the significance of the correlation between hub proteins and the ICU stays, indicated by -log10p-value. Bar charts (**F**, **I**) showing pathways significantly enriched for the hub proteins. STRING analysis (**G**, **J**) showing the protein-protein interactions (PPIs) of the hub proteins. For figures (**A**, **C**, **E**, **H**), two-sided Pearson's correlation (WGCNA-derived) was used, with significance denoted by *$p < 0.05$, **$p < 0.01$, ***$p < 0.001$. Abbreviations: CPB, cardiopulmonary bypass time; previous CE, previous cerebral embolism; CABG, coronary artery bypass grafting. Source data are provided as Source Data files.

positive correlation between the NYHA functional classification and advanced age (Fig. 3G; $r = 0.25$, $p = 2.9e-04$). These findings indicate a significant positive correlation between liver dysfunction and severe heart failure in IE progression.

To elucidate the crucial biological alterations underlying IE progression with heart failure, we performed comparative proteomic analyses between NYHA class IV patients and those in NYHA class II and III. Consistent with the above conclusion, both plasma and vegetation proteomes showed downregulation of pathways comprising negative regulation of endopeptidase activity, complement and coagulation cascades, and vitamin transport (Fig. 3H–K), as well as a reduction in the expression of proteins including APOA1, C4A, and SERPINA6 in NYHA class IV patients (Fig. 3L). In NYHA class IV, both the plasma and vegetation proteomes exhibited upregulation of neutrophil degranulation pathway (Fig. 3H–K). Furthermore, the plasma proteome showed upregulation of pathways including extracellular matrix organization, cell adhesion, and proteolysis (Fig. 3I) in NYHA class IV, and the vegetation proteome demonstrated enrichment of intracellular protein transport and spliceosome pathways (Fig. 3K). These pathways, which have been implicated in tissue damage, cell mortality, and disease progression[20–22], may contribute to cardiac extracellular matrix remodeling in IE, which is a key pathological feature of heart failure[23]. Importantly, we demonstrated that the proportions of mitral ($p = 0.003$) and tricuspid valve surgery ($p = 0.01$) were significantly higher in NYHA class IV patients compared to those in NYHA class II and III (Fig. 3M), indicating a more severe valvular damage in NYHA class IV patients. The vegetation proteomic analysis revealed that tyrosine phosphatase receptor type F (PTPRF), a known signaling molecule regulating cell adhesion, exhibited the most significant positive correlation with mitral valve surgery among the proteins upregulated in patients with NYHA class IV (Fig. 3N). Further correlation and pathway enrichment analyses confirmed that PTPRF was closely associated with cell adhesion (Fig. 3O, P), suggesting that the enhancement of cell adhesion plays a vital role in valve damage in IE.

Furthermore, in Cohort 2, we further validated that the levels of those proteins, including complement, serine protease inhibitors, and HDL, were significantly decreased in NYHA class IV patients (Fig. 3Q). In addition, we confirmed consistent positive correlations between liver dysfunction, indicated by the levels of CBMs, including AST and DBIL, and heart failure, represented by the NYHA functional classification (Fig. 3R). Collectively, these findings underscore the necessity of risk stratification for IE patients in NYHA class IV, who typically exhibit higher postoperative risk compared to those in NYHA class II and III. Furthermore, concomitant liver dysfunction may exacerbate clinical deterioration in IE complicated by severe heart failure (Fig. 3S).

## Plasma proteome-based models supplement CBMs in predicting disease severity of IE

Given the association between NYHA class IV and poor prognosis in IE, we developed three prognostic models based on blood-derived features to distinguish NYHA class IV patients from those in NYHA class II and III. These models were developed using CBM data (CBM-only),

plasma proteomic profiles (proteome-only), and their combination (CBM + proteome) (Fig. 4A; "Methods"). The CBM data and plasma proteomic data were obtained from the blood samples collected at admission. As a result, the proteome + CBM model yielded the highest overall AUC of 0.873 in the hold-out test set of the discovery cohort (Cohort 1), which was better than that of the proteome-only model (0.831) and significantly better than the CBM-only model (0.76), demonstrating the effectiveness of the proteome + CBM model in predicting poor prognosis (Fig. 4B–J). Furthermore, we validated the generalization performance of the three models in the external validation cohort (Cohort 3), achieving overall AUC values of 0.829 for the proteome + CBM model (Fig. 4J), 0.801 for the proteome-only model (Fig. 4G), and 0.743 for the CBM-only model (Fig. 4D), indicating the robustness and generalizability of the models. Additionally, we evaluated the classification efficacy of the proteome + CBM model using a confusion matrix, achieving a specificity of 95% and an F1 score of 82.4% in the hold-out test set of the discovery cohort (Cohort 1), which was further validated in the external validation cohort (Cohort 3). These results demonstrated the reliability of the proteome + CBM model (Fig. 4K).

The proteome + CBM model incorporated a panel of 4 CBMs, consisting of elevated levels of DBIL, BUN, and D-dimer, alongside a decreased level of PLT in IE patients with NYHA class IV, and 11 DEPs, including upregulated proteins of adiponectin (ADIPOQ), CST3, and calcium/calmodulin dependent protein kinase kinase 2 (CAMKK2), as well as downregulated proteins of APOA1, fetuin B (FETUB), talin 1 (TLN1), vitamin D binding protein (GC), paraoxonase 1 (PON1), vacuolar protein sorting 25 homolog (VPS25), inter-alpha-trypsin inhibitor heavy chain 1 (ITIH1), and G3BP stress granule assembly factor 1 (G3BP1) (Fig. 4H, I). Moreover, in the reorganized test cohort with pathogen proportions typical of developed countries, we further validated that the 11 proteins maintained consistent discriminative efficacy compared to Cohort 1, demonstrating minimal influence from variations in pathogen prevalence (Supplementary Fig. 3C, D). Collectively, these findings demonstrate the proteome + CBM model's robustness and generalizability in distinguishing IE patients with NYHA class IV from those in NYHA class II and III, suggesting the clinical potential of this blood-based biomarker panel for predicting disease severity and prognosis in IE.

## Infection severity-associated proteomic characterization distinguishes IE patients with insufficient microbiological evidence from non-IE cases, identifying LRG1 as a potential biomarker

In addition to valve damage and impaired cardiac function, the progression of IE is closely related to the degree and type of pathogen infections. To investigate functional alterations related to infection severity, we classified patients into three categories with gradually increasing yields of detectable pathogens, namely, double-positive, single-positive, and double-negative, based on the results of blood culture and mNGS, which are commonly used methods to identify pathogens in IE[24]. Although this classification related to the yield of detectable pathogen showed no obvious correlation with heart failure severity, it exhibited significant associations with the incidence of

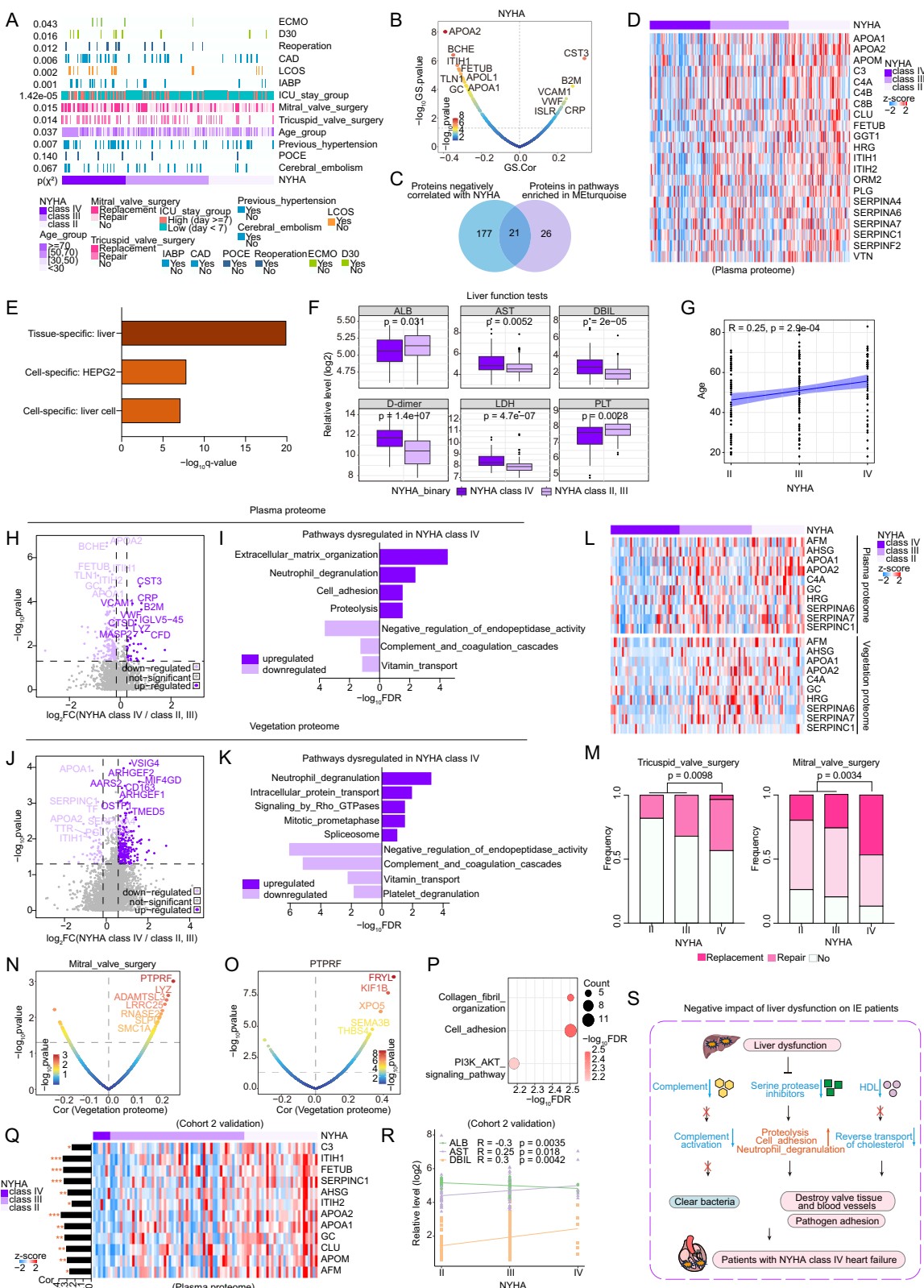

vegetation detected by Echo and fever, alongside poor postoperative outcomes, such as a longer ICU stay and more LCOS, ECMO, and IABP, which were significantly enriched in the double-negative group (Fig. 5A and Supplementary Fig. 5A). These findings indicated that IE patients with insufficient microbiological evidence could face a higher postoperative risk. Moreover, the comparative proteomic profiling revealed two distinct protein panels exhibiting progressive changes in

abundance from the double-positive category to the single-positive category and finally to the double-negative category in the plasma and vegetation proteomic profiles, respectively. Further pathway enrichment analysis was subsequently conducted on the two protein panels. In the plasma proteome, we observed elevated levels of immunoglobulins and inflammation-associated proteins in the double-positive group, whereas proteins related to lipid metabolism and transport

**Fig. 3 | Proteomic alterations associated with NYHA class in IE. A** Heatmap showing the association between NYHA class and 13 adverse events. **B** Scatter plot showing the correlation between proteins and NYHA class (colored by significance). **C** Venn diagram illustrating 21 proteins correlated with NYHA class (R < − 0.2, p < 0.05) and belonging to the pathways enriched in MEturquoise. **D, E** Heatmap and bar chart showing the 21 protein expression profiles and their histologic origin enrichment results. **F** Boxplots delineating the distribution of 6 CBMs. The central line represents the median, the box indicates the interquartile range (IQR), and whiskers extend to 1.5 × IQR, with outliers as individual points. **G** Scatter plot depicting the correlation between NYHA class and age. The error band represents the 95% CI of the regression line. **H–K** Volcano and bar plots showing the DEPs and dysregulated pathways enriched by these DEPs. **L** Heatmap showing the downregulated plasma and vegetation proteins in NYHA class IV. **M** Stacked bar plots showing the association between NYHA class and mitral and tricuspid valve surgeries. **N, O** Scatter plot showing the correlations of vegetation proteins with mitral valve surgery (**N**) and PTPRF (**O**). **P** Bubble chart showing significant pathways enriched by proteins correlated with PTPRF (R > 0.2, p < 0.05). **Q** Heatmap showing the complement, serine protease inhibitors, and HDL-related proteins in Cohort 2. Bar plot (left) showing the correlation between the proteins and NYHA class. **R** Scatter plot showing the correlation between NYHA class and ALB, AST, and DBIL levels in Cohort 2. **S** Schematic diagram summarizing the potential mechanisms by which liver dysfunction affects IE deterioration. Pearson's two-sided $\chi^2$ test was used for figures (**A** and **M**); two-sided Pearson's correlation (WGCNA-derived) for (**B** and **C**); and two-sided Spearman's correlation for (**G**, **N–R**), with p-values adjusted for FDR (*p < 0.05, **p < 0.01, ***p < 0.001). For (**F**, **H**, and **J**), unpaired two-sided Wilcoxon tests were used to compare NYHA class IV (n = 60) with NYHA class II and III (n = 134). Source data are provided as Source Data files.

were significantly downregulated (Fig. 5A). In the vegetation proteome, necroptosis and acute phase response pathways were preferentially enriched in the double-positive group, while cell adhesion-related pathways were predominantly enriched in the double-negative group (Supplementary Fig. 5A).

Next, we established a high-yield score and a low-yield score based on the above two plasma protein panels using single-sample gene set enrichment analysis (ssGSEA)[25] to represent the detectable pathogen yield ("Methods"). The two scores demonstrated a reasonably significant increasing or decreasing trend from the double-positive category to the single-positive category to the double-negative category and finally to the non-IE category, respectively (Fig. 5B). Of note, there were significant differences between double-negative category and non-IE category (high-yield score: p = 1.1e-08; low-yield score: p = 2.4e-08), which provided evidence for the diagnosis of IE patients with insufficient microbiological evidence (Fig. 5B). To further determine the relationship between the two scores and infection severity in IE, we examined their associations with established inflammatory markers[26], such as CRP, PCT, and the neutrophil ratio. The two scores were positively correlated with these markers, suggesting that they may serve as indicators for infection severity (Fig. 5C). In particular, the neutrophil ratio demonstrated a stepwise decline from the double-positive to the double-negative group, consistent with the decreasing pathogen yield (Supplementary Fig. 5B). Collectively, these findings indicate that our stratification system based on pathogen detectability can reflect infection severity, and provides potential diagnostic value for IE patients lacking conclusive microbiological evidence.

To further identify biomarkers associated with infection severity in IE, we screened proteins with the top 20 feature importance using an ML-based classifier to distinguish the double-positive category from the double-negative category, which included 11 upregulated proteins (e.g., C9 and LRG1) and 9 downregulated proteins, such as VPS25 and ferritin light chain (FTL), in the double-positive group (Fig. 5D, E). Notably, the protein LRG1 was also identified as a component of our diagnostic prediction model for distinguishing IE from non-IE individuals (Fig. 1G). Moreover, we found that the abundance of LRG1 exhibited significant differences, consistently decreasing across the double-positive, single-positive, and double-negative categories in both plasma and vegetation proteomic data (Fig. 5F). Similarly, in the plasma proteome, LRG1 levels were significantly elevated in the double-negative category compared to the non-IE category (p = 1.9e-04; Supplementary Fig. 5C). LRG1 plays a crucial role in the early differentiation of neutrophilic granulocyte differentiation[27] and has been increasingly recognized for its involvement in disease pathogenesis[28]. In the vegetation proteome, correlation and pathway enrichment analyses revealed positive associations between LRG1 and pathways related to neutrophil degranulation and scavenging of heme from plasma (Fig. 5G, H). Furthermore, enzyme-linked immunosorbent assay (ELISA)

validation confirmed that LRG1 levels were significantly higher in IE patients than those in non-IE individuals (p = 0.0045; Fig. 5I), with consistent trends observed in Cohorts 2 and 3 (Supplementary Fig. 2I, J). Interestingly, PPI analysis identified fifteen proteins significantly correlated with LRG1 across both plasma and vegetation datasets, including haptoglobin (HP, HPR), hemopexin (HPX), and hemoglobin subunits (HBB, HBG1, HBG2) (Fig. 5J and Supplementary Fig. 5D). Haptoglobin and hemopexin were significantly positively associated with LRG1, whereas hemoglobin was negatively correlated with LRG1 (Fig. 5K and Supplementary Fig. 5E). Hemoglobin and free heme–iron (Fe) are known to be rapidly bound by haptoglobin and hemopexin to prevent bacterial pathogen growth[29] and to safeguard tissue cells against direct heme toxicity[30]. Overall, these findings suggest that LRG1 has emerged as a potential biomarker for IE, capable of reflecting infection severity, and might play a role in regulating iron metabolism during IE infection (Fig. 5L).

## Neutrophil extracellular traps are potential therapeutic targets for IE patients infected with *Streptococcus* and *S. aureus*

To elucidate the pathogen-specific host response patterns in IE, we conducted comparative analyses of the plasma and vegetation proteomes among IE patients infected with the top four pathogens: *Streptococcus* (58.97%), *C. burnetii* (10.26%), *S. aureus* (8.33%), and CoNS (6.41%) (Fig. 6A, B and Supplementary Fig. 6A). Clinically, we observed a distinct correlation between the types of pathogen infections and heart failure severity, which was unrelated to the degree of infections (Fig. 6B and Supplementary Fig. 6B). Consistent with previous study[11], *S. aureus* infection, as a well-recognized marker of worse outcomes for IE, was significantly linked to severe heart failure, suggesting an increased risk in *S. aureus* infection compared to other infections (Supplementary Fig. 6C). Specifically, *C. burnetii* infection showed a higher incidence of non-fever presentation, abscess, AVC, and BAV compared to other infections (Fig. 6B), and it was featured by pathways including classical antibody mediated complement activation and immunoglobulin production in the plasma proteome, as well as pathways comprising metabolism of carbohydrates and cholesterol metabolism in the vegetation proteome (Fig. 6C, D). Moreover, CoNS infection was associated with a history of hypertension (Fig. 6B), and was characterized by fatty acid degradation and response to xenobiotic stimulus pathways in the plasma proteome, alongside vesicle-mediated transport and mRNA splicing pathways in the vegetation proteome (Fig. 6C, D). For *Streptococcus* infection, plasma proteomic alterations included the upregulation of pathways of mesenchyme migration, necroptosis, and neutrophil extracellular trap (NET) formation, which were mainly enriched by histones and actin-related proteins, while the vegetation profiling showed a significant increase in activation of complement and coagulation cascades and innate immune system pathways (Fig. 6C, D). In addition, *S. aureus* infection was associated with muscle contraction and phagosome pathways in the plasma proteome, and integrin cell surface interactions,

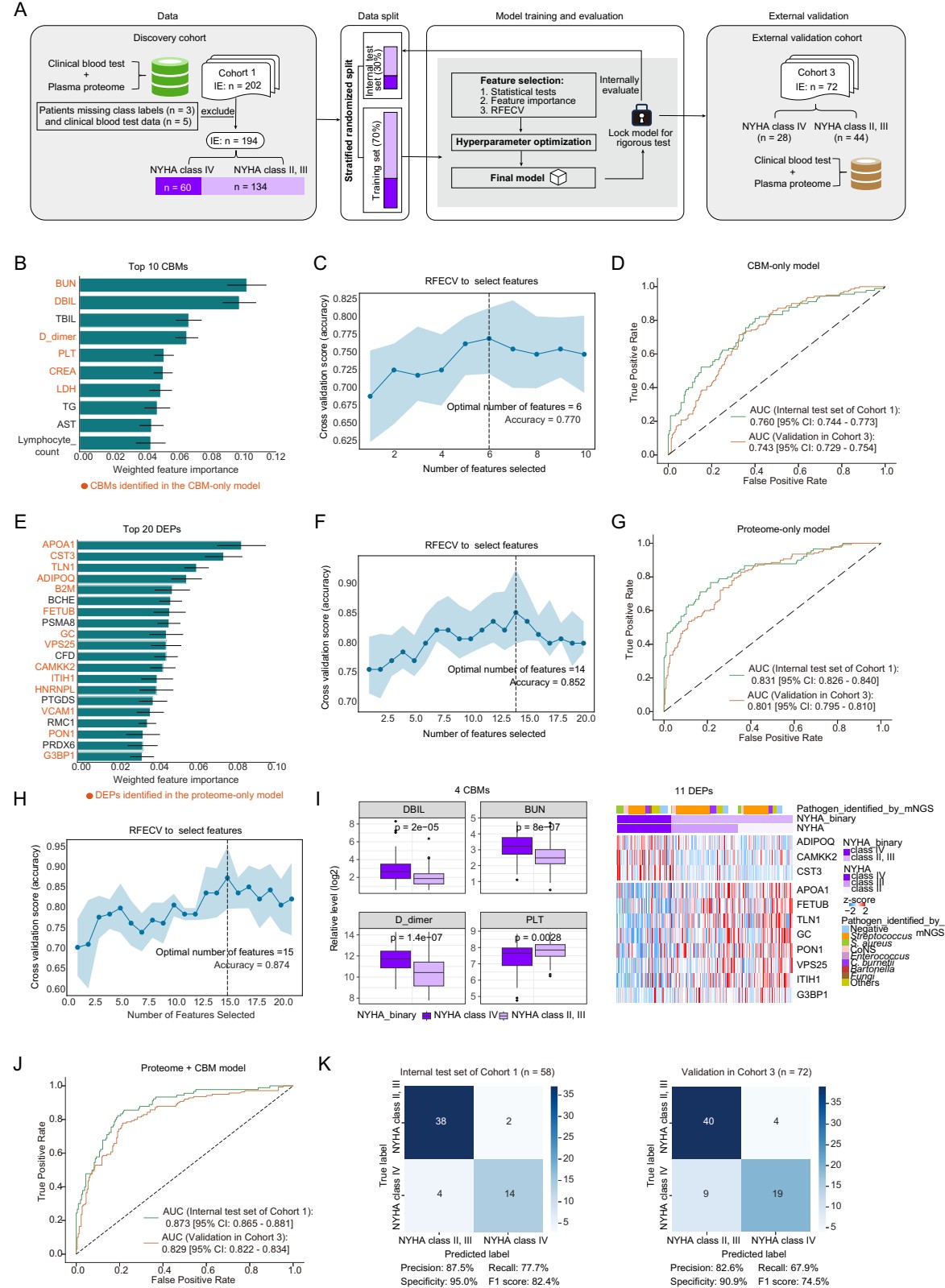

hemostasis, and NET formation pathways in the vegetation proteome (Fig. 6C, D).

Notably, *Streptococcus* and *S. aureus*, the two most dominant causative pathogens in IE patients, were found to share a common pathway of NET formation and to be associated with larger vegetation formation, compared to *C. burnetii* and CoNS infections (Fig. 6B–D). Excessive NETs could lead to vessel occlusion, tissue damage, and

bacterial biofilm formation[31,32]. We then explored the presence and role of NETs in IE. Actin cytoskeleton dynamics are necessary for the formation of NETs[33]. In the plasma proteome, we found that the top three significant PPI networks of *Streptococcus* cases were mainly composed of histones and cell migration-related proteins (Fig. 6E), and proteins positively correlated with neutrophil count and upregulated in *Streptococcus* cases were found to be enriched in processes related

**Fig. 4 | Prognostic models for distinguishing between NYHA class IV patients and those in NYHA class II and III. A** Schematic of the ML pipeline for developing prognostic models. **B–G** ML model establishment based on CBMs (**B–D**) and plasma proteome (**E–G**). Bar plots illustrating the weighted feature importance of the top 10 CBMs (**B**) and the top 20 DEPs (**E**) identified by the ensemble models. The error bars represent the SEM of the weighted feature importance calculated from five-fold cross-validation across six algorithms. Data are presented as mean ± SEM (*n* = 6). Scatter line plots (**C, F**) illustrating RFECV for optimally determining feature combinations with the highest model accuracy. ROC curves (**D, G**) depicting the AUC for the models. **H** Scatter line plot illustrating RFECV for optimal feature selection for the proteome + CBM model from the 6 CBMs and 14 DEPs identified independently in the CBM-only and proteome-only models to maximize model accuracy. **I** Boxplots and heatmap showing significant differences in the 15 features identified in the proteome + CBM model between NYHA class IV patients (*n* = 60) and those in NYHA class II and III (*n* = 134). For boxplots, the central line represents the median, the box indicates the IQR, and whiskers extend to 1.5 × IQR, with outliers as individual points. The unpaired two-sided Wilcoxon rank-sum test was used for differential analysis. The annotation of the pathogen identified by mNGS for each sample is displayed above the heatmap, with blanks indicating missing records. **J, K** ROC curves (**J**) and confusion matrices (**K**) illustrating the performance of the proteome + CBM model on the hold-out test set of the discovery cohort (Cohort 1) and the external validation cohort (Cohort 3). For figures (**C, F,** and **H**), the error bands represent the 95% CI from five-fold cross-validation. Source data are provided as Source Data files.

to actin cytoskeleton, complement system, and neutrophil degranulation pathways (Fig. 6F and Supplementary Fig. 6D). Vegetation proteomic data further corroborated these findings, showing elevated levels of myeloperoxidase (MPO), a well-established NET marker[32] in *Streptococcus* and *S. aureus* infections compared to CoNS and *C. burnetii* infections (*p* = 0.011; Fig. 6G). These results demonstrated potentially enhanced NET formation in *Streptococcus* and *S. aureus* infections. Moreover, NETs have been reported to promote biofilm formation paradoxically[31]. Echo imaging revealed a nearly universal detection of vegetations in *Streptococcus* and *S. aureus* infections, whereas vegetations were observed in only approximately half of the *C. burnetii* and CoNS infections (Fig. 6H), indicating that *Streptococcus* and *S. aureus* infections are more likely to cause a larger vegetation formation within the IE patients, and NETs may play a critical role in promoting vegetation development in IE.

To investigate the molecular basis of the association between redundant NETs and larger vegetation formation in *Streptococcus* or *S. aureus* infections, we subsequently performed differential analysis and pathway enrichment analysis between *Streptococcus* & *S. aureus* cases and CoNS & *C. burnetii* cases in the vegetation proteome. The results showed that proteins, including peptidoglycan recognition protein 1 (PGLYRP1), fibrinogen alpha chain (FGA), superoxide dismutase 2 (SOD2), and interleukin-8 (IL-8) were significantly upregulated in *Streptococcus* & *S. aureus* cases, while proteins such as HtrA serine peptidase 1 (HTRA1) and chemokine CXCL12 were remarkably enriched in CoNS & *C. burnetii* cases (Fig. 6I). Pathways involving complement and coagulation cascades and NET formation were significantly enriched in *Streptococcus* & *S. aureus* cases, while adhesion and metabolism-related pathways were increased in CoNS & *C. burnetii* cases (Supplementary Fig. 6E). Moreover, we demonstrated that proteins upregulated in *Streptococcus* & *S. aureus* cases and positively correlated with MPO were indeed enriched in neutrophil degranulation and cell movement-related pathways (Fig. 6J, K). The expression profiles of the proteins within these pathways exhibited significant differences between the two groups of patients (Fig. 6L). Immunofluorescence analyses further confirmed increased NET deposition in patients infected with *Streptococcus* & *S. aureus* (Fig. 6M). Overall, these findings demonstrated potential enhanced NET formation during the sustained inflammatory processes caused by *Streptococcus* or *S. aureus* infections, thereby facilitating a larger vegetation formation (Fig. 6N). Thus, we propose that targeting NETs may be an effective therapy for inhibiting vegetation formation in *Streptococcus* and *S. aureus* infections in IE.

### Alterations in postoperative plasma proteome reveal NDUFB4 as a potential biomarker for IE

To investigate the proteomic signatures associated with postoperative infection clearance, comparative proteomic analysis was performed among preoperative IE (collected at admission), postoperative IE (collected within three to seven days postoperatively), and non-IE plasma samples (collected at admission), following the methodology of Fig. 5A. We identified a decrease in proteins including

NADH:ubiquinone oxidoreductase subunit B4 (NDUFB4), pro-platelet basic protein (PPBP), and immunoglobulins, as well as an increase in proteins including ALB, FTL, and gamma-glutamyltransferase 1 (GGT1) in postoperative IE (Fig. 7A). Pathway enrichment analysis revealed the downregulation of pathways including immunoglobulin production and neutrophil degranulation in postoperative IE, while pathways involving intermediate filament organization, glutathione catabolic process, and vesicle-mediated transport were upregulated postoperatively (Fig. 7A).

We found that NDUFB4, an accessory subunit of complex I, was the most significantly downregulated protein in postoperative IE samples (Fig. 7A) and could reflect severe infection (Fig. 7B, C). Complex I provides energy for resisting infection[34]. We found that the expression level of NDUFB4 was significantly associated with neutrophil ratio (*r* = 0.29, *p* = 1.4e-04), neutrophil count (*r* = 0.23, *p* = 0.0025), and white blood cell (WBC) count (*r* = 0.19, *p* = 0.013), suggesting an underlying role of NDUFB4 in regulating inflammation (Fig. 7D). Subsequently, we found that proteins significantly correlated with NDUFB4 were enriched in pathways containing acute phase response, complement and coagulation cascades, and glycolysis gluconeogenesis (Fig. 7E), implying a close association between NDUFB4 and functions to inflammation and metabolism in IE (Fig. 7F). In both Cohort 2 and Cohort 3, we confirmed that NDUFB4 was consistently upregulated in preoperative IE compared to non-IE plasma samples (Supplementary Fig. 2I, J). In conclusion, we hypothesize that NDUFB4 might contribute to energy synthesis to resist infection and has emerged as a candidate biomarker for IE, indicating infection severity and clearance dynamics (Fig. 7G).

Overall, Fig. 7H graphically represents the key findings of this study spanning the early diagnosis, pathophysiology, and treatment of IE, highlighting potential biomarkers and therapeutic targets for IE.

## Discussion

IE has high morbidity and mortality and substantial clinical heterogeneity. Despite preliminary proteomic studies in IE[7,35], integrated analyses that encompass large-scale proteomic and matched clinical data remain a critical gap, essential for biomarker discovery and personalized treatment development. In this study, high-resolution MS-based proteomic analyses of plasma and vegetation samples were conducted on a total of 402 IE patients and 264 non-IE controls from the discovery cohort (Cohort 1) and two external validation cohorts (Cohort 2 and Cohort 3). This integrated analysis delineates the molecular landscape of IE, facilitating the development of early diagnostic approaches and targeted therapeutic strategies.

The plasma proteomic analysis comparing IE and non-IE samples revealed specific biological function alterations associated with IE and enabled the development of a diagnostic prediction model with robust validation performance. IE was characterized by pathways associated with inflammatory response and cell motility, while those related to plasma lipoprotein clearance and protein ubiquitination were downregulated. In addition, to identify effective biomarkers for early diagnosis, we developed an unbiased ML-based diagnostic prediction

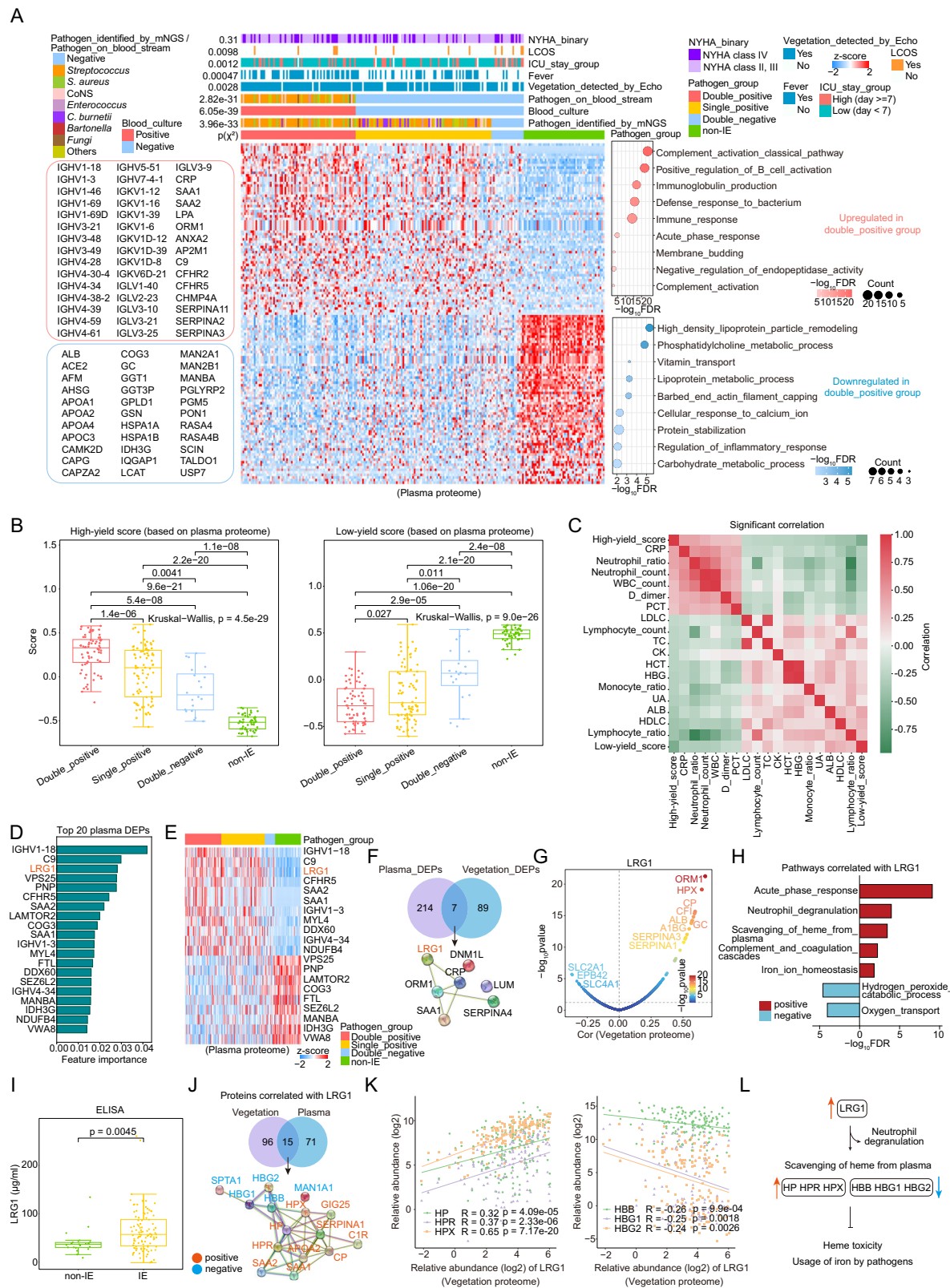

model for distinguishing IE from non-IE cases, utilizing proteomic data from prospectively collected plasma samples at admission. This model demonstrated reliable discriminatory performance, achieving an overall AUC of 0.97 in Cohort 1 and being independently validated in the two external validation cohorts with overall AUC values of 0.98 (Cohort 2) and 0.94 (Cohort 3), which surpassed those of inflammatory biomarkers[36,37], cardiac imaging[38], and mNGS[39], typically ranging from

0.6 to 0.9. This underscores the significant value of plasma protein examination as a rapid, accurate, and noninvasive diagnostic tool in challenging cases. This model identified 10 plasma proteins, including an increase in known inflammatory molecules (CRP, SAA1, and SAA2), actin-related proteins (ACTB and ACTG1), superoxide dismutase (SOD3), and LRG1 that is associated with early neutrophilic granulocyte differentiation and activation[27,33]; and a decrease in HDL (APOC1 and

**Fig. 5 | Proteomic alterations associated with infection severity in IE. A** Heatmap and bubble chart illustrating the two panels of plasma proteins with gradually increasing (top) or decreasing (bottom) abundance from double-positive to single-positive, double-negative, and non-IE category, along with pathway enrichment (right) and the proteins included (left). Pearson's two-sided $\chi^2$ test was used to evaluate the associations between the pathogen group and 8 variables. **B** Boxplots showing the high-yield and low-yield scores among the double-positive ($n = 71$), single-positive ($n = 85$), double-negative ($n = 20$), and non-IE ($n = 50$) categories. **C** Heatmap showing correlations among the inflammatory indicators. **D**, **E** Bar plot and heatmap illustrating the top 20 proteins for feature importance. **F** Venn diagram and STRING analysis illustrating the 7 proteins altered in both plasma and vegetation proteomes, along with their PPIs. **G** Scatter plot showing the correlations between LRG1 and the vegetation proteome. **H** Bar chart displaying pathways

enriched by the vegetation proteins correlated with LRG1 ($R > 0.3$ or $R < -0.2$; $p < 0.05$). **I** Box plot of LRG1 concentrations in IE ($n = 105$) versus non-IE ($n = 21$) samples measured by ELISA. **J** Venn diagram and STRING analysis illustrating the 15 proteins correlated with LRG1 in both plasma and vegetation proteomes ($R > 0.3$ or $R < -0.2$; $p < 0.05$), along with their PPIs. **K** Scatter plot depicting the correlation between the relative abundance (log2) of LRG1 and haptoglobin (HP and HPR), hemopexin (HPX), and hemoglobin (HBB, HBG1, and HBG2). **L** Schematic of the potential mechanisms mediated by LRG1 influencing infection severity in IE. For boxplots (**B** and **I**), the central line represents the median, the box indicates the IQR, and whiskers extend to $1.5 \times$ IQR, with outliers as individual points. The unpaired two-sided Wilcoxon rank-sum test and Kruskal–Wallis test were used for statistical tests. The two-sided Spearman's correlation was used for figures (**C**, **G**, **H**, **J**, and **K**), with $p$-values adjusted for FDR. Source data are provided as Source Data files.

APOA2) and toll interacting protein (TOLLIP). Notably, LRG1 emerged as a potential biomarker associated with infection severity, possibly playing a role in suppressing heme toxicity and resisting pathogen infection. Overall, the identified panel of 10 proteins presents promising early diagnostic biomarkers that could enhance clinical decision-making.

Prognostic assessment in IE remains clinically challenging due to insufficient biomarkers and unclear progression mechanisms[10]. Through integrated plasma and vegetation proteomics, we identified disease-relevant protein networks associated with clinical outcomes. Protein networks characterized by pathways related to glycometabolism, amino acid metabolism, and adhesion exhibited significant positive correlations with IE risk factors and adverse events, suggesting that these pathways may play a crucial role in the disease pathogenesis and could serve as potential targets for therapeutic intervention. In addition, a protein network showed significant negative correlations with crucial progression features, including microbiological persistence, detectable vegetation or abscess, incidence of valve surgery, and NYHA functional classification, typically used to assess heart failure, with the correlation with NYHA classification being the most pronounced. Valve dysfunction leading to heart failure is known to be the most common complication of IE and the primary indication for surgical intervention in these cases[13].

Interestingly, we observed significant differences between NYHA class IV patients and those in NYHA class II and III in terms of both clinical and proteomic characteristics. Specifically, factors such as advanced age, valve damage, liver dysfunction (defined by elevated levels of AST, DBIL, etc.), high postoperative adverse events (e.g., reoperation, LCOS, long-time ICU stay), and high mortality (e.g., D30) were significantly enriched in NYHA class IV patients, indicating the need for graded treatment for NYHA class IV patients in IE. Liver dysfunction has a prognostic impact on the outcomes of patients with heart failure among various diseases, but the underlying molecular mechanisms responsible for this phenomenon remain elusive[40,41]. In addition to abnormal liver function test results, proteomic analysis demonstrated decreased levels of liver-derived proteins, including complement (e.g., C3 and C4), HDL (e.g., APOA1 and APOA2), and serine protease inhibitors (e.g., SERPINC1 and SERPINA6) in NYHA class IV patients, while pathways related to neutrophil degranulation, cell adhesion, and proteolysis were upregulated. These biological alterations reportedly can increase vulnerability to infections[42], liver complaints[43], and tissue destruction[44]. These proteomic alterations could impede bacterial clearance and contribute to endocardial damage and pathogen adhesion, ultimately leading to the progression of IE. In conclusion, this study provides molecular insights suggesting that liver dysfunction may exacerbate the deterioration of IE in patients with severe heart failure, and highlights complement, HDL, and serine protease inhibitors as potential therapeutic targets for IE, especially for NYHA class IV patients.

Subsequently, we established three prognostic models for IE utilizing the blood-derived characteristics of CBMs, measured by clinical blood test, and plasma proteome, namely, CBM-only, proteome-only, and proteome + CBM, to distinguish between NYHA class IV patients and those in NYHA class II and III. The proteome + CBM model yielded the highest AUC of 0.873 in the hold-out test set of the discovery cohort (Cohort 1), validated in the external validation cohort (Cohort 3) with an overall AUC of 0.829; thus, equally effective or even superior discrimination is achieved compared to that of previous prognostic models based on clinical features[45]. These findings imply the promise of noninvasive blood detection as an effective method for distinguishing high-risk IE patients at an early stage. The model involved 4 CBMs reflecting abnormal liver and kidney function and 11 plasma proteins comprising upregulated proteins of ADIPOQ, CST3, and CAMKK2, as well as downregulated proteins of APOA1, FETUB, TLN1, GC, PON1, VPS25, ITIH1, and G3BP1. Among these proteins, consistent with previous research, CST3 and APOA1 have been identified as potential prognostic factors in IE[7,8]. The severity of rheumatoid arthritis is reported to correlate directly with circulating adiponectin levels. The increased ADIPOQ suggests a similar environment with sustained inflammation in NYHA class IV patients, resulting in endocardial degradation[46]. CAMKK2 is reported to be upregulated in primary human osteoarthritis and associated with elevated levels of proapoptotic and catabolic proteins[47]. FETUB is known to be reduced in patients with more severe infections[48]. Proteins TLN1, GC, PON1, VPS25, and ITIH1 play a role in modulating inflammation and tissue repair, suggesting their protective role in IE. Some RNA-binding proteins can recognize and attack non-self RNA[49], indicating that G3BP1 is likely related to inhibiting bacterial transcription in IE, and its downregulation might cause RNA metabolic disturbances during the progression of IE. Overall, this study emphasizes the significance of plasma proteome in disease severity prediction and provides important insights into biomarker exploration in IE.

In addition to valve damage and impaired cardiac function, the progression of IE is closely related to the degree and type of pathogen infections. Consequently, we further investigated the underlying pathological mechanisms associated with pathogen infections. We revealed a significant difference in the proteomic profiles associated with infection severity between IE patients with insufficient microbiological evidence and non-IE individuals, which supports the diagnosis of IE patients with insufficient microbiological evidence. The yield of detectable pathogens was determined to be associated with infection severity, as indicated by elevated levels of clinical infection markers in IE. Moreover, pathways related to immunoglobulin production, inflammation, and necroptosis were increased in the category with high detectable pathogen yield, while those related to lipid metabolism, transportation, and adhesion were reduced. Moreover, we found that proteins involved in the immune and inflammatory reactions, such as NDUFB4, PPBP, and immunoglobulins, were reduced after surgery, while proteins involved in carbohydrate metabolism and transportation, such as GGT1 and FTL, were increased.

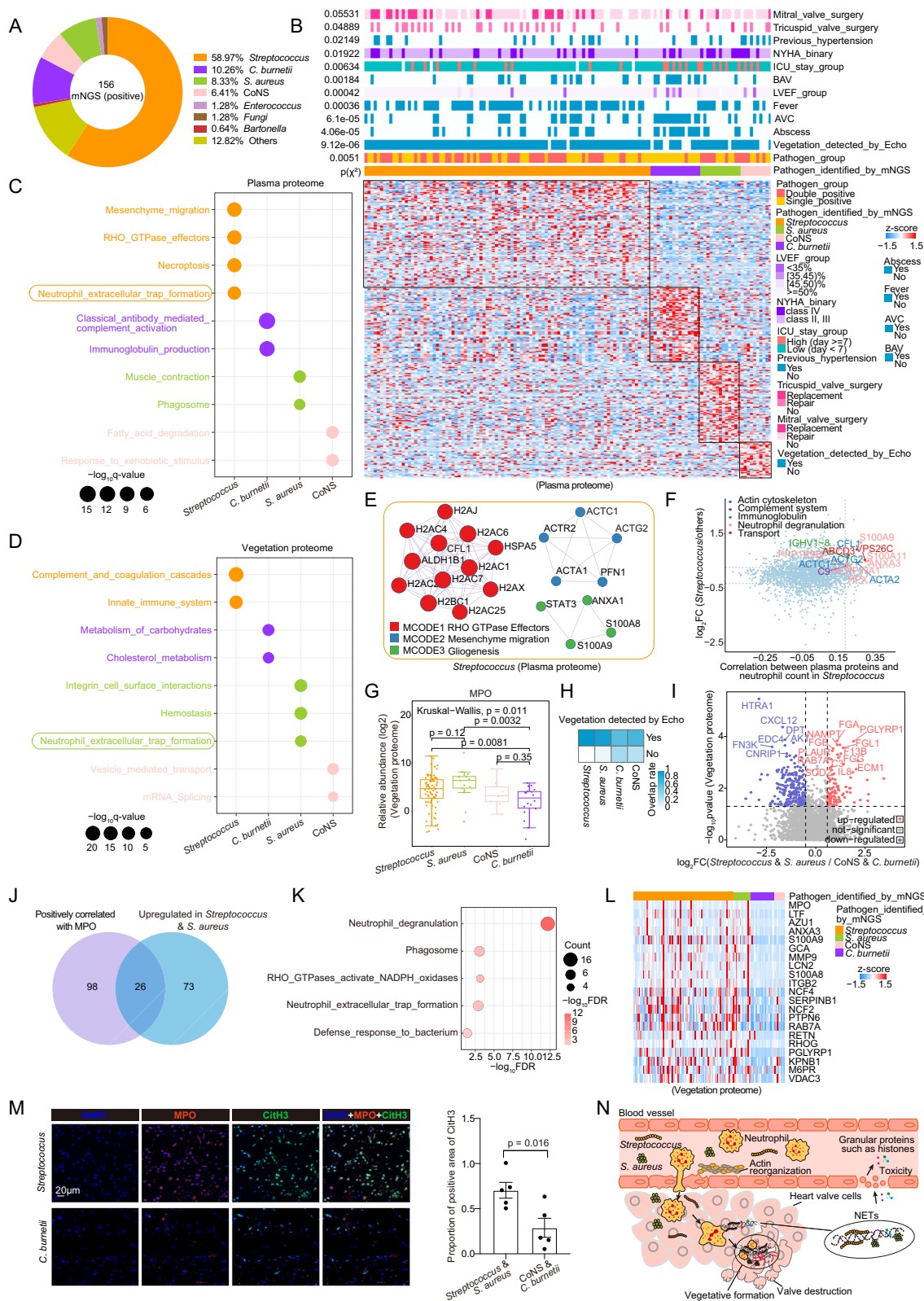

These results deepen our understanding of the pathophysiology of functional alterations associated with infection severity in IE.

In addition, we illustrated the proteomic landscape reflecting pathogen-specific host response patterns for several classical pathogens of IE, including *Streptococcus*, *C. burnetii*, *S. aureus*, and CoNS. Most patients infected with *Streptococcus* and *S. aureus* had fever and vegetation detectable by Echo, while most patients infected with *C.*

*burnetii* had non-fever and abscess. *C. burnetii* was characterized by antibody production and metabolism-related pathways, indicating that patients infected with *C. burnetii* might be in a chronic infection state and metabolic reprogramming. Most patients infected with CoNS were found to have previous hypertension and were enriched in pathways including fatty acid degradation, transport, and mRNA splicing, suggesting underlying alterations in energy metabolism and gene

**Fig. 6 | Proteomic analysis revealing pathogen-specific host response patterns.** **A** Doughnut chart showing pathogen proportions in 156 IE patients with positive mNGS results. **B** Heatmap showing differential plasma profiling across the four pathogen infections. Pearson's two-sided χ² test evaluated associations between pathogen types and 12 variables. **C**, **D** Bubble charts showing enriched pathways for the four pathogens in the plasma and vegetation proteomes. **E** PPIs depicting the top three significant molecular complex detection (MCODE) components of the *Streptococcus* group. **F** Scatter plot showing plasma proteins according to their correlation with neutrophil count in the *Streptococcus* group (*x*-axis) and abundance differences from other groups (*y*-axis). **G** Boxplot depicting the relative abundance of the vegetation protein MPO (log2) across *Streptococcus* (*n* = 75), *C. burnetii* (*n* = 18), *S. aureus* (*n* = 13), and CoNS (*n* = 9) groups. The central line represents the median, the box indicates the IQR, and whiskers extend to 1.5 × IQR, with outliers as individual points. **H** Overlap rate of detectable vegetation in each

group. **I** Volcano plot showing vegetation DEPs between the two groups. **J** Venn diagram illustrating 26 vegetation proteins correlated with MPO (*R* > 0.5, *p* < 0.05) and upregulated in the *Streptococcus* & *S. aureus* group (FC > 1.5, *p* < 0.05). **K**, **L** Bubble chart and heatmap displaying the enriched pathways and the proteins included for the 26 proteins. **M** IF staining of MPO and citrullination of histone H3 (CitH3) in representative *Streptococcus* and *C. burneti* infections. Scale bar, 20 μm. Boxplot showing the proportion of CitH3-positive areas in the two groups (*n* = 5). Data are presented as mean ± SEM. The unpaired two-sided Student's *t* test was used. **N** Summary of the harmful effects of excessive NET release in patients infected with *Streptococcus* & *S. aureus*. For figures (**F**, **G**, **I**, **J**), the unpaired two-sided Wilcoxon rank-sum test and Kruskal–Wallis test were used for differential analysis. For figures (**F**, **J**), the two-sided Spearman's correlation was used, with *p*-values adjusted for FDR. Source data are provided as Source Data files.

expression patterns within these patients. Notably, we found that *Streptococcus* and *S. aureus*, the two most prevalent causative pathogens in IE patients, shared a common pathway characterized by the NET formation, and we subsequently demonstrated that NETs were promising therapeutic targets in IE patients with *Streptococcus* and *S. aureus* infections, utilizing both integrated proteomic analysis and immunofluorescence quantitative assays. Specifically, IE patients with *Streptococcus* and *S. aureus* infections showed a significant increase in signs of NETs, including proteins MPO, citrullination of histone H3 (CitH3), IL-8, and neutrophil cytosolic factor 2 (NCF2), as well as pathways related to actin cytoskeleton dynamics, neutrophil degranulation, and oxidative stress. These findings demonstrated that NETs promote and expand vegetation formation in human subjects, providing critical evidence that supported earlier observations from mouse models indicating that *Streptococcus* and *S. aureus* induce NETs while CoNS do not[50,51]. Overall, these findings support that targeting NETs might be a valuable therapy for inhibiting vegetation formation in patients infected with *Streptococcus* and *S. aureus*.

This study has several limitations. First, we only enrolled surgical patients to investigate the vegetation proteome in IE. This selection bias inherent in our study design may limit the generalizability of identified plasma biomarkers to the broader population, particularly those receiving conservative treatment. Second, our findings may not fully apply to prosthetic valve IE, as most participants had native valve infections. Third, while this study highlights the need for risk stratification of NYHA class IV patients due to their higher postoperative risk, our prognostic model based on NYHA functional classification may not capture all at-risk subgroups in IE. Fourth, the predominance of streptococcal infections in our cohort, consistent with the microbiological spectrum observed in developing countries, may limit the applicability of our findings to regions where pathogens such as *S. aureus*, CoNS, and enterococci are also commonly encountered. Finally, our pathophysiological hypotheses were not validated in a patient-derived xenograft model, primarily due to the current lack of established, reliable models for IE[52]. These limitations underscore the need for broader validation. Nevertheless, this integrated analysis of the plasma and vegetation proteomes provides valuable insights into the pathophysiology of IE and can inform biomarker identification and the development of effective stratified therapies.

## Methods
### Study design and specimen acquisition
**Patient recruitment.** Initially, we prospectively collected a preliminary cohort of 4312 patients with heart valve disease who were hospitalized between August 2020 and May 2023 at Guangdong Provincial People's Hospital. Among them, a total of 1564 patients initially underwent blood culture due to clinical suspicion of IE. Both IE patients and non-IE individuals (control group) included in the discovery cohort (Cohort 1) and the external validation cohort (Cohort 2) of this study were selected from the above initial cohort, as illustrated in the flowchart

(Supplementary Fig. 1A). All participants were diagnosed based on the 2023 Duke-ISCVID Criteria[5].

To ensure a homogeneous study population and minimize confounding factors, we implemented rigorous inclusion and exclusion criteria. The inclusion criteria were as follows: (1) definite IE and non-IE individuals diagnosed according to the 2023 Duke-ISCVID Criteria; (2) patients with informed consent; (3) age ≥18 years; (4) patients with surgical treatment; and (5) patients with definitive microbiological assessment results, including blood culture, vegetation culture, and/or pathogen identification through mNGS of blood or vegetation samples. The exclusion criteria were as follows: (1) patients unable to tolerate surgery due to severe sepsis or septic shock; (2) severe end-stage liver disease (Child-Pugh class C); (3) known pregnancy; and (4) patients with immunosuppressive or immunomodulatory therapy. The non-IE individuals in our study refer to patients with heart valve disease who exhibited clinical features suggestive of IE (e.g., fever or heart murmur) or echocardiographic evidence of valvular lesions but were ultimately ruled out for IE (Supplementary Fig. 1A). The underlying conditions among the non-IE individuals primarily included rheumatic heart disease, degenerative valve disease, non-bacterial thrombotic endocarditis, bacteremia, and Behçet's disease.

Ultimately, a total of 522 individuals were ultimately enrolled from the initial study population, comprising a discovery cohort (Cohort 1) with 238 IE patients and 100 non-IE individuals, and an external validation cohort (Cohort 2) with 92 IE patients and 92 non-IE individuals. In addition, in accordance with the predefined inclusion and exclusion criteria, we additionally collected an external validation cohort (Cohort 3) consisting of 72 IE patients and 72 non-IE individuals from an external center, Heyuan People's Hospital.

**Ethics statement.** The Research Ethics Committee of Guangdong Provincial People's Hospital, Guangdong Academy of Medical Sciences, approved this study (Approval No. KY-N-2022-003-03), and written informed consent was obtained from all participants before enrollment. All procedures were conducted in accordance with the ethical principles outlined in the Declaration of Helsinki.

**Sample collection and processing.** To comprehensively characterize the proteomic landscape of IE, we conducted in-depth proteomic analyses in the discovery cohort (Cohort 1), using three types of samples obtained from the 238 patients with IE: (i) Preoperative plasma samples (*n* = 202), collected at the time of hospital admission, before definitive diagnosis; (ii) Postoperative plasma samples (*n* = 23), collected within three to seven days following surgical intervention; (iii) Vegetation samples (*n* = 158) collected intraoperatively. Additionally, to develop an early diagnostic prediction model for IE based on plasma protein biomarkers, we also included 100 admission plasma samples from non-IE individuals as the control group. In the two external validation cohorts of Cohort 2 and Cohort 3, admission

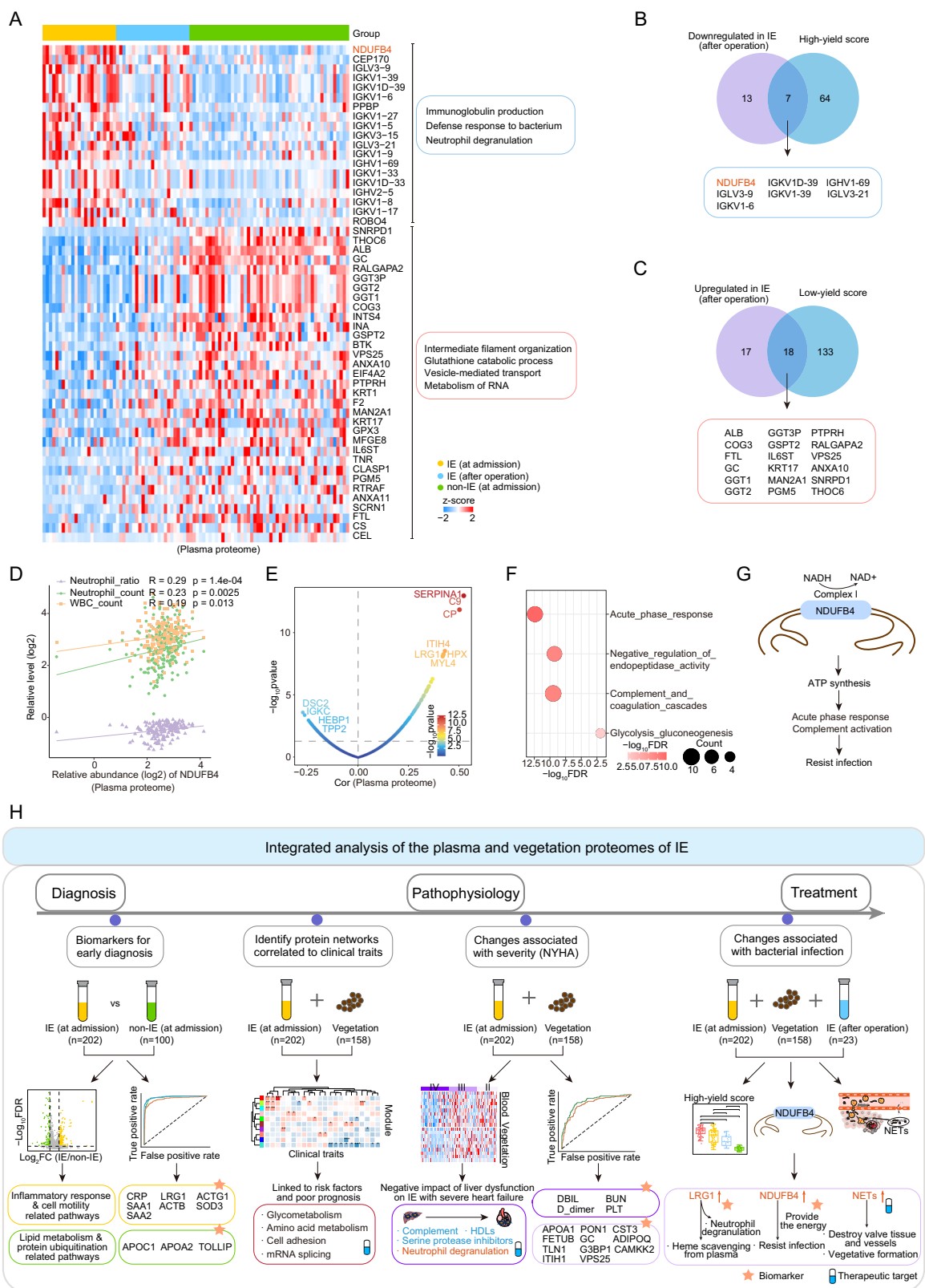

plasma samples were collected from each participant in accordance with standardized protocols (Supplementary Fig. 1A).

Plasma sample collection was performed following the Clinical and Laboratory Standards Institute (CLSI) guidelines GP41-A6[53]. Standardized venipuncture procedures and equipment were used to minimize pre-analytical variability. For each collection, 3 ml of peripheral blood was drawn into EDTA-treated tubes to prevent coagulation and preserve protein integrity. Blood samples were processed within 2 h of collection to minimize ex vivo changes in the proteomic profile. Plasma was separated by centrifugation at $1500 \times g$ for 10 min at 4 °C. The plasma aliquots were immediately stored at −80 °C to preserve protein integrity. Vegetation samples were collected intraoperatively following established procedures described in previous studies[35,54], to conduct vegetation proteome research for IE.

**Fig. 7 | Plasma proteomic alterations following cardiac surgery in IE. A** Heatmap showing two plasma protein panels with gradually decreasing (top) or increasing abundance (bottom) from preoperative IE to postoperative IE and non-IE group, along with pathway enrichment (right). **B** Venn diagram illustrating 7 plasma proteins associated with the high-yield score and decreased in the postoperative IE samples. **C** Venn diagram illustrating 18 plasma proteins associated with the low-yield score and increased in the postoperative IE samples. **D** Scatter plot showing the correlation between the relative levels (log2) of NDUFB4 (x-axis) and inflammatory indicators (y-axis). **E** Scatter plot displaying the correlations of NDUFB4 with other proteins in the plasma proteome. **F** Bubble chart showing enriched pathways in the proteins correlated with NDUFB4 ($R > 0.2$; $p < 0.05$). **G** Schematic of the potential role of NDUFB4 in facilitating resistance to infection in IE. **H** Graphical summary of key molecular findings, highlighting potential biomarkers and treatment targets for IE, spanning early diagnosis to pathophysiology and treatment. For figures (**D–F**), the two-sided Spearman's correlation was used, with p-values adjusted for FDR. Source data are provided as Source Data files.

All patient samples were obtained with written informed consent. Both plasma and vegetation samples were stored at -80 °C until proteomic analysis to maintain sample integrity. To minimize dietary influences on plasma proteomic composition, all samples were collected after an overnight fast of 8–12 h under standardized conditions. Each sample was assigned a unique identifier and recorded in a secure, access-controlled database, along with metadata, including collection time, processing timeline, and storage details. The sample collection and processing methods we employed are widely accepted and commonly used in clinical disease research and proteomic studies[55,56].

**Sample size calculation and statistical power analysis.** Before study initiation, we conducted a thorough statistical power analysis to determine the minimum sample size required to detect a clinically significant effect. Assuming a two-tailed Wilcoxon–Mann–Whitney test, a sample size of at least 134 participants was necessary to detect a medium effect size (Cohen's $d = 0.5$) with 80% power at a conventional significance level ($\alpha = 0.05$). When employing a more stringent threshold ($\alpha = 0.01$), the required sample size increased to 200 participants. To account for potential dropouts or unusable samples due to unforeseen circumstances, we prospectively recruited a slightly larger cohort and applied strict inclusion and exclusion criteria during subject selection. This analysis was conducted using G*Power software (v3.1)[57]. Ultimately, we incorporated 238 IE patients and 100 non-IE individuals in the discovery cohort. In addition, two external validation cohorts comprising 184 and 144 individuals, respectively, were incorporated to further validate the robustness and generalizability of our findings.

**Cell line**

The HEK293T cell line (ATCC CRL-11268, RRID: CVCL_QW54) used for quality control was obtained from the Chinese Academy of Sciences and cultured in Dulbecco's Modified Eagle Medium (DMEM; Gibco), supplemented with 10% fetal bovine serum (FBS; Gibco) and 1% penicillin-streptomycin (Sigma-Aldrich). Cells were maintained at 37 °C in a humidified atmosphere with 5% $CO_2$. All cell lines underwent routine testing for mycoplasma contamination and were authenticated using short tandem repeat profiling.

**Peptide preparation for MS analysis**

**Protein extraction and tryptic digestion.** For vegetation samples, approximately 50 mg of vegetation sample was homogenized separately in an appropriate volume of urea lysis buffer (8 M urea, 100 mM Tris hydrochloride, pH 8.0) containing PMSF protease inhibitor (Thermo Fisher Scientific). Lysates were centrifuged at $16,000 \times g$ for 15 min at 4 °C for clarification, and the NanoDrop One A280 method was applied to measure the protein concentration. Next, protein samples were reduced with a final concentration of 5 mM dithiothreitol (DTT) for 30 min at 56 °C, which were then alkylated with 20 mM iodoacetamide (IAA) for 30 min at 25 °C in the dark. After incubation, DTT was added to the samples to a final concentration of 5 mM and maintained for 15 min in the dark[58]. Protein samples were purified using the filter-aided sample preparation (FASP) method[59]. Protein samples were then digested with trypsin at 37 °C overnight (-16 h) with an enzyme-to-protein mass ratio of 1:25[60]. Tryptic peptides were dried using SpeedVac.

For plasma samples, 2 µL of plasma sample was mixed with 98 µL of 50 mM $NH_4HCO_3$, and the protein was inactivated at 95 °C for 3 min. The plasma sample was cooled to room temperature, and then trypsin was added at an enzyme-to-protein mass ratio of 1:25 for digestion at 37 °C overnight (-16 h). After the overnight incubation, 10 µL of aqueous ammonia was added to each sample to quench the digestion reaction, and the samples were then dried using SpeedVac. After drying, the tryptic peptides were redissolved in 100 µL of 0.1% formic acid (FA) and then desalted on reversed-phase C18 SPE columns (Waters tC18 SepPak, 200 mg) and dried using SpeedVac.

**Liquid chromatography-tandem mass spectrometry (LC-MS/MS) acquisition.** For plasma samples, the digested peptides were analyzed using Q Exactive HF-X Hybrid Quadrupole-Orbitrap mass spectrometer coupled to an EASY-nLC 1200 HPLC system (Thermo Fisher Scientific). The dried peptide samples were redissolved in 0.1% FA in water and loaded on a 9 cm × 75 µm internal diameter column with 1.9 µm ReproSil-Pur C18-AQ beads (Dr Maisch GmbH) over a 10 min gradient (Solvent A: 0.1% FA in water; Solvent B: 0.1% FA in 80% acetonitrile) at a constant flow rate of 600 nL/min. The eluted peptides were ionized via electrospray at 2 kV and introduced into the mass spectrometer. MS acquisition was performed in data-independent acquisition (DIA) mode. Full MS1 scans were recorded in the m/z range of 300–1400 at a resolution of 30,000 (Orbitrap analyzer), with an AGC target of 4e5 and a maximum injection time of 50 ms. For DIA MS2 scans, the AGC target was 5e4, the injection time was 22 ms, and the resolution was 15,000. Fragmentation was conducted using higher-energy collisional dissociation (HCD) with a normalized collision energy (NCE) of 27%. The inject ions for all available parallelizable time options was enabled. All MS data were acquired in profile mode with the default charge state for MS2 set to 3. Raw data files were saved in.RAW format.

For the vegetation samples, the digested peptides were analyzed using an Orbitrap Fusion Lumos mass spectrometer coupled with an EASY-nLC 1200 HPLC system (Thermo Fisher Scientific). The dried peptides were redissolved in 0.1% FA in water and loaded onto a 2 cm × 100 µm internal diameter trap column (120 Å pores; 3 µm C18 particles; SunChrom) and then separated on a 30 cm × 150 µm internal diameter silica microcolumn (120 Å pores; 1.9 µm C18 particles; SunChrom) with a 150 min gradient (Solvent A: 0.1% FA in water; Solvent B: 0.1% FA in 80% acetonitrile) at a flow rate of 600 nL/min. The eluted peptides were ionized by electrospray at 2 kV for mass spectrometric analysis. MS analysis was performed in data-dependent acquisition (DDA) mode coupled with field asymmetric ion mobility spectrometry (FAIMS). The FAIMS compensation voltages were set to −40, −60, and −80 V. MS parameters were configured as follows: full MS1 scans were acquired at a resolution of 120k (at 200 m/z) with an AGC value of 5e5, maximum injection time of 50 ms, and scan range of 300-1800 m/z. MS2 scans were performed using HCD fragmentation with ion trap detection (rapid scan rate, 1.6 m/z isolation window, 10 ms maximum injection time, AGC target of 1e4, and 30% normalized collision energy). Dynamic exclusion was set to 45 s with a 1 s cycle time[60]. MS data were recorded in.RAW format.

**MS data analysis**

**Peptide identification and label-free protein quantification.** All MS raw plasma and vegetation proteomic data files were processed with

Firmiana, a one-stop proteomic cloud platform[61]. The DIA data were searched against the UniProt human protein database using FragPipe (v12.1) with MSFragger (v2.2)[62] configured in the Firmiana computational platform. Search parameters were configured with a precursor mass tolerance of 20 ppm and a fragment ion tolerance of 0.05 Da. A maximum of two missed tryptic cleavages was permitted. Carbamidomethylation of cysteine was defined as a fixed modification, and the N-terminal acetylation and methionine oxidation were specified as variable modifications. Precursor ion charge states were limited to +2, +3, and +4. All data were additionally searched against a decoy database to ensure protein-level identification at a 1% false discovery rate (FDR). For spectral library generation, DDA data were processed using SpectraST software, incorporating 327 reference spectral libraries[63]. DIA data analysis was conducted using DIA-NN (v1.8.1)[64], with default parameters: precursor FDR (5%), log level (1), mass accuracy (20 ppm), MS1 accuracy (10 ppm), scan window (30), protein grouping by genes, and robust LC (high accuracy) quantification strategy. Peptide-level quantification was derived from the averaged peak areas of fragment ions across matching entries in the spectral libraries. Protein quantification was conducted using the MaxLFQ algorithm[65], with delayed normalization and maximal peptide ratio extraction. Final protein intensities were normalized using the fraction of total (FOT) method, calculated from the cumulative peak areas across all identified proteins.

The DDA data were searched against the National Center for Biotechnology Information (NCBI) human RefSeq protein database using the Mascot 2.4 search engine configured within Firmiana, with the following parameters: 20 ppm precursor mass tolerance and 0.5 Da fragment mass tolerance, up to two missed cleavages allowed, carbamidomethylation (C) as a fixed modification, and N-terminal acetylation plus methionine oxidation as variable modifications. Protein identification confidence was controlled through a target-decoy approach, maintaining peptide- and protein-level FDR below 1%. A percolator was employed to calculate q-values, and peptide-spectrum matches (PSMs) with ion scores < 20 or peptide lengths shorter than seven amino acids were filtered out. For enhanced stringency, all PSMs across fractions were pooled, and protein-level FDR was enforced based on the parsimony principle, adjusting q-values until global protein FDR fell below 1%. Label-free quantification was performed within Firmiana using mzXML-converted MS data. Then, each identified peptide was retrieved according to the identification information of MS1 to obtain the extracted-ion chromatogram (XIC), and the abundance was estimated by calculating the area under the extracted XIC curve. A nonredundant peptide list was used to assemble the proteins according to the parsimony principle. Protein abundance was then estimated using the intensity-based absolute quantification (iBAQ) method, in which protein intensity was normalized by the number of theoretically observable peptides. We employed the FOT metric to enable cross-sample normalization and relative abundance comparisons.

**Quality control of the MS data.** To monitor the stability and performance of the LC-MS/MS system, digested peptides from HEK293T cells were analyzed every two days between IE sample runs as quality control (QC) standards. In addition, blank injections of 0.1% FA were performed after every five sample injections to assess and prevent potential carry-over effects. In addition, all the plasma samples were randomly distributed in fifteen different batches for the peptide preparation experiments. Every batch contains a pooled sample, i.e., a mixture of all peptide samples, as the control sample for aligning data from different batches and evaluating the quantitative accuracy. Spearman's correlation coefficients of all quality-control runs are displayed in our study (Supplementary Fig. 2A, B), demonstrating the robustness of our QC strategy.

**Missing value imputation.** A threshold of less than 30% missing values was set for each protein to ensure that sufficient data were obtained from each sample for imputation. K-nearest neighbor (KNN) imputation was applied to impute the missing values using the Python module of sklearn.impute.KNNImputer (v1.1.3) with default parameters. Ultimately, a total of 2314 plasma proteins and 3694 vegetation proteins were included in further statistical analyses.

## Global proteomics data analysis
**Construction of the machine learning classification models.** We built a machine learning (ML) model with the extreme gradient boosting (XGBoost) algorithm[16] to develop the diagnostic prediction model for distinguishing IE patients from non-IE individuals, as well as the prognostic model for distinguishing NYHA class IV patients from those in NYHA class II and III. This was accomplished utilizing the plasma proteomic data from the discovery cohort (Cohort 1) and two external validation cohorts (Cohort 2 and Cohort 3), implemented in Python (v3.9.7) with the pandas (v1.2.3), scikit-learn (v1.1.3), and XGBoost (v1.5.0) modules. The schematic workflow for the development and evaluation of the ML model was shown in Supplementary Fig. 1B and Fig. 4A. Specifically:

**Dataset and model training.** The diagnostic model was trained using plasma proteomic data from 202 IE and 100 non-IE plasma samples collected at admission from Cohort 1 (Supplementary Fig. 1B). For the prognostic model, clinical blood test and proteomic data from 194 IE plasma samples collected at admission in Cohort 1 were utilized (Fig. 4A). During model derivation, the discovery cohort was randomly divided into a training set (70%) and a hold-out test set (30%) using a stratified randomized split approach, ensuring a balanced distribution of key variables across both sets. Importantly, all model development processes, including feature selection, hyperparameter tuning, and final model training, were exclusively performed on the training set to prevent potential data leakage. Once the final models were established, they were locked for rigorous testing and validation. This procedure ensures the test set remained completely independent and was not involved in any feature selection or model training processes, thereby maintaining the authenticity and reliability of the model's evaluation on the hold-out test set. The locked models were then evaluated internally on the hold-out test set and externally validated on independent, unseen datasets (Cohort 2 and Cohort 3) to assess their robustness and generalizability (Supplementary Fig. 1B).

**Feature selection.** Feature selection was performed on the training set to identify the most significant features contributing to the model's ability to distinguish between different classes. Firstly, we identified the differentially expressed proteins (DEPs) between different classes using the statistical test methods. After determining DEPs, we employed an ensemble learning scheme for further feature selection, which has been widely applied in previous studies[66,67]. Specifically, we employed an ensemble of ML models to evaluate feature importance, including Random Forest, Logistic Regression, and Support Vector Classifier (SVC) implemented in scikit-learn (v1.1.3), along with XGBoost (v1.5.0), LightGBM (v3.2.1), and CatBoost (v1.1.1). Each model was trained on the entire training set to assess the relative contribution of each feature to the model's predictions. The method of determining feature importance varied across models: tree-based models (XGBoost, Random Forest, LightGBM, CatBoost) measured importance based on a feature's frequency in making split decisions at each tree node, while linear models (SVC, Logistic Regression) relied on feature coefficients. To reduce the error and generate a more robust feature importance ranking, this process was performed with a 5-fold cross-validation procedure to compute the average importance scores of each feature. The final feature importance ranking was determined by aggregating the results from six feature-ranking algorithms using a

weighted voting approach. All models were implemented in Python using the scikit-learn module. By quantifying the relative importance of each feature using this ensemble method, we can accurately and robustly identify key features that most significantly contribute to the model's ability to discriminate between classes. Finally, we further refined the selection of the previously identified top-ranked features for feature importance by employing the recursive feature elimination cross-validation (RFECV) method[68], implemented in Python using the 'scikit-learn' module. This process ensured a meticulous selection of the most critical features by systematically eliminating less informative features, leading to a more simplified and robust feature set.

**Hyperparameter optimization.** ML models with properly tuned hyperparameters significantly improve model's accuracy, training efficiency, and generalization, enabling state-of-the-art results. To optimize the hyperparameters of the XGBoost-based model, we employed Bayesian optimization using the tree-structured Parzen estimator (TPE) algorithm[69] with 5-fold cross-validation on the entire training set. The specific hyperparameters were defined as a dictionary with the following hyperparameters:

- Number of rounds (num_boost_round): This parameter determines the number of boosting rounds or trees to build. It is important to tune it properly to avoid overfitting. We explored values between 100 and 1000.
- Maximum depth of trees (max_depth): The maximum depth of the tree is defined as follows: We tested depths from 3 to 18.
- Gamma (gamma): Specifies the minimum loss reduction needed to make a split. The search space was between 0.1 and 1 in increments of 0.25.
- Regularization (L1 and L2: reg_alpha and reg_lambda, respectively): These add regularization terms to the objective function. We searched within a range of 0 to 1 in increments of 0.01 for both parameters.
- Column Sampling by Tree (colsample_bytree): Denotes the fraction of features to be randomly sampled for building each tree. We tested values ranging from 0.5 to 1 in increments of 0.05.
- Minimum Child Weight (min_child_weight): This parameter determines the minimum sum of the instance weights needed for a child. A search was performed between 0 and 10.
- Learning rate (eta): This makes the optimization more robust by shrinking the weights in each step. We used a range from 0.025 to 0.5 in increments of 0.05.
- Subsample: This denotes the fraction of observations to be randomly sampled for each tree. The range for this hyperparameter was between 0.5 and 1 in increments of 0.05.

This hyperparameter grid was used as input to a hyperparameter optimization algorithm to search for the optimal combination of hyperparameters that maximized the area under the ROC curve[70]. We defined the objective function objective(space), which took a hyperparameter space as input and returned a dictionary containing the loss and status values. This function was used in Bayesian optimization to evaluate the performance of the XGBoost model. For the training process, we utilized the 'gbtree' booster and the exact tree method. Furthermore, since our problem was a binary classification task, we employed the 'binary:logistic' objective function and used the area under the receiver operating characteristic curve (eval_metric: 'auc') as the evaluation metric.

Hyperparameter optimization was performed using the fmin() function from the HyperOpt package, which executed the objective() function with different hyperparameter values specified in the hyperparameter_grid. The fmin() function employed the TPE algorithm to suggest subsequent hyperparameters for evaluation, based on the outcomes of previous evaluations. The optimization continued until either the maximum number of evaluations specified by max_evals was

reached or convergence was achieved. The best_hyperparams dictionary returned by fmin() contained the optimal hyperparameters identified during the optimization, within the search space defined by the hyperparameter_grid, aiming to minimize the loss value as defined by the objective function.

Hyperparameter optimization was conducted over 200 iterations, and the best set of hyperparameters was selected based on the highest AUC value achieved during these iterations. After optimization, the model was retrained on the entire training set using the optimal hyperparameters.

**Model evaluation.** After model training, the final model was locked to ensure complete independence during model evaluation. This procedure maintained the complete independence of the hold-out test set, which was not involved in any stage of the model training process, preventing potential information leakage. Finally, the final locked model was then tested on the hold-out test set for internal evaluation, and the external independent, unseen datasets (Cohort 2 and Cohort 3) to evaluate its generalizability performance. Multiple evaluation metrics were employed to comprehensively assess the robustness, generalization performance, and prediction accuracy of the constructed classification models, including the ROC AUC, Confusion Matrix, Accuracy, Specificity, Precision, Recall, and F1 Score. These metrics provide a multifaceted perspective on model performance, encompassing various aspects of prediction accuracy, robustness, and reliability.

**Weighted gene correlation network analysis (WGCNA).** WGCNA was used to identify groups of coregulated genes in an unsupervised manner[18]. For the plasma proteome, we input 2314 proteins and 36 clinical traits of the 202 IE patients into WGCNA. For the vegetation proteome, we input 3694 proteins and 35 clinical traits of the 158 IE patients into WGCNA. The co-expression network analysis was performed in R using the WGCNA package (v1.72) as previously described[71,72]. The following standard parameters were used: a soft threshold at a power of 3 (plasma proteome) or 5 (vegetation proteome), an unsigned network, and a minimum module size of 30. Following module detection, pathway enrichment analysis was employed to elucidate the biological relevance of proteins clustered within each module. In addition, each module can be further filtered to identify the top hubs, namely, hub proteins, relative to the desired criteria using measures, such as intramodular connectivity (kME) and gene significance (GS). The correlation between genes and their corresponding modules was represented by the module membership (MM). Proteins with high within-module connectivity (MM > 0.70) and a strong correlation with the corresponding clinical traits (GS > 0.25) were considered hub proteins.

**Pathway enrichment analysis.** Pathway enrichment analysis was performed using the online tools DAVID and ConsensusPathDB (http://cpdb.molgen.mpg.de/). Significantly enriched pathways were identified from the GO, KEGG, and Reactome databases, with a false discovery rate (FDR) < 0.05 for DAVID and a $q$-value < 0.05 for ConsensusPathDB considered statistically significant.

**Histologic origin enrichment analysis.** Histologic origin enrichment analysis was performed using the online tool Metascape (https://metascape.org/)[73]. Tissue-specific gene signatures were derived from PaGenBase, which provides comprehensive gene expression profiles across diverse histologic origins.

**Protein-protein interaction (PPI) network analysis.** PPI analysis was performed using the online tools STRING database (https://string-db.org/) and Metascape (https://metascape.org/)[73]. For STRING analysis, the interaction confidence was set according to the default parameters

at a score ≥ 0.4 (medium confidence), and disconnected nodes were excluded. In the Metascape analysis, molecular complexes were identified using the molecular complex detection (MCODE) algorithm with default parameters to detect densely connected network components.

**High-yield and low-yield scores.** High-yield and low-yield scores were inferred using single-sample gene set enrichment analysis (ssGSEA)[25] via the GSVA package[74] with the plasma proteomic matrix and a plasma protein set as background. The plasma protein set was generated from the two protein panels with gradually increasing (used for high-yield score) or decreasing (used for low-yield score) abundance from the double-positive category to single-positive category, double-negative category, and non-IE category by comparing the average protein abundance of each group. In addition, the protein panel with a high yield score exhibited a significantly greater average abundance in the double-positive category than in the other three categories (FC > 1.5, FDR < 0.05), and the protein panel with a low yield score exhibited a significantly greater average abundance in the non-IE category than in the other three categories (FC > 1.5, FDR < 0.05).

**Enzyme-linked immunosorbent assay (ELISA)**
The concentration of the plasma protein LRG1 was assessed by a 96-well Human LRG1 ELISA kit (Elabscience, Wuhan, China) according to the manufacturer's instructions. The sample dilution factor was adjusted to obtain the suggested range for the assay. The samples were added to the wells of an anti-LRG1 microplate and incubated at 37 °C for 2 h. After washing, the detector antibody was added to each well and incubated at 37 °C for 1 h. After washing, a substrate specific to the enzyme conjugated to the detection antibody was added to each well for subsequent color rendering and termination. After washing, the absorbance of the samples at 450 nm was measured immediately using a microplate reader (Bio-Rad Laboratory, Hercules, CA, USA).

**Immunofluorescence staining**
For immunofluorescence, more than 5 μm thick frozen sections of vegetation samples were fixed in 4% paraformaldehyde for 24 h followed by 8 h in 30% sucrose in PBS. To detect NET formation, the sections were incubated with anti-CitH3 (1:500, ab281584, Abcam) and anti-MPO (1:3000, ab208670, Abcam) antibodies. 4′,6-Diami-dino-2-phenylindole (DAPI, Servicebio, Wuhan, China) was used to detect DNA. Finally, the sections were visualized using an Olympus microscope (IX73, Tokyo, Japan), and image processing and analysis were performed with SlideViewer (v2.5) software.

**Statistical analysis**
Statistical analyses were performed in the R (v4.1.2) and Python (v3.9.7) environments. Pearson's $\chi^2$ test was used to analyze the association between categorical variables. The unpaired Wilcoxon rank-sum test and Student's $t$ test were used to compare two groups, and the Kruskal-Wallis test was employed for multiple groups in the statistical analysis of continuous variables. Pearson's correlation and Spearman's correlation analyses were utilized for correlation analysis. Categorical variables were numerically encoded for correlation analysis; for example, blood culture results were coded as 0 (negative) and 1 (positive), and mitral valve surgery status was represented as 0 (no surgery), 1 (repair), and 2 (replacement). All the statistical tests were two-sided, and a $p$-value < 0.05 was considered to indicate statistical significance. The p-values were adjusted (false discovery rate, FDR) using the Benjamini–Hochberg method when applicable, and an FDR < 0.05 was considered to indicate statistical significance. Statistical details are provided in the figure legends and Source Data files.

**Reporting summary**
Further information on research design is available in the Nature Portfolio Reporting Summary linked to this article.

## Data availability
The raw mass spectrometry proteomic data generated in this study have been deposited to the ProteomeXchange Consortium (https://proteomecentral.proteomexchange.org) via the iProX partner repository[75] with the dataset identifier PXD062668. All software tools and publicly available resources used in this study are detailed in the Methods section. Additional processed data supporting the findings are available in the Supplementary Information or Source Data files. Source data are provided with this paper.

## Code availability
No custom code or mathematical algorithms were used in this study. All analyses were conducted using established methods and standard software ("Methods"). The code used to generate figures is available from the corresponding authors upon request.

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

## Acknowledgements

This research is supported by the Noncommunicable Chronic Diseases-National Science and Technology Major Project (Grant 2024ZD0532700 [B.G.]), the Key-Area Research and Development Program of Guangdong Province (2023B0101200014 [B.G.]), Research foundation for advanced talents of Guangdong Provincial People's Hospital (KJ012021097 [B.G.]), National Key Research and Development Program of China (2022YFA1303200 [C.D.], and 2022YFA1303201 [C.D.]), National Natural Science Foundation of China (32330062 [C.D.], and 31972933 [C.D.]), sponsored by Program of Shanghai Academic/Technology Research Leader (22XD1420100 [C.D.]), the Major Project of Special Development Funds of Zhangjiang National Independent Innovation Demonstration Zone (ZJ2019-ZD-004 [C.D.]), Shanghai Municipal Science and Technology Major Project (2023SHZDZX02 [C.D.]), the Fudan Original Research Personalized Support Project [C.D.]. This work is supported by the Shanghai Municipal Science and Technology Major Project, the Human Phenome Data Center of Fudan University, and the Shanghai Phenomic precision measurement professional technical service platform (23DZ2290800).

## Author contributions

C.D., X.J., and B.G.: conceptualization and methodology. S.H., C.D., X.J., and B.G.: project administration. C.D. and B.G.: funding acquisition. W.W., D.Z., and X.H.: resources (clinical data and follow-up information collection). J.M., L.G., H.Q., F.C., O.C., and W.W.: resources (sample collection). J.Z., C.M., and S.H.: investigation (sample handling). C.M., J.Z., and S.H.: investigation (proteomic data generation). C.M., P.R., J.Z., and S.H.: formal analysis (bioinformatics and statistical analyses). S.Tan and Y.W.: mass spectrometry data generation and instrument maintenance. Z.Q. and J.F.: quantification of proteome data. X.H. and S.H.: investigation (IF staining and analyses). S.H. and W.L.: investigation (ELISA and analyses). S.Tian and C.D.: resources (experimental materials). X.H. and S.H.: writing – original draft preparation. All authors: writing – review and editing.

## Competing interests

The authors declare no competing interests.
