## [Transparent Peer Review file · Nature Communications]

Integrated Plasma and Vegetation Proteomic Characterization of Infective Endocarditis for Early Diagnosis and Treatment

Corresponding Author: Professor Bing Gu

Version 0:

Reviewer comments:

Reviewer #1

(Remarks to the Author)

My review focuses on the clinical aspects of this work.

Abstract: What is an independent plasma cohort? Please define more appropriately the control group.

There are many study endpoints: 1) discriminate between IE and non-IE; 2) discriminate disease severity/prognosis; 3) determine pathogen-specific host response patterns; this makes the results a bit confusing and I am not sure improves the message conveyed.

Line 87: RHD is an immune sequela of Strep. pyogenes infection, and is not 'mainly caused by Streptococcus'. Maybe authors wanted to underscore that IE in developing countries is still mostly observed in subjects with prior RHD AND is caused by streptococci other than S. pyogenes, mostly viridans group Strep and group D strep.

Lines 91-97: authors reasoning, although largely based on real issues, appears confusing and requires better explanation. It's true that the biological alterations caused by IE causative pathogens is not very well studied, but such alterations mostly affect – at least in the early phases of the disease, when multiorgan failure has not yet developed – the endocardial surface, heart valves and possibly arterial vessels. Authors in contrast introduce the issue of renal and liver failure, that can be observed in IE but are not always present or occur late in the disease course. Thus, deepening our awareness of organ failure in IE mostly requires investigation of heart failure pathogenesis and the mechanisms underlying embolic complications, that only infrequently affect kidneys or the liver.

To study proteomics of IE vegetations, in addition to plasma, authors understandably enrolled IE patients who underwent cardiac surgery. This entails a well defined selection bias that should be taken into account when assessing data. In particular, findings may only apply to 'surgical IE patients', and not the majority of IE patients who are cared for medically only or who are not undergoing surgery due to high risk, contraindication or death before surgery can be performed. This should be regarded as a limitation of the study.

Study inclusion criteria: suspected IE is not a defined diagnostic category based on prior or current (2023 ESC or Duke-ISCVID) definitions. If a case is confirmed by histopathology, then it is a pathologically defined definite IE. Authors might reclassify 'clinically-defined definite IE' OR definite IE based on clinical criteria and definite IE based on histopathology criteria. I suppose histopathology was done when surgeon intra-operatively detected signs compatible with IE: if that is the case, this should be stated.

Inclusion criteria apparently restricted/influenced patient enrollment. Subjects were included IF culture and mNGS of blood and vegetation samples for pathogen identification had been performed. How many of the screened subjects (initial cohort 4312 heart valve disease subjects) who fulfilled clinical criteria for IE actually underwent culture and mNGS of blood and vegetation samples?

There is no clear mention of the type of IE patients were affected by, i.e. native valve infection, prosthetic valve infection, both. Were there any implantable cardiac device infection cases? If so, which samples were actually taken?

Patients with severe infection were excluded. How was 'severe infection' defined?

Which non-IE diseases were present in the 100 controls of the discovery set?

There is inconsistency in number of study patients:

Line 613: two cohorts: Cohort 1 (discovery set) with 238 IE patients and 100 non-IE individuals

Lines 622-624: In Cohort 1, 202 preoperative and 23 postoperative plasma samples (n=225) were collected from 202 IE patients and 100 non-IE plasma samples for plasma proteomics analysis, and 158 patient vegetation samples were collected for vegetation proteomic analysis.

Please explain and/or modify.

It is unclear to me whether proteomic analysis was done on 'vegetation samples' (line 115) or on 'cryopulverized IE valve tissue' (line 633). There may be significant proteomic differences between vegetation samples, that are made of thrombotic material derived from coagulation system activation and bacteria, and valve tissue that is cardiac in origin and made of completely different cells and matrix.

Timing of plasma sample retrieval should be more clearly stated. There may be significant differences between proteomic asset at IE onset/early phase after diagnosis and in the post-operative phase. I know very well how difficult the timely retrieval of samples in a diverse population of subjects, such as IE patients, may be, but a better standardization would be desirable. Indeed, authors did not provide any sample size calculation, thereby implying that possibly more than needed patients were studied.

NYHA functional classification is not the best way to classify IE patients. This classification focuses on the limitations of patients in daily activities, which is not the case for hospitalized subjects with IE undergoing heart valve surgery.

The actual microbial etiology of included IE patients deserves consideration: there is a very high preponderance of Streptococci (58.97%), and a very high rate of *C. burnetii* (10.26%), with in contrast very few cases due to *S. aureus* (8.33%) or CoNS Staph (6.41%). This is another limitation of the study when trying to apply findings to the broader population of IE patients where predominant pathogens are *S. aureus*, CoNS and enterococci, in addition to streptococci.

Reviewer #2

(Remarks to the Author)

The authors presented a solid machine-learning pipeline. The trained models performed well, demonstrating the applicability of plasma proteomics to differentiate preoperative IE and non-IE patients, as well as NYHA class IV from NYHA class II and III patients. The use of a second cohort to validate the models strengthens their findings.

Although the overall work results and methods are solid, some points about the machine learning pipeline could be clarified to understand the findings better. Therefore, I list in the sequence some questions that arose during the review:

1) (lines 758- 774) The authors state that the data was split randomly into train and test. Since the groups are sometimes unbalanced, the usual approach would be to use a stratified randomized split.

The authors discuss the feature selection method employed during model training. Among the feature selection methods employed, the authors used statistical tests. In this part, were the statistical tests performed only in the training dataset or the total data? This could be considered data leakage if they were performed in the total data.

2) (lines 778-780) The authors discuss the model evaluation and hyperparameter tuning. In this part, it would be good to have some extra information. How was the hyperparameter tuning performed? Was it a cross-validation, and were the best-scoring fold hyperparameters selected? Or was the model run several times, and the mean performance was used to select the best set of hyperparameters? Was the model evaluated after hyperparameter tuning? The phrasing places model evaluation before hyperparameter tuning and it might generate some confusion.

3) (lines 784-787) In this passage, the authors talk about the validation in a second cohort. The authors say that they repeated the steps of model evaluation and hyperparameter tuning. Was the model trained with Cohort 1 used to predict Cohort 2, or was there a model training using only the Cohort 2 data? Also, how was the hyperparameter tuning and model

evaluation in cohort 2 performed? Was cohort 1 used as a training dataset to select the best set of hyperparameters again and then evaluated on cohort 2?

4) (lines 280-283) The authors talk about the model results for the NYHA class prediction. In Figure 4 the legend might be switched, as the authors wrote in the legend:

“ML model establishment based on plasma proteomics (B-D) and CBMs (E-G).The top 10 CBMs (B) or the top 20 DEPs (E) for feature importance in distinguishing NYHA class IV from NYHA class II and III patients”

But in the figure (B-D) show results for CBMs and (E-G) show results for plasma DEPs. The main text matches the figure. Please verify if that is correct.

5) In Figure 4A there is a schematic of the machine learning pipeline. There are two datasets represented there, one with 194 samples and another with 199 samples. Why are there two different datasets and why are they used differently? Accordingly to the image, one is used to search for optimal CBMs and the other is used to search for optimal DEPs, please provide more information about this schematic and about the datasets used in this step. Also, in the split section of the image the authors says that the datasets were split into 70% train and 30% validation and further on the image there is a CV with $k=6$, in this case, the 30 % validation dataset is actually a hold-out test set? Since $k=6$ for the cross-validation, the validation dataset should be approx. 16.67%.

6) In Figures 1H and 4I, the authors present ROC curves for the test set. It is unclear to me why this plot is multiple fold. If the model was trained using cross-validation, the hold-out test set should be tested on the final model with the best set of hyperparameters thus you shouldn't have multiple test ROC curves. Is that ROC curve related to the test set or the validation set (from the cross-validation using the training set)? If there is a reason for the multiple test ROC curves, please provide more detail about how the models were trained and evaluated in the methods section.

In summary, the model performances are good, and the methods were well thought. I believe that clarifying the mentioned points would be beneficial for the overall understanding of the machine learning framework used here and could increase the reliability of the interesting findings they reported in this work.

Reviewer #4

(Remarks to the Author)

I co-reviewed this manuscript with one of the reviewers who provided the listed reports. This is part of the Nature Communications initiative to facilitate training in peer review and to provide appropriate recognition for Early Career Researchers who co-review manuscripts

Version 1:

Reviewer comments:

Reviewer #1

(Remarks to the Author)

I have carefully read the response authors provided to my comments.

I believe that authors have carefully addressed all issues raised in the revised version of their article. I would have no further comments.

Reviewer #5

(Remarks to the Author)

Thanks for the revision. I am fine with the reply to Reviewer #2.

**Response to Reviewers Point-by-Point**

**REVIEWER COMMENTS**

**Reviewer #1 (Remarks to the Author):**

**My review focuses on the clinical aspects of this work.**

**Response:**

We sincerely appreciate your thoughtful and valuable feedback. In response to the reviewer's comments,
we have systematically revised the manuscript, significantly improving its clarity, rigor, and overall
quality. This included providing more explicit and detailed information, incorporating supplementary
analyses, expanding the discussion of limitations, enhancing the overall logical coherence and
significance, and adding pertinent clarifications to address each question raised by the reviewer. **All**
**changes in the revised manuscript have been marked in red for clarity.** The detailed point-by-point
responses were provided as follows.

**Q1. Abstract: What is an independent plasma cohort? Please define more appropriately the control**
**group.**

**Response to Q1:**

We sincerely apologize for the unclear definitions of the external validation cohort and control group in
our original manuscript. **In the revised manuscript**, according to the reviewer's insightful comments,
we have provided a clearer and more straightforward schematic of the study design (**Figure RL1**).
Additionally, we have updated the relevant descriptions for the external validation cohort and control
group to provide clearer definitions. The detailed response was provided as follows.

**(1) As for the Definition of the External Validation Cohort in Our Study:**

In prediction model studies, it is strongly recommended to evaluate the performance of the model on
other unseen participant data (rather than data used for model development) after developing the
predictive model, namely external validation data. **External validation** mainly includes two forms: 1)
temporal or narrow validation: using participant data collected by the same investigators, but sampled
from a later period; 2) geographic or broad validation: using participant data obtained by other
investigators in another hospital or country (*BMJ*, 2015, PMID: 25569120). This research strategy is
widely applied in studies focused on biomarker identification using plasma proteomic data, to validate
the predictive power and robustness of the identified disease-related biomarkers (*Nat Med*, 2022, PMID:
**35654907**; *Lancet Neurol*, 2019, PMID: 30981640).

In our study, we collected a discovery cohort, including plasma and vegetation samples from 238 IE
patients and 100 controls from Guangdong Provincial People's Hospital, referred to as Cohort 1. We also
collected **an external validation cohort, consisting of plasma samples from 92 IE patients and 92**
**controls, named Cohort 2**, which was collected from Guangdong Provincial People's Hospital but is
independent of the Cohort 1 for validation. **In this revision, we have newly incorporated another**
**external validation cohort**, which embraces plasma samples from 72 IE patients and 72 controls from

an external center, Heyuan People’s Hospital, referred to as Cohort 3, to further evaluate the
reproducibility and generalizability of the models in the original manuscript (Figure RL1).

Above all, the “an independent plasma cohort” in our original abstract actually referred to an external
validation cohort (Cohort 2) in our study. According to the reviewer’s constructive comments, we have
revised the description of “an independent plasma cohort” to “an external validation cohort with
plasma samples”. For more details, please refer to the “Abstract” section of the revised manuscript
(lines 35-36 on page 2).

**Figure RL1.** Flowchart illustrating the patient inclusion and exclusion process.

**(2) As for the Definition of the Control Group in Our Study:**

Reaching a rapid and accurate diagnosis in cases of suspected IE is a central challenge of the disease.
Delayed diagnosis and initiation of therapy lead to complications and worse clinical outcomes (*JACC*,
**2017, PMID: 28104075**). The 2023 ESC guidelines (*EHJ*, **2023, PMID: 37622656**) highlight that IE
poses a diagnostic challenge due to its variable clinical presentation. Initial clinical assessment of a
patient with suspected IE involves evaluation of risk factors and a search for a supportive history and
examination findings. Core risk factors include previous IE, a prosthetic valve or cardiac device, and
valvular or congenital heart disease (*Lancet*, **2016, PMID: 26341945**). Consequently, diagnosis should
be considered for any patient with infection symptoms, especially those with valvular damage.

There are limited numbers of well-designed randomized controlled trials studying IE. Based on earlier
inter-group comparative studies related to IE (*JACC*, 2021, PMID: 33795037; *Int J Infect Dis*, 2020,
PMID: 32087365), we designed the study cohorts to identify potential diagnostic biomarkers for IE,
aiming to facilitate early and accurate diagnosis (**Figure RL1**). Initially, we prospectively collected a
**preliminary cohort of 4,312 patients with heart valve disease** from Guangdong Provincial People’s
Hospital. **Among them, a total of 1,564 patients initially underwent blood culture due to clinical**
**suspicion of IE. Both IE patients and non-IE individuals (control group)** included in the discovery
cohort (Cohort 1) and the external validation cohort (Cohort 2) of this study **were selected from the**
**above initial cohort. All IE patients and non-IE individuals included in our study were diagnosed**
**according to the 2023 Duke-ISCVID Criteria** (*Clin Infect Dis*, 2023, PMID: 37138445).

**Specifically, the control group consisted of non-IE individuals included in this study who exhibited**
**clinical features suggestive of IE (e.g., fever or heart murmur) or echocardiographic evidence of**
**valvular lesions but were ruled out for IE according to the 2023 Duke-ISCVID Criteria.** To ensure
a homogeneous study population and minimize confounding factors, **non-IE individuals were required**
**to meet the following inclusion and exclusion criteria:**

• Inclusion criteria: 1) patients ruled out for IE; 2) patients with surgical treatment; 3) patients with
informed consent; 4) age \geq 18 years.

• Exclusion criteria: 1) patients unable to tolerate surgery due to severe sepsis or septic shock; 2) severe
end-stage liver disease (Child-Pugh class C); 3) patients with immunosuppressive or immunomodulatory
therapy; 4) known pregnancy.

For the newly collected external validation cohort from Heyuan People’s Hospital (Cohort 3) in the
revision, the inclusion and exclusion criteria for IE patients and non-IE individuals were consistent with
those of the Cohort 1 and Cohort 2.

**In the revision**, we have provided a clearer and more detailed description of patient recruitment in the
**“Methods”** section of the revised manuscript, specifically under the subtitles: “Study design and
specimen acquisition” (**lines 724-755 on pages 25-26**). We have also clarified the definition of the non-
IE individuals (control group) in the **“Results”** section of the revised manuscript (**lines 122-124 on page**
**5**). Additionally, the newly added straightforward schematic of the study design has been incorporated
into the **Figure S1** in the updated **Supplementary Figures** file.

**Q2. There are many study endpoints: 1) discriminate between IE and non-IE; 2) discriminate**
**disease severity/prognosis; 3) determine pathogen-specific host response patterns; this makes the**
**results a bit confusing and I am not sure improves the message conveyed.**

**Response to Q2:**

We apologize for any lack of clarity in the organization and presentation of our research objectives and
findings in the original manuscript. We fully agree with the reviewer’s insightful comments that
including various study endpoints may complicate results, obscure critical information, and diminish the
clarity of our conclusions. **To enhance the clarity, logic, and readability, we have comprehensively**

**revised the manuscript to** elaborate on the logical progression of our findings; highlight key findings
of each study endpoint and their translational prospects in clinical practice; and articulate the connections
between the main findings from various research endpoints. Our detailed responses are presented below:

**(1) About the Aims and Methodological Framework of Our Study:**

The complexity and inaccuracy of pathogen identification and echocardiographic technologies pose
challenges to achieving an early and accurate diagnosis of IE. Moreover, antibiotic resistance and
surgical risks further exacerbate the already limited treatment options in IE. Therefore, there is a pressing
need for simple and effective diagnostic strategies to assist in patient screening, especially for those at
high risk, as well as for novel therapies to help optimize treatment options. **To this end, our study aims**
**to identify early diagnostic and prognostic biomarkers and to elucidate the underlying pathological**
**mechanisms of disease onset and progression, with the goal of providing valuable insights to**
**enhance the clinical management of IE.**

The mass spectrometry (MS)-based proteomic research strategy enables specificity in the identification
and quantification of thousands of proteins present in clinical samples and is widely used to study disease
mechanisms and identify biomarkers for non-invasive diagnostic applications (*Nat Med*, 2022, PMID:
35654907). **In the original manuscript**, we collected a discovery cohort (Cohort 1) with 238 IE patients
and 100 non-IE individuals, and conducted in-depth proteomic analyses on three types of clinical samples,
plasma samples collected at admission (IE: n = 202, non-IE: n = 100), plasma samples collected within
three to seven days postoperatively (IE: n = 23), and vegetation samples (IE: n = 158). Furthermore, we
recruited an external validation cohort (Cohort 2), consisting of plasma samples collected at admission
from 92 IE patients and 92 non-IE individuals, to confirm the findings. **In the revision**, according to the
reviewer's constructive suggestions, we have newly incorporated another external validation cohort
(Cohort 3), with plasma samples collected at admission from 72 IE patients and 72 non-IE individuals.
The Cohort 3 was recruited from an external center, distinct from the Cohort 1 and Cohort 2, to further
evaluate the reproducibility and generalizability of the model derived from the discovery cohort (**Figure**
**RL2A**).

Overall, in response to the reviewer's thoughtful feedback, **we have refined the introduction in the**
**revised manuscript to enhance clarity and explicitly outline the research objectives and strategies.**
For more details, please refer to the "**Introduction**" section of the revised manuscript (**lines 57-113 on**
**pages 2-4**).

**Figure RL2A.** The workflow depicting the cohort populations and research strategy.

**(2) About the Key Findings and Their Connections in Our Study:**

We sincerely apologize for any confusion caused by the unclear description of the rationale and logical
 progression in investigating the proteomic alterations associated with various endpoints in the original
 manuscript. **In the revision**, we have enhanced the clarity of the descriptions regarding the key findings,
 their clinical significance, and the connections among these findings, which were outlined as follows.

**i) Regarding Discriminating Between IE and Non-IE:**

**In the original manuscript**, we developed a machine learning-based diagnostic prediction model to
 identify effective biomarkers for the early diagnosis of IE. This model distinguished between IE patients
 and non-IE individuals, utilizing proteomic data from plasma samples that were prospectively collected
 at admission in the discovery cohort (Cohort 1). Moreover, the robustness and reliability of this model
 were validated in the external validation cohort (Cohort 2). Finally, we established a model that included
 16 proteins, such as CRP, leucine rich alpha-2-glycoprotein 1 (LRG1), and apolipoprotein A2 (APOA2),
 demonstrating outstanding performance and reliability, with overall AUC values of 0.98 in the Cohort 1
 and 0.99 in the Cohort 2 (**original Figure 1 and Figure S1**).

**In this revision**, in response to the reviewer's insightful feedback, we have optimized the machine
 learning pipeline to further enhance the accuracy and reliability of the model, as well as newly
 incorporated an external validation cohort recruited from an external center (Cohort 3) to further evaluate
 the generalization of the model. As a result, the updated diagnostic prediction model to distinguish IE
 from non-IE individuals contained 10 proteins (**Figure RL2B-F**), including 7 that were upregulated in
 IE (**Figure RL2D**), such as CRP, LRG1, and actin beta (ACTB), and 3 that were downregulated,
 including apolipoprotein C1 (APOC1), APOA2, and toll interacting protein (TOLLIP). This updated
 model achieved an overall AUC of 0.97 (Cohort 1), which was further validated in the two external
 validation cohorts with overall AUC values of 0.98 (Cohort 2) and 0.94 (Cohort 3), respectively (**Figure**
 **RL2E**). **Compared to the results in the original manuscript**, these demonstrated that the updated
 model maintained its initial effectiveness while reducing complexity and redundancy by narrowing the

feature set from 16 to 10. Among the 10 features, 8 features were selected from the original set, while
 actin gamma 1 (ACTG1) and superoxide dismutase 3 (SOD3) were newly identified. The combination
 of the 10 features preserved the original discriminatory power of the 16-feature combinations, thereby
 enhancing the model's efficiency and effectiveness in practice. In addition, the reproducibility and
 generalizability of the updated model were further validated in the Cohort 3. **In summary, these findings**
 **highlight the outstanding discriminatory performance of the diagnostic prediction model for IE,**
 **suggesting that the 10 proteins identified in this model are promising early diagnostic biomarkers**
 **that could aid in clinical decision-making.**

**Figure RL2B-F.** (B) Bar plot illustrating the weighted feature importance of the top 20 plasma proteins
 identified by the ensemble model to distinguish between IE and non-IE samples. The error bars represent
 the standard error of the feature importance estimates. (C) Scatter line plot illustrating the optimal feature
 combinations for the model that yield the highest accuracy, determined through recursive feature
 elimination cross-validation (RFECV). (D) Heatmap showing the expression profiles of the 10 plasma
 proteins identified in the developed model. Each column represents a patient sample and rows indicate
 proteins. The color range in the heatmap represents the row z-score of the normalized protein expression
 values ranging from -1.5 (blue) to $+1.5$ (red). The annotation of pathogen identified by metagenomic
 next-generation sequencing (mNGS) for each sample is displayed above the heatmap, with blanks
 indicating missing records. (E, F) Receiver Operating Characteristic (ROC) curves (E) and confusion
 matrices (F) illustrating the model performance on the hold-out test set of the discovery cohort (Cohort
 1) and two external validation cohorts: Cohort 2 and Cohort 3. The area under the ROC curve (AUC)
 provides a measure of the model's discriminative ability, with values closer to 1 indicating better
 performance. The matrix shows the number of true positives, true negatives, false positives, and false
 negatives, providing insight into the model's accuracy and classification errors.

Notably, the protein LRG1, identified as a component of the diagnostic prediction model mentioned
 above (**Figure RL2D**), was found to be linked with the severity of infection in both plasma and
 vegetation proteomic data in the subsequent analysis of the original manuscript (**Figure RL2G**).
 Specifically, to investigate functional alterations related to the severity of infection, we first classified IE
 patients into three categories with gradually increasing yields of detectable pathogen, namely, double-
 positive, single-positive, and double-negative, based on the results of blood culture and metagenomic
 next-generation sequencing (mNGS). This classification related to the yield of detectable pathogen was
 subsequently shown to reflect the severity of infection (defined by elevated levels of clinical infection
 markers) (**Figure 5**). Moreover, we validated that the level of LRG1 in IE patients was significantly
 higher than that in non-IE individuals (**Figure RL2H**) using the enzyme-linked immunosorbent assay
 (ELISA). **In the revision**, in addition to the original external validation cohort (Cohort 2), we validated
 a consistent result showing a significant upregulation of the plasma protein LRG1 in IE patients
 compared to non-IE individuals in the newly incorporated external validation cohort (Cohort 3) (**Figure**
 **RL2I**). **Consequently, through the findings related to the two study endpoints—discriminating**
 **between IE and non-IE individuals, as well as differentiating IE patients with varying degrees of**
 **infection—we proposed LRG1 as a potential crucial biomarker for IE, further demonstrating its**
 **specificity and reliability.**

**Figure RL2G-I.** (G) Boxplots depicting the LRG1 abundance at the plasma and vegetation proteomic
 levels across patients in the double-positive, single-positive, double-negative, and non-IE groups.
 Differences among groups were assessed using the Kruskal-Wallis test and Wilcoxon rank-sum test. (H)
 Boxplot showing significant differences in the LRG1 concentration between IE and non-IE samples
 measured by ELISA. The Wilcoxon rank-sum test was used. (I) Boxplots illustrating the LRG1
 abundance at the plasma proteomic levels across IE and non-IE samples.

**ii) Regarding Discriminating Disease Severity/Prognosis:**

**Subsequent to the understanding of the overall proteomic characteristics that distinguish IE from**
 **the controls, we investigated the proteomic characteristics associated with various clinical features**
 **among IE patients by analyzing matched vegetation and plasma proteomes.** By weighted correlation
 network analysis (WGCNA), we identified multiple modules of highly correlated proteins that showed
 significant associations with clinical traits, including ICU stays, NYHA functional classification, and
 positive blood culture. For instance, protein co-expression networks characterized by glycometabolism,
 amino acid metabolism, and adhesion were found to be linked to multiple adverse events in IE (**Figure**
 **2**). **The comprehensive WGCNA results provided valuable insights into the biological functions**
 **and interconnections associated with various clinical traits in IE, which are crucial for elucidating**
 **the underlying disease pathogenesis and improving treatment strategies.**

Exploring the underlying biomarkers and pathological mechanisms related to disease severity and prognosis is crucial for effective risk prediction and stratified therapy in IE, where severity and prognosis are linked to valvular damage, infection severity, and complications. Notably, in our study, the above WGCNA results showed that the MEturquoise module of the plasma proteome exhibited significant negative correlations with several clinical traits related to disease severity/prognosis, like severe infection, abscess, and NYHA, **with the most pronounced correlation observed with NYHA (Figure RL2J).** Heart failure complicating IE is recognized as an independent factor associated with poor outcomes and is the primary indication for surgery for IE. NYHA functional classification, a routinely used system for assessing the severity of heart failure symptoms in clinical practice, has been identified as an important prognostic indicator in IE (*J Am Heart Assoc*, 2016, PMID: 27091179). **We further observed significant differences between NYHA class IV patients and those in NYHA class II/III in terms of both clinical and proteomic characteristics. Specifically,** factors such as advanced age, valve damage, liver dysfunction (defined by elevated levels of AST, DBIL, etc.), high postoperative adverse events (e.g., reoperation, LCOS, long-time ICU stay), and high mortality (e.g., D30) were notably enriched in NYHA class IV patients, compared to those in NYHA class II and III (**Figure RL2K**). In the MEturquoise module, we screened 21 plasma proteins that exhibited a significant negative correlation with the NYHA functional classification, with expression profiles showing notable differences between NYHA class IV patients and those in classes II and III (**Figure RL2K**). Further analysis suggested that the decrease of these proteins in IE patients with severe heart failure might be due to synthesis inhibition resulting from liver dysfunction (defined by elevated levels of AST, DBIL, etc.) (**Figure RL2L, M**). Consequently, we then conducted a comprehensive differential analysis between NYHA class IV patients and NYHA class II and III patients to identify underlying mechanisms exacerbating the deterioration of IE with severe heart failure. We detected a decrease in the levels of many liver-derived plasma proteins, including complement (e.g. C3 and C4), HDL (e.g. APOA1 and APOA2), and serine protease inhibitors (e.g. SERPINC1), in NYHA class IV patients, while an increase in pathways comprising neutrophil degranulation, cell adhesion, and proteolysis (**Figure 3**). These biological alterations could lead to increased heightened vulnerability to infections (*J Autoimmun*, 2023, PMID: 36535812) and tissue destruction (*NEJM*, 1989, PMID: 2536474) for IE patients. Taken together, **these findings suggest the need for risk stratification of NYHA class IV patients in IE, who often face higher postoperative risk.**

Figure RL2J-M. (J) Heatmap showing the correlation between modules (co-expressed protein network) and clinical traits. Each row corresponds to a module. Each column corresponds to a clinical trait. Color-coded cells represent correlation strengths according to the provided legend, with respective correlation coefficients and p values noted. (K) Heatmap exhibiting the expression profiles of 21 plasma proteins across NYHA classes II, III, and IV. Each column represents a patient sample and rows indicate proteins. The color range in the heatmap represents the row z-score of the normalized protein expression values ranging from -2 (blue) to +2 (red). The clinical annotation of each sample is shown above the heatmap. The chi-square test was used to evaluate the association of NYHA class with the clinical features. Statistical significance is indicated as follows: *P < 0.05, **P < 0.01, ***P < 0.001. (L) Bar chart showing the histologic origin enrichment of the 21 proteins using Metascape (<https://metascape.org/>). (M) Boxplots delineating the significant differences of 6 routine blood indexes that reflect liver function between NYHA class IV patients and those in NYHA class II and III.

Nearly half of IE patients require surgical intervention, and this proportion has been gradually increasing in recent years. However, according to the 2023 ESC IE Guidelines, **preoperative risk assessment continues to be a challenge for IE in clinical decision-making** due to the lack of preoperative risk stratification models with robust and efficient performance. **Given the finding that IE patients with NYHA class IV had significantly worse postoperative outcomes compared to those in NYHA class II and III, we subsequently developed three prognostic models for IE** utilizing the blood-derived characteristics of the clinical blood markers (CBMs), measured by clinical blood tests, and plasma proteome: CBM-only, proteome-only, and proteome + CBM, to distinguish between the two groups of patients. Both clinical blood tests and plasma proteomic data were obtained from the plasma samples collected at admission. **In the original manuscript**, the proteome + CBM model, containing 4 CBMs (e.g., AST; BUN) and 11 proteins, such as adiponectin (ADIPOQ) and fetuin B (FETUB), yielded the highest overall AUC value of 0.85, which was slightly better than that of the proteomics-only model

(0.83) and significantly better than that of the CBM-only model (0.73) (original Figure 4). These results suggest the promise of plasma protein biomarkers for predicting the risk of IE patients.

In the revision, we optimized the machine learning pipeline to further enhance the accuracy and reliability of the model, as well as newly incorporated an external validation cohort recruited from an external center (Cohort 3) to further evaluate the generalization of the model, in accordance with the reviewer’s suggestions. Compared to the results in the original manuscript, the updated prognostic models showed the following improvements: 1) The updated models exhibited performance metrics that were consistent with, or even better than, those of the original version; 2) The generalizability and robustness of the models were further validated in the newly incorporated external validation cohort (Cohort 3). For instance, in the results for the development of the proteome + CBM model (Figure RL2L-O), we found that the final model contained 15 features of 4 CBMs (e.g., DBIL, BUN) and 11 differentially expressed proteins (DEPs) including ADIPOQ, cystatin C (CST3), and FETUB, with an overall AUC value of 0.873 in the hold-out test set of the discovery cohort (Cohort 1) and that of 0.829 in the external validation dataset (Cohort 3). The confusion matrix results revealed good model performance in distinguishing between NYHA class IV patients and those in NYHA class II and III.

In summary, our in-depth proteomic analysis comparing NYHA class IV patients with those in NYHA class II and III allowed us to elucidate the underlying pathological mechanisms driving the disease severity and adverse outcomes in IE. Additionally, we identified potential prognostic biomarkers. This preoperative risk stratification model might improve the convenience and efficiency of clinical decision-making.

Figure RL2N-Q. (N) Scatter line plot illustrating the optimal feature combinations for the proteome + CBM model that yield the highest accuracy, determined through RFECV. (O) Boxplots and heatmaps illustrating significant differences in the 15 features identified in the proteome + CBM model between NYHA class IV patients and those in NYHA class II and III. The annotation of pathogen identified by mNGS for each sample is displayed above the heatmap, with blanks indicating missing records. (P, Q)

ROC curves (P) and confusion matrices (Q) illustrating the performance of the proteome + CBM model
on the hold-out test set of the discovery cohort (Cohort 1) and the external validation cohort (Cohort 3).

**iii) Regarding Discriminating Pathogen-Specific Host Response Patterns:**

**In addition to valve damage and impaired cardiac function, the progression of IE is closely related**
**to the degree and type of pathogen infections. Consequently,** we further investigated the underlying
pathological mechanisms associated with pathogen infections (**Figure 5 and Figure 6**). As a result, we
illustrated the underlying molecular characteristic associated with severe infection (defined by elevated
levels of clinical infection markers) and the proteomic landscape reflecting pathogen-specific host
response patterns for several classical pathogens of IE, including *Streptococcus*, *C. burnetii*, *S. aureus*,
and CoNS (**Figure RL2R**). For instance, *C. burnetii* infection exhibited a higher incidence of non-fever
presentations, abscesses, aortic valve calcification (AVC), and bicuspid aortic valve (BAV). In the
plasma and vegetation proteomic data, *C. burnetii* infection was characterized by biological processes
related to antibody production, as well as lipid and carbohydrate metabolism, reflecting its long-term
chronic infection characteristics (**Figure 6**).

Notably, *Streptococcus* and *S. aureus*, the two most prevalent causative pathogens in IE patients, were
found to share a common pathway characterized by the formation of neutrophil extracellular traps (NETs)
and to be associated with larger vegetation formation, compared to *C. burnetii* and CoNS infections.
Subsequently, we further demonstrated that NETs were promising therapeutic targets in IE patients with
*Streptococcus* and *S. aureus* infections, utilizing both integrated proteomic analysis and
immunofluorescence quantitative assays (**Figure 6**). Specifically, **IE patients with *Streptococcus* and**
***S. aureus* infections showed a significant increase in signs of NETs**, including proteins of
myeloperoxidase (MPO), citrullination of histone H3 (CitH3), interleukin-8 (IL-8), and neutrophil
cytosolic factor 2 (NCF2), as well as pathways related to actin cytoskeleton dynamics, neutrophil
degranulation, and oxidative stress (**Figure RL2S**). **These findings represented the first identification**
**that NETs promote and expand vegetation formation in human subjects, providing critical**
**evidence that supported earlier observations from mouse models indicating that *Streptococcus* and**
***S. aureus* induce NETs, while CoNS do not** (*Circulation*, 2015, PMID: 25527699; *Arterioscler*
*Thromb Vasc Biol*, 2023, PMID: 36453281). Overall, these findings support that **targeting NETs might**
**be a valuable therapy for inhibiting vegetation formation in patients infected with *Streptococcus***
**and *S. aureus*.**

Figure RL2R, S. (R) Heatmap showing differential plasma proteomic profiles among patients infected with four pathogens, including *Streptococcus*, *C. burnetii*, *S. aureus*, and CoNS. Each column represents a patient sample and rows indicate proteins. The clinical annotation of each sample is shown above the heatmap. The chi-square test was used to evaluate the association of pathogens identified by mNGS with other clinical features. (S) Volcano plot showing DEPs between patients infected with *Streptococcus* & *S. aureus* and patients infected with CoNS & *C. burnetii* in the vegetation proteomic data.

In the revision, we connected the pathogen infection results to our above findings on NYHA risk stratification. The results showed that there was no obvious correlation disclosed between the severity of infection and heart failure (**Figure RL2T**), but a noticeable association between the types of bacterial infections and heart failure severity. Consistent with previous research, we found that *S. aureus* infection, as a well-recognized marker of worse outcomes for IE, was significantly linked to an increased risk compared to other infections (**Figure RL2U**). *S. aureus* infection was characterized by enriched biological functions involving muscle contraction, phagosome, integrin cell surface interactions, hemostasis, and NETosis, which might serve as key mechanisms contributing to severe infection and organizational damage within the host (**Figure 6**). Overall, these findings deepen our understanding of the complex and specific pathological mechanisms associated with pathogen-specific host response patterns in IE, which could help in the development of therapeutic targets.

**Figure RL2T, U.** (T) Stacked bar plots illustrating the proportions of patients across various NYHA
functional classes among the double-positive, single-positive, and double-negative groups. (U) Stacked
390 bar plots showing the proportions of patients across various pathogen infections among the NYHA class
II, II, and III groups (left), as well as the proportions of patients across various NYHA functional classes
among the four pathogen infection groups (right). The p-value was calculated using the chi-square test.

**In summary, while our study intends to explore multiple endpoints, the research questions are**
**interrelated, and the results are comparable and complementary, summarized as:**

**1)** Based on proteomic analysis of plasma samples collected at admission from IE patients and non-IE
individuals, we demonstrated specific biological function alterations associated with IE and developed a
diagnostic prediction model with outstanding performance.

**2)** Subsequent to the understanding of the overall proteomic characteristics of IE, in the integrated
proteomic analysis of the plasma and vegetation samples of IE patients, we characterized the biological
functions and interconnections related to various clinical traits for IE, particularly focusing on
pathological mechanisms associated with NYHA class IV heart failure, which indicated high risk and
adverse outcomes for IE, and developed a prognostic model for IE.

**3)** In addition to valve damage and impaired cardiac function, the progression of IE is closely related to
the degree and type of pathogen infections. Thus, we uncovered the underlying pathological mechanisms
associated with pathogen-specific host response patterns and proposed that NETs might be a promising
therapeutic target for IE patients infected with *Streptococcus* and *S. aureus*.

Each endpoint's results provide essential information for the next, ultimately contributing to a
comprehensive understanding of the complex pathophysiology of IE and offering valuable insights into
its diagnosis and treatment from multiple perspectives.

In response to the reviewer's valuable comments, we have comprehensively updated relevant contents
in the "**Results**" section of the revised manuscript (**lines 117-121 on page 5; lines 167 on page 6; lines**
**200-202 on page 7; lines 253-259 on page 9; lines 315-317 on page 11; and lines 360-511 on pages**
**13-18**) to enhance the clarity and accessibility of our key findings for each study endpoint and their
clinical significance, as well as to ensure a logical progression of our findings. In addition, we have
revised the "**Discussion**" section (**lines 584-600 on pages 20-21; lines 654-697 on pages 22-24**) to
provide a comprehensive explanation of the innovations and significance of the study endpoint results,
emphasizing their collaborative efforts in facilitating early diagnosis and stratified treatment for IE. The
results from updated and supplementary analysis have been incorporated into **Figure 1, Figure 3A,**
**Figure 4, Figure S2, and Figure S6.** Once again, we sincerely thank the reviewer for the constructive
and valuable comments, which have enhanced the organization and preciseness of our results, ensuring
that the key findings and their significance are conveyed more effectively.

**Q3. Line 87: RHD is an immune sequela of Strep. pyogenes infection, and is not 'mainly caused by**
**Streptococcus'. Maybe authors wanted to underscore that IE in developing countries is still mostly**
**observed in subjects with prior RHD AND is caused by streptococci other than S. pyogenes, mostly**
**viridans group Strep and group D strep.**

**Response to Q3:**

We sincerely appreciate your constructive comments and apologize for our inappropriate description of
RHD in the “**Introduction**” section of the original manuscript. **The reviewer’s comments accurately**
**articulate our intended message, and we have revised our relevant statements accordingly. As**
**follows:**

“IE patients resulting from prior rheumatic heart disease remain prevalent in developing countries,
accounting for up to two-thirds of cases. In this context, IE is commonly caused by infections with
viridans group streptococci or group D streptococci.”

As the reviewer noted, RHD is an immune sequela of *Strep. pyogenes* infection, rather than what we
stated: ‘mainly caused by *Streptococcus*’ in the original manuscript. To be precise, acute rheumatic fever
(ARF) is the result of an autoimmune response to pharyngitis caused by infection with the sole member
of the group A Streptococcus (GAS), *Strep. pyogenes*, and ARF can lead to persistent cumulative cardiac
valvular damage, a chronic condition known as RHD (*Nat Rev Dis Primers*, 2016, PMID: 27188830).
Moreover, As the reviewer commented, RHD remains the key risk factor for IE in developing countries
and underlies up to two-thirds of cases, though in developed countries, degenerative valve disease,
intravenous drug use, and congenital heart disease have gradually replaced RHD as the major risk factors
for IE, due to improvements in medical care and living conditions (*Lancet*, 2016, PMID: 26341945). In
countries where RHD is prevalent, IE is commonly caused by viridans group streptococci or group D
streptococci (*Circulation*, 2015, PMID: 26373316).

In the revision, we have updated the corresponding context into the “**Introduction**” section of the revised
manuscript (**lines 89-92 on page 4**). We greatly appreciate the reviewer’s constructive comments,
improving the rigor and quality of our manuscript.

**Q4. Lines 91-97: authors reasoning, although largely based on real issues, appears confusing and**
**requires better explanation. It’s true that the biological alterations caused by IE causative**
**pathogens is not very well studied, but such alterations mostly affect – at least in the early phases**
**of the disease, when multiorgan failure has not yet developed – the endocardial surface, heart**
**valves and possibly arterial vessels. Authors in contrast introduce the issue of renal and liver**
**failure, that can be observed in IE but are not always present or occur late in the disease course.**
**Thus, deepening our awareness of organ failure in IE mostly requires investigation of heart failure**
**pathogenesis and the mechanisms underlying embolic complications, that only infrequently affect**
**kidneys or the liver.**

**Response to Q4:**

We sincerely apologize for our unclear description regarding the rationale for investigating the proteomic
alterations associated with **two risk factors for IE: causative pathogens and heart failure** in the
“**Introduction**” section of the original manuscript. Exploring the underlying biomarkers and pathological
mechanisms related to disease severity and prognosis is crucial for effective risk prediction and stratified
therapy in IE, where severity and prognosis are linked to valvular damage, infection severity, and
complications. **Causative pathogens and heart failure are two key risk factors influencing these**
**outcomes. Actually, as the reviewer’s insightful comments, we separately explored** how proteomic

alterations associated with causative pathogens and heart failure influence the occurrence and
progression of IE in the original manuscript. **In the revision**, we have updated the relevant content in the
introduction section to enhance clarity and readability. The detailed response was provided as follows.

**(1) Rationale and Results of Investigating Biological Alterations Associated with Causative**
**Pathogens:**

As noted by the reviewer, investigating the biological alterations associated with causative pathogens
enhances our understanding of the pathogenic mechanisms by which these pathogens affect damaged or
inflamed endothelium, heart valves, or arterial vessels, ultimately leading to the formation of vegetations,
which serve as the nidus of intracardiac infection in IE patients. While it is known that the spontaneous
deposition of platelet-fibrin aggregates on abnormal valves and injured or inflamed cardiac endothelium
plays a significant role in the pathogenic mechanisms of pathogens in IE (*Nat Rev Cardiol*, 2014, PMID:
**24247105**), **the pathogen-specific host response patterns in IE are not well understood, which limits**
**our insights into the mechanisms through which they contribute to the development of further**
**intracardiac lesions.**

To this end, we **illustrated the proteomic landscape reflecting pathogen-specific host response**
**patterns for four classical pathogens of IE**, including *Streptococcus*, *C. burnetii*, *S. aureus*, and CoNS
(**Figure 6**). For instance, consistent with previous research, we found that *S. aureus* infection, as a well-
recognized marker of worse outcomes for IE, was significantly linked to an increased risk compared to
other infections (**Figure RL2S**). *S. aureus* infections were characterized by enriched biological functions
involving muscle contraction, phagosome, integrin cell surface interactions, hemostasis, and NETosis,
which might serve as key mechanisms contributing to severe infection and organizational damage within
the host. Importantly, our results support that targeting NETs might be a valuable therapy for inhibiting
vegetation formation in patients infected with *Streptococcus* and *S. aureus*. **These findings indeed**
**enhance our insights into the potential pathogenic mechanisms through which causative pathogens**
**contribute to the development of further intracardiac lesions in IE.** For further details of the
clarification of the results, please refer to the **Response to Q2**.

**(2) Rationale and Results of Exploring Biological Changes Associated with Heart Failure:**

We apologize for our inappropriate description regarding the rationale for investigating the biological
changes associated with heart failure in the introduction section of the original manuscript. As the
reviewer suggested, **our findings regarding organ failure in IE indeed primarily focus on heart**
**failure.** Structural valvular damage from vegetation and related lesions often leads to heart failure,
worsening the condition of IE patients. Heart failure complicating IE is independently associated with
poor outcomes and the main indication for surgery for IE. **Yet, there is still a gap in our understanding**
**of the molecular mechanisms through which heart failure exacerbates IE.**

**In our data, we proposed the need for risk stratification of IE patients with NYHA class IV heart**
**failure, who had more significant associations with severe valvular damage, postoperative adverse**
**events, and poor prognosis, compared to those in NYHA class II and III.** **In the revision**, according
to the reviewer's comments, we have further investigated the relationship between heart failure and
embolization in our data and uncovered that **patients with NYHA class III or IV heart failure were**
**indeed more prone to cerebral embolism compared to patients with low-level heart failure in**

NYHA class II, though there was no significant difference between NYHA class IV patients and NYHA
class III patients (Figure RL2K). In the proteomic data, we detected a decrease in the levels of many
liver-derived plasma proteins, including complement (e.g. C3), HDL (e.g. APOA1), and serine protease
inhibitors (e.g. SERPINC1), in NYHA class IV patients, which might be attributed to synthesis inhibition
resulting from liver dysfunction, as evidenced by elevated levels of AST and DBIL in these patients
(Figure RL2K-M), while an increase in pathways comprising neutrophil degranulation, cell adhesion,
and proteolysis (Figure 3). These biological alterations could lead to increased heightened vulnerability
to infections (*J Autoimmun*, 2023, PMID: 36535812) and tissue destruction (*NEJM*, 1989, PMID:
2536474) for IE patients. Overall, our findings improved our understanding of the potential pathogenic
mechanisms underlying the progression of IE in conjunction with heart failure and provided critical
insights for optimizing the risk stratification and treatment of IE patients. For further details of the
clarification of the results, please refer to the Response to Q2.

In response to the reviewer’s insightful comments, we have refined the introduction section of the
revised manuscript to improve the validity and feasibility of the background on investigating the
pathogenic mechanisms associated with the two significant risk factors for IE: causative pathogens and
heart failure. Specifically, as follows:

“Exploring the underlying biomarkers and pathological mechanisms related to disease severity and
prognosis is crucial for effective risk prediction and stratified therapy in IE, where severity and prognosis
are linked to valvular damage, infection severity, and complications. The microbial etiology of IE varies
across regions and socioeconomic contexts. IE patients resulting from prior rheumatic heart disease
remain prevalent in developing countries, accounting for up to two-thirds of cases (*Lancet*, 2016, PMID:
26341945). In this context, IE is commonly caused by infections with viridans group streptococci or
group D streptococci, predominantly affecting the mitral and aortic valves. Other common causative
pathogens of IE include *Staphylococcus aureus* (*S. aureus*), coagulase-negative staphylococci (CoNS),
and *Coxiella burnetii* (*C. burnetii*) (*Circulation*, 2015, PMID: 25527699). Nevertheless, the pathogen-
specific host response patterns for these organisms are not well understood, limiting our insights into the
mechanisms by which they contribute to the development of further intracardiac lesions in IE.
Furthermore, structural valvular damage from vegetation and related lesions often leads to heart failure,
worsening the condition of IE patients. Heart failure complicating IE is independently associated with
poor outcomes and the main indication for surgery for IE (*EHJ*, 2023, PMID: 37622656). Yet, there is
still a gap in our understanding of the molecular mechanisms through which heart failure exacerbates IE.
Deepening our awareness of the molecular mechanism associated with disease progression may provide
new approaches for therapeutic management for IE.”

Please refer to lines 85-103 on pages 3-4 of the “Introduction” section of the revised manuscript for
details. We sincerely thank the reviewer for the valuable insights that helped improve our manuscript to
be more rigorous and understandable.

**Q5. To study proteomics of IE vegetations, in addition to plasma, authors understandably enrolled**
**IE patients who underwent cardiac surgery. This entails a well-defined selection bias that should**
**be taken into account when assessing data. In particular, findings may only apply to ‘surgical IE**
**patients’, and not the majority of IE patients who are cared for medically only or who are not**

**undergoing surgery due to high risk, contraindication or death before surgery can be performed.**
**This should be regarded as a limitation of the study.**

**Response to Q5:**

We greatly appreciate your understanding of this limitation for selection bias in our study. The need to
investigate the vegetation proteome of IE led us to include patients who underwent surgical treatment,
which imposed certain constraints on the evaluation of our results. According to the reviewer's
suggestions, **we have clearly discussed this limitation in our revised manuscript.** The detailed
response was provided as follows.

In this study, all IE patients from the three cohorts were selected from those who had undergone surgical
treatment, with both the timing and indications following the 2023 ESC guidelines. Surgery is reportedly
performed in about half of IE patients. As the reviewer's valuable comments, **the lack of IE patients**
**who are cared for medically only or who are not undergoing surgery** due to high risk,
contraindication or death before surgery in our population cohorts **may impose certain limitations on**
**the generalization of our findings, primarily regarding prognostic outcomes that focus on post-**
**operative prognosis in our study. Accordingly, in our future research,** we will collect plasma samples
from a larger and more diverse group of IE patients, including those from different centers and receiving
various treatment approaches, not limited to those undergoing surgical treatment, to further validate the
generalizability and validity of our biomarkers and findings.

**In the revision,** we have incorporated this limitation into the "**Discussion**" section (**lines 701-704 on**
**page 24**). **Specifically, as follows:**

"To investigate the vegetation proteome in IE, we enrolled only patients undergoing cardiac surgery.
This selection bias inherent in our study design could limit the generalizability of the plasma biomarkers
identified to the broader population, particularly those receiving conservative treatment. This limitation
underscores the need for further research involving a more diverse patient population to enhance the
generalizability of our findings, particularly regarding plasma biomarkers."

Additionally, we have provided explicit statement that all participants in this study received surgical
treatment in the "**Results**" section of the revised manuscript (**lines 133-134 on page 5**). Once again, we
would like to express our gratitude to the reviewer for their comments, which have improved the rigor
and quality of our manuscript.

**Q6. Study inclusion criteria: suspected IE is not a defined diagnostic category based on prior or**
**current (2023 ESC or Duke-ISCVID) definitions. If a case is confirmed by histopathology, then it**
**is a pathologically defined definite IE. Authors might reclassify 'clinically-defined definite IE' OR**
**definite IE based on clinical criteria and definite IE based on histopathology criteria. I suppose**
**histopathology was done when surgeon intra-operatively detected signs compatible with IE: if that**
**is the case, this should be stated.**

**Response to Q6:**

We sincerely appreciate your insightful comments regarding the inclusion criteria, and apologize for our
improper description of the inclusion criteria in the original manuscript. As noted by the reviewer, **in**
**accordance with the 2023 Duke-ISCVID Criteria, all IE patients included in our study were**
**diagnosed with definite IE, encompassing both clinically-defined and pathologically-defined cases.**
The detailed response was provided as follows.

The diagnosis of IE is based on a combination of major and minor criteria rooted in microbiologic,
echocardiographic, and clinical metrics, as outlined in the 2023 Duke-ISCVID Criteria. A definite
pathological diagnosis can be made if organisms are identified on histologic analysis or culture of the
vegetation, intracardiac abscess, or peripheral embolus or if evidence of vegetation or intracardiac
abscess is confirmed by histologic analysis showing active endocarditis (*NEJM*, 2020, PMID:
**32757525**). As noted by the reviewer, **for the pathologically-defined IE cases** in our study, although an
initially definitive diagnosis could not be made due to the clinical criteria not being satisfied, such as the
absence of positive blood culture results or the lack of characteristic echocardiographic findings, **a**
**histopathological examination was conducted when the surgeon observed intra-operative signs**
**indicative of IE, ultimately leading to a diagnosis of definite IE based on histopathological criteria.**

**In the revision, we have revised** the improper statement of “met defined IE according to the Modified
Duke’s Criteria or met suspected IE criteria but confirmed by histopathology” in the original manuscript
to a more clear and precise description of “**met definite IE in accordance with the 2023 Duke-ISCVID**
**criteria**” in the “**Methods**” section of the revised manuscript, specifically under the subtitles: “Study
design and specimen acquisition” (**lines 736-737 on page 25**). Additionally, **we have provided a clearer**
**and more straightforward schematic of the study design (Figure RL1) in the Figure S1A** in the
updated **Supplementary Figures** file. Once again, we sincerely thank the reviewer for the thoughtful
and valuable comments, which have helped refine our manuscript and enhance its overall quality and
professionalism.

**Q7. Inclusion criteria apparently restricted/influenced patient enrollment. Subjects were included**
**IF culture and mNGS of blood and vegetation samples for pathogen identification had been**
**performed. How many of the screened subjects (initial cohort 4312 heart valve disease subjects)**
**who fulfilled clinical criteria for IE actually underwent culture and mNGS of blood and vegetation**
**samples?**

**Response to Q7:**

Thank you for your valuable feedback regarding the inclusion criteria and its impact on patient
enrollment. **We sincerely apologize for the inappropriate and unclear description** of the patient
inclusion criteria in the original manuscript. Regarding this point of the inclusion criteria, **what we**
**intended to convey was that all subjects in this study had definitive microbiological assessment**
**results**, including blood culture, vegetation culture, and/or pathogen identification through mNGS of
blood or vegetation samples, to support the diagnosis of IE.

**All participants included in our study underwent blood culture tests** during the diagnostic process
in our clinical practice; **if a definitive diagnosis could not be made** due to the lack of clear
microbiological evidence, **a vegetation culture was typically performed after surgical treatment to**

**further confirm the presence of the organism.** Additionally, a histopathological examination was
conducted when the surgeon observed intra-operative signs indicative of IE to support the diagnosis, as
outlined in the 2023 Duke-ISCVID Criteria. **Actually, not all patients included in our study**
**underwent mNGS of blood or vegetation samples, though mNGS, as an advanced molecular**
**diagnostic technique, is typically recommended at our center for patients with a high suspicion of**
**IE,** particularly when conventional methods are inconclusive, to increase the sensitivity and efficiency
of the diagnosis for IE (*Annu Rev Pathol*, 2019, PMID: 30355154).

**Our strategy follows the 2023 ESC guidelines** for a microbiological diagnostic algorithm in suspected
IE, **proposing that** tissue or prosthetic material obtained during surgery must undergo systematic culture,
histological examination, and 16S or 18S rRNA sequencing to document the presence of organisms (*EHJ*,
**2023, PMID: 37622656**). Many studies confirm that **mNGS is a valuable complement to** blood cultures,
vegetation cultures, and 16S or 18S rRNA sequencing in terms of etiologic diagnosis in IE patients
(*Annu Rev Pathol*, 2019, PMID: 30355154; *Clin Infect Dis*, 2022, PMID: 35362534). Earlier
identification of the etiologic pathogen allows for more prompt optimization of anti-infective treatment
regimens. **Therefore,** in our study, mNGS was typically recommended for the etiologic diagnosis of IE,
especially in patients with negative blood cultures.

In response to the reviewer’s insightful comments, **we have reviewed the clinical records of the initial**
**cohort 4,312 heart valve disease subjects and counted the number of individuals who actually**
**underwent culture and mNGS of blood and vegetation samples.** Among them, a total of 1,564 patients
originally underwent blood culture due to clinical suspicion of IE. Of these, 748 subjects underwent
vegetation culture, 393 cases received mNGS of blood samples, and 417 individuals underwent mNGS
of vegetation samples. We ultimately incorporated a total of 522 individuals from the initial cohort,
comprising the Cohort 1 (IE patients: n = 238; non-IE individuals: n = 100) and Cohort 2 (IE patients: n
= 92; non-IE individuals: n = 92) of this study. **Thus,** to provide clear microbiological evidence to support
the diagnosis of IE, **most suspected IE patients in our center commonly underwent various**
**microbiological testing methods,** including blood culture, vegetation culture, and mNGS, thereby
increasing the sensitivity of the diagnosis. **The prerequisite for conducting these examinations is**
**obtaining informed consent from the patient.** Consequently, the inappropriate description of
“underwent culture and mNGS of blood and vegetation samples from patients for pathogen identification”
in the original manuscript should be more accurately described as **“had definitive microbiological**
**assessment results, including blood culture, vegetation culture, and/or pathogen identification**
**through mNGS of blood or vegetation samples”.** Overall, these demonstrated that **patient enrollment**
**in our study was not significantly affected by the microbiological testing methods,** and there was
minimal potential for selection bias related to mNGS testing in our sample inclusion criteria.

**In the revision,** according to the reviewer’s valuable feedback, **we have incorporated a clearer**
**explanation of how the subjects in our study were selected** from an initial cohort of 4,312 patients
with heart valve disease (**Figure RL1**), which have been included in the updated **descriptions for the**
**inclusion and exclusion criteria** in the “Methods” section of the revised manuscript (**lines 738-740 on**
**page 25**) and the **Figure S1** in the updated **Supplementary Figures** file. We would like to express our
sincere appreciation once again to the reviewer for their insightful comments regarding the inclusion

criteria for patient enrollment, which assisted in improving the overall quality and professionalism of the
manuscript.

**Q8. There is no clear mention of the type of IE patients were affected by, i.e. native valve infection,**
**prosthetic valve infection, both. Were there any implantable cardiac device infection cases? If so,**
**which samples were actually taken?**

**Response to Q8:**

Thank you for your insightful comments. We sincerely apologize for the lack of clarity regarding the
types of valve infections in IE patients in the original manuscript, and we have clearly stated this
information in the revised manuscript. The detailed response was provided as follows.

**The majority of IE patients in our cohorts had native valve IE (NVIE), with only eight cases of**
**prosthetic valve IE (PVIE)—four in the Cohort 1, two in the Cohort 2, and two in the Cohort 3—while**
**the rest were NVIE. This was consistent with a recent study that reported NVIE as the most prevalent**
**form of IE, accounting for 90% of cases in a U.S. study conducted between 2003 and 2017 (JACC, 2024,**
**PMID: 38599719). Patients involving implanted cardiac devices were not included in our study.**
**Moreover, the IE patient cohorts in the study exhibited a pattern consistent with NVIE,**
**predominantly affecting the mitral and aortic valves.** In the discovery cohort (Cohort 1), the majority
of IE patients underwent surgical intervention for mitral valve repair or replacement (>70%) and/or aortic
valve surgery (>50%) (Table RL1).

**We would like to clarify that this study did not specifically exclude patients with PVIE.** The Global
Burden of Disease database highlights that the epidemiology of IE varies across regions and
socioeconomic contexts (JACC, 2020, PMID: 33309175). Consistent with previous research, the
primary type of IE is NVIE in areas where IE is predominantly induced by RHD, especially in developing
countries. For example, a study from Brazil showed that in a prospective cohort of 2,321 adult patients
with IE, all cases involved native valves (Braz J Infect Dis, 2024, PMID: 38971178). In contrast, in
developed countries, the proportion of PVIE is relatively higher, reportedly reaching up to 20% to 30%,
with this subset of patients primarily affected by staphylococcal infections (EHJ, 2019, PMID:
31504413). Notably, a national study from the United States indicated that between 2003 and 2017, there
were 646,325 hospitalized IE patients, of whom 585,974 (90%) had NVIE, 27,257 (4.2%) had cardiac
implantable electronic device-related IE (CIED-IE), and 26,111 (4%) had PVIE (J Am Heart Assoc,
2022, PMID: 36000421). Thus, the low incidence of PVIE in our cohort is understandable to some
extent. **Nonetheless, since our cohort only included IE patients from the past three years, the small**
**sample size might limit the representation of the actual proportion of infection types among IE**
**patients.** We have explicitly stated this limitation in the “Discussion” section of the revised manuscript
(lines 704-707 on page 24). **Specifically, as follows:**

“Our findings may have limitations in applicability to IE patients with other types of valvular infections,
such as prosthetic valves or implanted devices, as the majority of enrolled patients had native valve IE.”

Additionally, we have provided relevant statements of the type of valve infections in IE patients in the
“Results” section of the revised manuscript (lines 134-137 on page 5). Once again, we sincerely

appreciate the reviewer for their insightful comments, which contributed to enhancing the overall quality
 and professionalism of our manuscript.

Cohort 1 (n = 238)	
Mitral valve surgery:	
Replacement	69 (29.0%)
Repair	111 (46.6%)
No	58 (24.4%)
Aortic valve_ surgery:	
Replacement	127 (53.4%)
Repair	2 (0.84%)
Wheat	3 (1.26%)
Bentall	2 (0.84%)
No	104 (43.7%)
Tricuspid valve surgery:	
Replacement	2 (0.84%)
Repair	63 (26.5%)
No	173 (72.7%)
Pulmonary valve surgery:	
Replacement	1 (0.42%)
Repair	3 (1.26%)
No	234 (98.3%)

 **Table RL1.** Summarization of the proportion of various types of valve surgical interventions in IE
 patients within the discovery cohort (Cohort 1).

**Q9. Patients with severe infection were excluded. How was ‘severe infection’ defined?**

 **Response to Q9:**

Thank you for your insightful comments. We sincerely apologize for the unclear description regarding
 the term ‘severe infection’ in the exclusion criteria of the original manuscript. In response to the
 reviewer’s insightful comments, we have updated this point of the exclusion criteria to “**patients unable
 to tolerate surgery due to severe sepsis or septic shock**” in the revised manuscript. The detailed
 response was provided as follows.

 The 2023 ESC guidelines note that a significant proportion of patients with clear indications for surgery
 for IE may have multiple risk factors or other reasons that lead to surgery not being performed, and these
 patients have the worst prognosis. **Actually**, the “patients with severe infection” excluded from our study
 **referred to those who were in a severe infection status, including severe sepsis or septic shock.**
 **These patients often present with multiple organ dysfunction syndrome (MODS), indicating their**
 **poor overall condition, which makes them unable to tolerate surgery due to the high risk of death**
 **during or after the procedure.**

In the revision, according to the reviewer’s valuable feedback, we **have updated the descriptions**
**regarding the study exclusion criteria** in the “Methods” section of the revised manuscript (**lines 740-**
**741 on page 25**), and **have updated this point in the Figure S1A** in the updated **Supplementary**
**Figures** file.

**Q10. Which non-IE diseases were present in the 100 controls of the discovery set?**

**Response to Q10:**

We sincerely apologize for the lack of clarity regarding the definitions of the control group in the original
manuscript. In the revised manuscript, we have provided a comprehensive clarification on this point. The
detailed response was outlined as follows.

Reaching a rapid and accurate diagnosis in cases of suspected IE is a central challenge of the disease.
Delayed diagnosis and initiation of therapy lead to complications and worse clinical outcomes (*JACC*,
**2017, PMID: 28104075**). Initial clinical assessment of a patient with suspected IE involves evaluation
of risk factors and a search for a supportive history and examination findings. Core risk factors include
previous IE, a prosthetic valve or cardiac device, and valvular or congenital heart disease (*Lancet*, **2016**,
**PMID: 26341945**). Consequently, diagnosis should be considered for any patient with infection
symptoms, especially those with valvular damage. In our study, **the non-IE individuals (control group)**
**were** selected from patients with heart valve disease who exhibited clinical features suggestive of IE
(e.g., fever or heart murmur) or echocardiographic evidence of valvular lesions but were ultimately ruled
out for IE according to the 2023 Duke-ISCVID Criteria (**Figure RL1**). **The disease types of the non-**
**IE individuals primarily included rheumatic heart disease, degenerative valve disease, non-**
**bacterial thrombotic endocarditis, bacteremia, and Behçet’s disease**. For further details of the
clarification of definitions of the control group, please refer to the **Response to Q1**.

In the revision, according to the reviewer’s constructive comments, **we have incorporated a clearer**
**explanation of how the non-IE cases were selected** from an initial cohort of 4,312 patients with heart
valve disease, and **provided a more detailed and clear description of the definitions of the control**
**group** in the “Methods” section of the revised manuscript (**lines 743-748 on page 25**). Additionally, we
have provided relevant descriptions of the non-IE in the “Results” section of the revised manuscript
(**lines 122-124 on page 5**). We sincerely appreciate the reviewer for their insightful comments, which
enhanced the clarity and rigor of the manuscript.

**Q11. There is inconsistency in number of study patients:**

**Line 613: two cohorts: Cohort 1 (discovery set) with 238 IE patients and 100 non-IE individuals**
**Lines 622-624: In Cohort 1, 202 preoperative and 23 postoperative plasma samples (n=225) were**
**collected from 202 IE patients and 100 non-IE plasma samples for plasma proteomics analysis, and**
**158 patient vegetation samples were collected for vegetation proteomic analysis.**

**Please explain and/or modify.**

**Response to Q11:**

We sincerely apologize for any confusion caused by the unclear description of the number of study
patients in the original manuscript. **We have carefully reviewed the sections noted by the reviewer**

**and confirmed that the numbers of patients and samples mentioned in both instances are indeed**
**accurate.** In response to the reviewer’s insightful feedback, **we have refined the flow chart (Figure**
**RL3A) and the relevant descriptions in the revised manuscript** to enhance the clarity and readability
of the study cohort design and specimen acquisition. The specific clarifications were as follows:

**(1) About the Patient Numbers:**

**In the original manuscript,** we collected Cohort 1 with 238 IE patients and 100 non-IE individuals as
a discovery cohort and Cohort 2 with 92 IE patients and 92 non-IE individuals as an external validation
cohort from Guangdong Provincial People’s Hospital. **In the revision,** we have newly incorporated
another external validation cohort, namely Cohort 3, which included 72 IE patients and 72 non-IE
individuals. This new Cohort 3 was recruited from an external center, Heyuan People’s Hospital, to
further evaluate the reproducibility and generalizability of the developed models in this study (**Figure**
**RL3A**).

**(2) About the Sample Numbers:**

To comprehensively characterize the proteomic landscape of IE, in the discovery cohort (Cohort 1), we
conducted in-depth proteomic analyses on **three types of samples from the 238 IE patients:** i)
Preoperative plasma samples (n = 202): Collected at admission, before diagnosis and the initiation of any
admission treatments; ii) Postoperative plasma samples (n = 23): Collected within three to seven days
postoperatively; iii) Vegetation samples collected intraoperatively (n = 158). As not each patient was
able to provide all three types of samples in our study, the number of each type of sample did not fully
match the number of patients, which is a common occurrence in proteome research (*Nat Med*, 2022,
**PMID: 35654907**; *Nature*, 2022, **PMID: 34875674**). Specifically, for the 238 IE patients in the Cohort
1, we were able to collect admission plasma samples from 202 out of 238 IE patients, postoperative
plasma samples from 23 out of 238 IE patients, and vegetation samples from 158 out of 238 IE patients;
the detailed matched relationships of the three types of samples in the 238 patients is shown in **Figure**
**RL3B**. Meanwhile, to develop an early diagnostic prediction model for IE based on plasma protein
biomarkers, we also included admission plasma samples (n = 100) from 100 non-IE individuals as the
control group.

To validate the robustness and generalizability of the predictive models developed based on the Cohort
1 plasma proteomic data, we collected two external validation cohorts comprising plasma samples
collected at admission (**Figure RL3A**), namely Cohort 2 (IE: n = 92 and non-IE: n = 92) and Cohort 3
(IE: n = 72 and non-IE: n = 72).

In the revision, according to the reviewer’s valuable suggestions, we have provided a clearer and more
straightforward schematic of the study design (**Figure RL1**) to clearly delineate the different types of
samples collected, along with their respective numbers. This schematic has been incorporated in the
updated **Figure 1A and Figure S1A**. For further details, please refer to the “**Methods**” section of the
revised manuscript (**lines 762-791 on pages 26-27**). We sincerely thank the reviewer once again for their
significant suggestions, which have enhanced the clarity and readability of our study design and specimen
acquisition.

**Figure RL3.** (A) A schematic overview of the study cohort composition has been included in the revision.
(B) Euler diagram (left) and UpSet diagram (right) illustrating visual set relationships from the three
types of samples from the 238 IE patients in the Cohort 1. For instance, the number “108” indicates there
were 108 IE patients who had both admission plasma samples and matched vegetation samples collected;
the number “14” denotes there were 14 IE samples that had all three types of samples (admission plasma,
vegetation, and postoperative plasma); and the number “36” signifies there were 36 IE patients who had
only vegetation samples available.

**Q12.** It is unclear to me whether proteomic analysis was done on ‘vegetation samples’ (line 115) or
on ‘cryopulverized IE valve tissue’ (line 633). There may be significant proteomic differences
between vegetation samples, that are made of thrombotic material derived from coagulation
system activation and bacteria, and valve tissue that is cardiac in origin and made of completely
different cells and matrix.

**Response to Q12:**

Thank you for your valuable feedback. We sincerely apologize for the inconsistent and inappropriate
description of the “vegetation sample” as “cryopulverized IE valve tissue” in the original manuscript. As
noted by the reviewer, **our vegetation proteomic analysis was performed on ‘vegetation samples’**
**rather than ‘valve tissue’.**

In this study, the timing and indications for surgical intervention in IE patients were aligned with the
2023 ESC guidelines. Following the sampling method described in earlier studies (*JCI Insight*, 2020,
**PMID: 32544089**; *J Thorac Cardiovasc Surg*, 2014, **PMID: 24507402**), vegetation samples were
collected prospectively during open-heart surgery for IE, and all patient samples were obtained with
signed written informed consent. Here, **we collected a total of 158 vegetation samples from IE patients.**
**These samples were used to conduct comprehensive research on the vegetation proteome,**
**integrated with the matched plasma proteome of IE in our study.** We fully agree with the reviewer’s
insightful comments that **vegetation and valve tissue are completely different entities.** Vegetation is
the nidus of endocardial infection, made of thrombotic material derived from coagulation system
activation and bacteria, and is thought to form through interactions between pathogens and innate
immune mechanisms on damaged valvular or perivalvular tissue, resulting in aggregates of pathogens,
platelets, other cells, clotting factors, and various blood proteins (*NEJM*, 2020, **PMID: 32757525**).

**In the revision**, according to the reviewer’s insightful feedback, we have thoroughly reviewed and
modified the manuscript to ensure that all relevant descriptions consistently refer to “vegetation samples”.
Please refer to the “**Methods**” section in the revised manuscript for details (**lines 781-784 on page 27;**
**line 808 on page 27**). Once again, we sincerely appreciate the reviewer for the kind comments that have
enhanced the rigor and quality of our manuscript.

**Q13. Timing of plasma sample retrieval should be more clearly stated. There may be significant**
**differences between proteomic asset at IE onset/early phase after diagnosis and in the post-**
**operative phase. I know very well how difficult the timely retrieval of samples in a diverse**
**population of subjects, such as IE patients, may be, but a better standardization would be desirable.**
**Indeed, authors did not provide any sample size calculation, thereby implying that possibly more**
**than needed patients were studied.**

**Response to Q13:**

We sincerely appreciate your insightful comments regarding the timing and standardization of plasma
sample retrieval, as well as the sample size calculation in our study. We apologize for the insufficient
details provided on these aspects in the original manuscript. In the revision, according to the reviewer’s
suggestions, we have revised our manuscript to provide a clear and detailed elaboration on these aspects.
Below, we have divided our responses into two parts.

**(1) The Timing and Standardization of Plasma Sample Retrieval:**

As noted by the reviewer, the timing and standardization of plasma sample retrieval are essential for
ensuring the reliability and comparability of our findings. We apologize for the insufficient details
provided in the original manuscript. Our plasma samples were collected following standardized
guidelines to ensure consistency and minimize pre-analytical variability. **In the revised manuscript, we**
**have now included a comprehensive description of our sample collection protocol.**

**Specifically:**

- • **Timing of Sample Collection:** The detailed information about the sample design for this study
was illustrated in the above figure (**Figure RL1**). There were two types of plasma samples
included in our study: i) Preoperative plasma: Collected upon admission, before diagnosis and
the initiation of any admission treatments. ii) Postoperative plasma: Collected within three to
seven days postoperatively. This approach allowed us to capture the proteomic profile at both
the disease onset and post-intervention stages. In this study, the plasma proteomic analysis was
primarily based on proteomic data obtained from preoperative samples, except for those clearly
indicated as derived from postoperative plasma samples, namely results related to **Figure 7**.
- • **Sample Collection Protocol:** We adhered to the Clinical and Laboratory Standards Institute
(CLSI) guidelines GP41-A6, “Procedures for the Collection of Diagnostic Blood Specimens by
Venipuncture” (*J Med Biochem*, 2015, PMID: 28356839). Venipuncture was performed using
standardized equipment and procedures to reduce variability.
- • **Sample Volume and Anticoagulant:** For each collection, 3 ml of blood was drawn into EDTA-
treated tubes to prevent coagulation and preserve protein integrity.
- • **Sample Processing Time:** All blood samples were processed within 2 hours of collection to
minimize ex vivo changes in the proteomic profile.

- • Sample Processing and Storage: Plasma was separated by centrifugation at $1,500 \times g$ for 10
minutes at 4°C . The plasma aliquots were immediately stored at -80°C to preserve protein
integrity.
- • Vegetation Sample Collection: In addition to plasma samples, we collected vegetation samples
intraoperatively, following the sampling method described in earlier studies (*JCI Insight*, 2020,
**PMID: 32544089**; *J Thorac Cardiovasc Surg*, 2014, **PMID: 24507402**), to conduct vegetation
proteome research of IE in our study.
- • Storage of Samples: Both plasma and vegetation samples were stored at -80°C until analysis to
maintain sample integrity.
- • Patient Selection Criteria: To ensure a homogeneous study population and minimize
confounding factors, we implemented strict inclusion and exclusion criteria: i) Inclusion criteria:
definite IE and non-IE individuals diagnosed according to the 2023 Duke-ISCVID Criteria;
patients with informed consent; scheduled for surgical intervention; and age ≥ 18 years. ii)
Exclusion criteria: patients unable to tolerate surgery due to severe sepsis or septic shock; severe
end-stage liver disease (Child-Pugh class C); patients who had received immunosuppressive or
immunomodulatory therapy; known pregnancy.
- • Standardization of Collection Conditions: All samples were collected under fasting conditions
(8-12 hours overnight fast) to minimize the influence of diet on the plasma proteome.
- • Sample Tracking: Each sample was assigned a unique identifier and logged into a secure
database, recording the exact time and date of collection, processing, and storage.

The sample collection and processing methods we employed are widely accepted and commonly used in
clinical disease research and proteomic studies (*Cell System*, 2016, **PMID: 27135364**; *EMBO Mol Med*,
**2019, PMID: 31566909**). This standardized approach to sample collection and processing is crucial for
obtaining reliable and reproducible proteomic data.

(2) Comprehensive Sample Size Calculation and Statistical Power Analysis:

We fully agree with the reviewer's comments that appropriate sample size calculation is essential to
ensure that the findings have adequate statistical power to detect meaningful differences in clinical cohort
studies. We apologize for the insufficient details provided in the original manuscript. **Actually**, we
performed sample size estimation during the study design phase. **In the revised manuscript**, we have
now included a clear elaboration of the sample size calculation.

Specifically, before initiating the study, we conducted a thorough statistical power analysis to determine
the minimum sample size required to detect a clinically significant effect with acceptable levels of
statistical confidence (80% power at a 5% significance level). This analysis was conducted using
G*Power software (version 3.1) (*Behavior Research Methods*, 2009, **PMID: 19897823**). In line with
established standards (*Psychol Bull*, 1992, **PMID: 19565683**), we anticipated an effect size (Cohen's d)
of 0.5 to represent a moderate effect in our study findings between the two groups. **As a result**, assuming
a two-tailed test, an expected effect size of 0.5, a typical significance level (α) of 0.05, and aiming for a
power ($1 - \beta$) of 0.8, the minimum required sample size was calculated to be 134 patients for two groups
(**Figure RL4A**). With a more stringent significance level (α) of 0.01, the required total sample size
increased to 200 patients for two groups. (**Figure RL4B**). **To account for potential dropouts or**
**unusable samples due to unforeseen circumstances, we initially recruited a slightly larger cohort**

and selected individuals according to the inclusion and exclusion criteria. Ultimately, we
 incorporated 238 IE patients and 100 non-IE individuals as our discovery cohort, along with two external
 validation cohorts of 184 and 144 individuals, respectively. In the subsequent analysis, the p value was
 computed using a two-sided Wilcoxon rank-sum test and adjusted (false discovery rate, FDR) using
 Benjamini–Hochberg correction. The significance criterion for the differential proteins between IE and
 non-IE was strictly set at an FDR of less than 0.05. **Overall, conducting sample size calculations was**
 **essential in guiding us toward an appropriate sample size that would enable us to achieve results**
 **with both clinical significance and statistical power.**

 **Figure RL4. Statistical power analysis to determine the required minimum sample size.** The scatter
 plot illustrates the relationship between statistical power (1 - β error probability) and the total sample
 size needed for a two-tailed Wilcoxon-Mann-Whitney test (comparing two groups). The top panel shows
 that the parent distribution is assumed to be normal, with an allocation ratio of $N_2/N_1 = 1$ and α error
 probability set to 0.05 (A). The bottom panel shows that the parent distribution is assumed to be normal,
 with an allocation ratio of $N_2/N_1 = 1$ and α error probability set to 0.01 (B). The different lines represent
 varying effect sizes (d), ranging from 0.5 to 0.8.

 **In the revised manuscript,** we have updated the section related to the timing and standardization of
 plasma sample retrieval, as well as the sample size calculation in the “**Methods**” section. These revisions
 are detailed under the subtitles: “Sample collection.” (lines 773-791 on pages 26-27), and “Sample size
 calculation and statistical power analysis.” (lines 793-804 on page 27). We greatly appreciate the
 reviewer’s constructive suggestions, which have improved the rigor and quality of our manuscript.

 **Q14. NYHA functional classification is not the best way to classify IE patients. This classification**
 **focuses on the limitations of patients in daily activities, which is not the case for hospitalized**
 **subjects with IE undergoing heart valve surgery.**

**Response to Q14:**

Thank you for your valuable feedback. We fully agree with the reviewer’s insightful comments that
NYHA functional classification is not the best way to classify IE patients. We would like to clarify that
**the classification method used for patient risk stratification in our study was not arbitrary; rather,**
**it was grounded in a systematic analysis.** The rationale for classifying IE patients into two groups—
NYHA class IV and NYHA class II/III—based on the NYHA functional classification was that our
analysis revealed that **the newly identified plasma biomarkers associated with patients with NYHA**
**class IV heart failure might facilitate early risk stratification for surgical interventions in IE.**
Nonetheless, as noted by the reviewer, this classification may not be the optimal approach due to **its**
**potential inadequacy in accurately predicting all high-risk IE patients. In the revision, we have**
**provided a discussion on this limitation in the revised manuscript.** The detailed response was
provided as follows.

**(1) About the Rationale and Significance for Distinguishing Patients Based on the NYHA**
**Functional Classification:**

Nearly half of IE patients require surgical intervention, and this proportion has been gradually increasing
in recent years. Therefore, **preoperative risk assessment is particularly important for clinical**
**decision-making.** According to the 2023 ESC IE Guidelines, although several scoring systems have
been specifically designed for the context of IE, such as the AEPEI (Association for the Study and
Prevention of IE) score, the STS (Society of Thoracic Surgeons) IE score, and the ANCLA (anaemia,
NYHA class IV, critical state, large intracardiac destruction, surgery of thoracic aorta) score, **these**
**systems’ performance is variable, and none of these scoring systems are routinely used in clinical**
**practice.** Thus, **there is still a need to continue efforts to develop more accurate and widely**
**applicable preoperative stratification models.**

It is known that valve dysfunction resulting in heart failure is the most frequent complication of IE and
the main indication for surgical treatment for IE. The NYHA functional classification has long served as
a foundational tool for risk stratification of heart failure, determining clinical trial eligibility and
candidacy for drugs and devices, and is widely used in clinical practice (*J Am Heart Assoc*, 2019, PMID:
**31771438**). **In our preliminary analysis,** we observed a significant negative correlation between NYHA
functional classification and a plasma protein co-expression functional network (**Figure RL2J**).
Subsequent analyses revealed that IE patients classified as NYHA class IV exhibited substantial
differences compared to those with lower levels of heart failure (NYHA class II/III) in terms of both
clinical and proteomic characteristics (**Figure RL2K**). Specifically, factors such as advanced age, valve
damage, liver dysfunction (defined by elevated levels of AST, DBIL, etc.), high postoperative adverse
events (e.g., reoperation, LCOS, long-time ICU stay), and high mortality (e.g., D30) were notably
enriched in NYHA class IV patients, compared to those in NYHA class II and III. Additionally, we
detected a decrease in the levels of many liver-derived plasma proteins, including complement (e.g. C3
and C4), HDL (e.g. APOA1 and APOA2), and serine protease inhibitors (e.g. SERPINC1), in NYHA
class IV patients, while an increase in pathways comprising neutrophil degranulation, cell adhesion, and
proteolysis (**Figure 3**). These biological alterations could lead to increased heightened vulnerability to
infections (*J Autoimmun*, 2023, PMID: **36535812**) and tissue destruction (*NEJM*, 1989, PMID:
**2536474**) for IE patients. Taken together, **these findings indicate the need for risk stratification of**
**NYHA class IV patients in IE, who often face higher postoperative risk.**

Thus, we developed a high-performance machine learning model (AUC = 0.873) for preoperative risk
assessment in IE to differentiate between NYHA class IV patients and those in NYHA class II and III,
utilizing integrative plasma proteomic and clinical blood test data from plasma samples collected upon
admission. **In the revision**, we have further validated the robustness of this model in a newly
incorporated external validation cohort (AUC = 0.829). The identified biomarker combination, including
4 CBMs (e.g., DBIL and BUN) and 11 plasma proteins, including ADIPOQ, CST3, APOA1, and FETUB,
could aid in the early prediction of surgical risk stratification in IE (**Figure 4**). For further details of the
clarification for the rationale and significance, please refer to the **Response to Q2**.

**(2) About the Potential Limitation for Distinguishing Patients Based on the NYHA Functional**
**Classification:**

Although our blood biomarker-based preoperative risk model, which adopts a clinically fundamental
classification method to differentiate between NYHA class IV patients and those in NYHA class II/III,
might improve the convenience and efficiency of clinical decision-making, **NYHA class IV patients do**
**not completely and accurately represent all postoperative patients at high risk for IE**. Additionally,
due to the limited patient types and short follow-up period, larger-scale longitudinal investigations are
necessary to further explore this potential link. Consequently, **the generalizability and accuracy of this**
**model need further validation in larger and more diverse patient populations**.

In sum, we have revised our manuscript to provide a clear discussion on this point in the limitation section.
**Specifically, as follows:**

“While our findings underscore the need for risk stratification of NYHA class IV patients due to their
higher postoperative risk of IE, our prognostic model based on NYHA functional classification has
limitations, as class IV patients do not fully reflect all postoperative patients at high risk for IE. Given
the limited patient types and short follow-up period in our study, larger-scale longitudinal investigations
are needed to further explore this potential link and validate the prognostic model’s generalizability and
accuracy in more diverse populations.”

For further details, please refer to the “**Discussion**” section of the revised manuscript (**lines 602-608 on**
**page 21; lines 707-710 on page 24**). We greatly appreciate the reviewer’s valuable feedback, which
improves the rigorousness and understandability of our study.

**Q15. The actual microbial etiology of included IE patients deserves consideration: there is a very**
**high preponderance of Streptococci (58.97%), and a very high rate of C. burnetii (10.26%), with**
**in contrast very few cases due to S. aureus (8.33%) or CoN Staph (6.41%). This is another**
**limitation of the study when trying to apply findings to the broader population of IE patients where**
**predominant pathogens are S. aureus, CoNS and enterococci, in addition to streptococci.**

**Response to Q15:**

We appreciate the reviewer’s thoughtful comments that the regional specificity of our patient population
might limit the generalizability of our findings, due to there are indeed significant geographical
differences in the global epidemiology of IE. In accordance with the reviewer's insightful suggestions,

we have carefully considered and **provided a more in-depth discussion on the limitations of our**
**findings** in the revised manuscript, underscoring the need for future global research to improve the
assessment of our findings. Additionally, **we have modified our manuscript to visually present the**
**actual microbial etiology of included IE patients in our findings**. The detailed response was provided
as follows.

**(1) The Current Differences in Global Microbiological Spectrum:**

In recent years, the microbial etiology of IE has shown significant geographical differences, particularly
between developed countries and developing countries, though globally, *Staphylococcus*, *Streptococcus*,
and *Enterococcus* are the top three most prevalent microorganisms for IE, accounting for about 80% of
cases (*Lancet*, 2024, PMID: 39067905). Actually, different from developed countries, where in addition
to streptococci, predominant pathogens include *Staphylococcus* (commonly *S. aureus* and CoNS) and
enterococci, **Streptococcal IE remains most common in developing countries, primarily caused by**
**viridans group streptococci and underlies up to two-thirds of cases**. This is mainly due to
improvements in medical and living conditions in developed countries, which have led to an increase in
the incidence of intravenous-drug users and patients with prosthetic-valve IE or healthcare-related IE,
typically resulting in Staphylococci infections (*JAMA*, 2018, PMID: 29971402; *Lancet*, 2016, PMID:
26341945).

**(2) Preliminary Evaluation of the Efficacy of Identified Biomarkers in Reorganized Test Cohort:**

Among 156 IE patients with positive mNGS results documented in electronic health records from Cohort
1 (**Figure 6A**), the top four pathogens were *Streptococcus* (58.97%), *C. burnetii* (10.26%), *S. aureus*
(8.33%), and CoNS (6.41%). To more accurately evaluate the microbiological spectrum of IE in our
study, **we first recalculated the proportions of pathogens in the overall plasma cohort of the**
**discovery cohort (Cohort 1)**, including cases with both negative and positive mNGS results and
compared them with those reported in previous research (**Table RL2**). In line with epidemiological
trends, more than half of the IE patients in our study had streptococcal infections. For a more thorough
assessment of the potential limitations associated with the microbiological spectrum in our findings, **we**
**randomly selected IE patients based on pathogen proportions typical of developed countries** and
matched them with an equal number of non-IE individuals **from the Cohort 1, creating a reorganized**
**test cohort for validation (Table RL2)**. As a result, in the reorganized test cohort, we validated that the
plasma biomarkers included in the diagnostic prediction and prognostic models showed significant
differences between IE and non-IE cases, as well as between patients with NYHA class IV and those in
NYHA class II and III, respectively, consistent with the results from Cohort 1 (**Figure RL5A-D**). **This**
**demonstrated that** our model exhibited good generalizability and were largely unaffected by changes
in pathogen proportions, further supporting their promising potential for clinical practice in IE.

**(3) About the Discussion on the Limitations in This Regard:**

As the reviewer commented, although our cohort includes typical pathogens consistent with those in
developed countries, the proportions differ significantly, which could limit the generalization of our
conclusions. **Despite the above preliminary evaluation** in a broader population of IE patients where
predominant pathogens are *S. aureus* (n = 13, 30.23%) in addition to streptococci (n = 13, 30.23%)
**supports our findings, firm conclusions cannot be drawn from this trial on the efficacy of our**

**identified biomarkers, given the small sample size of the reorganized test cohort.** Further evaluation
in global collaborative research with broader populations is needed.

In sum, in the revised manuscript, we have incorporated this limitation in the discussion section as
follows:

“The predominance of streptococcal infections in our study, consistent with the microbiological spectrum
observed in developing countries, might limit the applicability of our findings to the broader IE
population, where pathogens such as *S. aureus*, CoNS, and enterococci are also commonly encountered.
Therefore, further evaluation through global collaborative research involving larger and more diverse
populations is essential.”

For more details, please refer to **lines 710-714 on page 24** in the “**Discussion**” section of the revised
manuscript. Additionally, we have included the supplementary analysis results of the validation in the
reorganized test cohort from Cohort 1 in the “**Results**” section of the revised manuscript (**lines 185-193**
**on page 7; lines 350-354 on page 12**), as well as the **Figure S3** in the updated **Supplementary Figures**
file. Thank you again to the reviewer for their thoughtful review and suggestions, which have improved
the quality and rigor of our work.

Pathogens	Proportion of pathogens in our Cohort 1¹	Reported proportion of pathogens in developed countries²	Proportion of pathogens in our reorganized test cohort³
Streptococci	92 (52.27%)	30%	13 (30.23%)
S. aureus	13 (7.39%)	30%	13 (30.23%)
CoNS	10 (5.68%)	10%	4 (9.30%)
Enterococci	2 (1.14%)	10%	2 (4.65%)
C. burnetii	16 (9.09%)	<5%	1 (2.33%)
Bartonella	1 (0.57%)	<5%	1 (2.33%)
Fungi	2 (1.14%)	<5%	1 (2.33%)
Others	20 (11.36%)	10%	4 (9.30%)
Negative	20 (11.36%)	10%	4 (9.30%)

1. Plasma samples of Cohort 1 (n = 176) exhibiting definitive pathogen identification results in the electronic health records.

2. Proportion of pathogens reported in developed countries (PMID: 26341945; PMID: 31504413; PMID: 39067905).

3. We randomly selected plasma samples from Cohort 1 and recombined them into a new recombination cohort (n = 43) with the
1154 proportion of pathogens consistent with that in developed countries to validate our results.

**Table RL2.** Microbiology of IE in specific patient populations.

Figure RL5. (A, B) Heatmap showing the expression profiles of the 10 plasma proteins identified in the diagnostic model across patients in the Cohort 1 and reorganized test cohort. Each column represents a patient sample and rows indicate proteins. (C, D) Heatmap depicting the expression profiles of the 11 plasma proteins identified in the prognostic model across patients in the Cohort 1 and reorganized test cohort. The color range in the heatmap represents the row z-score of the normalized protein expression values. The annotation of pathogen identified by mNGS for each sample is displayed above the heatmap, with blanks indicating missing records.

Reviewer #2 (Remarks to the Author):

The authors presented a solid machine learning pipeline. The trained models performed well, demonstrating the applicability of plasma proteomics to differentiate preoperative IE and non-IE patients, as well as NYHA class IV from NYHA class II and III patients. The use of a second cohort to validate the models strengthens their findings.

Although the overall work results and methods are solid, some points about the machine learning pipeline could be clarified to understand the findings better. Therefore, I list in the sequence some questions that arose during the review:

Response:

We greatly appreciate your positive and constructive feedback, which has been instrumental in refining our manuscript, leading to improved clarity, rigor, and overall quality. In the revision, in response to the reviewer’s insightful comments, we have carefully considered all suggestions and have systematically revised the manuscript, including supplementing the suggested extra analyses, experimental data, and

1183 pertinent clarifications to address each question raised by the reviewer. **All changes in the manuscript**
**have been marked in red for clarity.** The detailed point-by-point responses were provided as follows.

**Q16. (lines 758- 774) The authors state that the data was split randomly into train and test. Since**
**the groups are sometimes unbalanced, the usual approach would be to use a stratified randomized**
**split.**

**The authors discuss the feature selection method employed during model training. Among the**
**feature selection methods employed, the authors used statistical tests. In this part, were the**
**statistical tests performed only in the training dataset or the total data? This could be considered**
**data leakage if they were performed in the total data.**

**Response to Q16:**

We sincerely apologize for the unclear description of the methodology used in the machine learning
pipeline in the original manuscript. We appreciate the reviewer’s insightful comments regarding **the**
**handling of class imbalance and the specifics of the statistical tests used in the feature selection**
**steps during model training within the machine learning pipeline.** To clearly address the reviewer’s
questions, we have divided our response into two parts as follows.

**(1) About the Data Processing for Addressing Class Imbalance in the Training and Testing Splits:**
**We fully agree with the reviewer that a stratified randomized split is a more appropriate and**
**common approach for handling an unbalanced class dataset.** This method first stratifies the entire
dataset into subgroups based on class labels, then followed by simple random sampling from the stratified
subgroups to ensure the final training and test sets reflect the original data distribution. This data split
strategy enables a more reliable evaluation of the model’s performance.

Initially, we employed a simple random split approach in the original manuscript, dividing the data into
a 70% training set and a 30% hold-out test set for model training and evaluation. **In the revised**
**manuscript,** according to the reviewer’s constructive suggestions, we addressed the potential class
imbalance in the training and test sets by **employing the stratified randomized split approach with a**
**70/30 split.** This stratification was implemented using the “train_test_split()” function from the
“sklearn.model_selection” module in Python, with the “stratify=y” parameter. Meanwhile, in this
revision, **we have added a schematic workflow for the development and evaluation of machine**
**learning models to enhance the clarify and intelligibility (Figure RL6A).** For instance, we used the
proteomic data derived from 202 IE and 100 non-IE plasma samples from the discovery cohort (Cohort
1) to develop a diagnostic prediction model that distinguished IE from non-IE individuals. The discovery
cohort dataset was stratified to form a 70% training set, comprising 211 individuals—141 from IE
samples and 70 from non-IE samples—and a 30% hold-out test set, consisting of 91 individuals—61
from IE samples and 30 from non-IE samples. This stratification ensured representative distributions of
the target variable **(Figure RL6A).**

Figure RL6A. Schematic workflow for developing and evaluating the machine learning model. The discovery cohort (Cohort 1) was divided into a training set (70%) and a test set (30%) with a stratified randomized split during model derivation. The feature selection was conducted on the training set to identify the most significant features contributing to the model's performance to distinguish between different classes. Subsequently, hyperparameter optimization was performed on the training set, and the optimal parameters were selected. In the machine learning architecture, hyperparameters were tuned using the Bayesian optimization with the tree-structured Parzen estimator (TPE) algorithm (*Neurocomputing*, 2020, <https://doi.org/10.1016/j.neucom.2020.07.061>), employing five-fold cross-validation. Ultimately, the final model was tested on the hold-out test set. In addition to internal validation, the final model was validated in external independent, unseen datasets to assess its generalizability. The external independent validation included two cohorts: Cohort 2 and Cohort 3.

Consequently, we retrained the models and evaluated their classification performance on the hold-out test set for internal validation. Moreover, to further assess the generalizability of our constructed classification models, we have newly collected an external validation cohort (Cohort 3) from outside center for validation. External validation was performed on the models using external validation cohorts: Cohort 2 and Cohort 3, confirming their robustness and effectiveness in various unseen

scenarios. Compared to the results in the original manuscript, the revision for the development of
 models included the following improvements: 1) The updated model exhibited better performance
 metrics than the original version; 2) The updated model demonstrated a reduction in complexity while
 preserving its initial effectiveness, entailing a reduction in features; 3) In the newly incorporated external
 validation cohort recruited from an external center (Cohort 3), we further validated the generalizability
 and robustness of the updated model.

 **For example**, as a result, we developed a diagnostic prediction model to distinguish IE from non-IE
 individuals, containing 10 proteins (**Figure RL6B-F**), including 7 that were upregulated in IE (**Figure**
 **RL6D**), such as leucine rich alpha-2-glycoprotein 1 (LRG1), and 3 that were downregulated, including
 apolipoprotein A2 (APOA2). This model achieved an overall area under the curve (AUC) of 0.97, which
 was further validated in the two external validation cohorts with overall AUC values of 0.98 and 0.94,
 respectively (**Figure RL6E**). Compared to the original model, these demonstrated that the updated
 model maintained its initial effectiveness while reducing complexity and redundancy by narrowing the
 feature set from 16 to 10. Among the 10 features, 8 features were selected from the original set, while
 actin gamma 1 (ACTG1) and superoxide dismutase 3 (SOD3) were newly identified. The combination
 of the 10 features preserved the original discriminatory power of the 16-feature combinations, thereby
 enhancing the model's efficiency and effectiveness in practice. In addition, the reproducibility and
 generalizability of the updated model were further validated in the Cohort 3. These findings
 demonstrated the outstanding discriminatory performance of the updated diagnostic prediction model for
 IE.

 **Figure RL6B-F.** (B) Bar plot illustrating the weighted feature importance of the top 20 plasma proteins
 identified by the ensemble model to distinguish between IE and non-IE samples. The error bars represent
 the standard error of the feature importance estimates. (C) Scatter line plot illustrating the optimal feature
 combinations for the model that yield the highest accuracy, determined through recursive feature

elimination cross-validation (RFECV). (D) Heatmap showing the expression profiles of the 10 plasma
proteins identified in the developed model. Each column represents a patient sample and rows indicate
proteins. The color range in the heatmap represents the row z-score of the normalized protein expression
values ranging from -1.5 (blue) to +1.5 (red). The annotation of pathogen identified by metagenomic
next-generation sequencing (mNGS) for each sample is displayed above the heatmap, with blanks
indicating missing records. (E, F) Receiver Operating Characteristic (ROC) curves (E) and confusion
matrices (F) illustrating the performance of the developed model on the hold-out test set of the discovery
cohort (Cohort 1) and two external validation cohorts: Cohort 2 and Cohort 3.

**(2) About the Detailed Descriptions of the Feature Selection:**

We sincerely appreciate the reviewer for their careful review and for highlighting potential confusion
regarding the feature selection process in the machine learning model architecture. We apologize for any
confusion caused by the unclear explanation of the statistical test methods performed in feature selection
in the original manuscript. Actually, in developing the classification models, **we have carefully**
**considered potential challenges in the model training and model evaluation processes, such as data**
**leakage, model underfitting, and model overfitting.** We strictly adhered to standard model
construction practices to mitigate these potential issues and limitations. **All the model training**
**processes, including feature selection, hyperparameter optimization, and the final model training, were**
**exclusively conducted on the training set to prevent data leakage, ensuring the robustness and**
**reliability of the models.** In this revision, we have provided a detailed description for the feature
selection steps implemented during model training, as follows:

**“Feature selection.** The discovery cohort was divided into a 70% training set and a 30% hold-out test
set using a stratified randomized split approach during model derivation. To avoid potential data leakage,
all model training processes, including feature selection, hyperparameter optimization, and the final
model training, were exclusively conducted on the training set. Specifically, feature selection was
performed on the training set to identify the most significant features contributing to the model’s ability
to distinguish between different classes. Firstly, we identified the differentially expressed proteins (DEPs)
between different classes using the statistical test methods. After the determination of DEPs, we
employed an ensemble learning scheme for further feature selection, which has been widely applied in
previous studies (*Cell*, 2022, PMID: 35917817; *Front Microbiol*, 2022, PMID: 35910662). Specifically,
we employed an ensemble of machine learning models to evaluate feature importance, including Random
Forest, XGBoost, LightGBM, CatBoost, Support Vector Classifier (SVC), and Logistic Regression. Each
model was trained on the entire training set to assess the relative contribution of each feature to the
model’s predictions. The method of determining feature importance varied across models: tree-based
models (XGBoost, Random Forest, LightGBM, CatBoost) measured importance based on a feature’s
frequency in making split decisions at each tree node, while linear models (SVC, Logistic Regression)
relied on feature coefficients. To reduce the error and generate a more robust feature importance ranking,
this process was performed with a 5-fold cross-validation procedure to compute the average importance
scores of each feature. The final feature importance ranking was determined by aggregating the results
from six feature-ranking algorithms using a weighted voting approach. All models were implemented in
Python using the “scikit-learn” module. By quantifying the relative importance of each feature using this
ensemble method, we can accurately and robustly identify key features that most significantly contribute
to the model’s ability to discriminate between classes. Finally, we further refined the selection of the

previously identified top-ranked features for feature importance by employing the Recursive Feature
Elimination with Cross-Validation (RFECV) method, implemented in Python using the ‘scikit-learn’
module. This process ensured a meticulous selection of the most critical features by systematically
eliminating less informative features, leading to a more simplified and robust feature set (*Machine*
*Learning*, 2002, <https://doi.org/10.1023/A:1012487302797>).

In summary, in response to the reviewer’s constructive and valuable feedback, we have supplemented a
schematic workflow for the development and evaluation of machine learning models (**Figure RL6A**) in
**Figure S1B** in the updated **Supplementary Figures** file. Additionally, we have provided a more detailed
and clear description of the methodology used in the machine learning pipeline in the “**Methods**” section
of the revised manuscript, specifically under the main title: “Construction of the machine learning
classification models.” (**lines 933-1058 on pages 32-36**). The revisions related to data split using a
stratified randomized split approach with a 70/30 split to handle the class imbalance are detailed under
the subheading: “(1) Dataset and model training.” (**lines 944-960 on pages 32-33**). The revisions related
to the statistical test methods performed in feature selection are detailed under the subheading: “(2)
Feature selection.” (**lines 962-988 on page 33**). Moreover, the latest relevant methods and results have
been incorporated in the updated figures and text throughout the revised manuscript (**Figure 1 and**
**Figure 4**). For more details, please refer to “**Results**” section (**lines 167-183 on page 7; lines 322-358**
**on pages 11-12**) and “**Discussion**” section (**lines 558-582 on pages 19-20; lines 629-647 on page 22**)
of the revised manuscript. We would like to express our gratitude to the reviewer again for their insightful
and thoughtful comments, which have helped us improve the manuscript’s clarity, accuracy, and overall
quality.

**Q17. (lines 778-780) The authors discuss the model evaluation and hyperparameter tuning. In this**
**part, it would be good to have some extra information. How was the hyperparameter tuning**
**performed? Was it a cross-validation, and were the best-scoring fold hyperparameters selected?**
**Or was the model run several times, and the mean performance was used to select the best set of**
**hyperparameters? Was the model evaluated after hyperparameter tuning? The phrasing places**
**model evaluation before hyperparameter tuning and it might generate some confusion.**

**Response to Q17:**

We sincerely appreciate the reviewer’s constructive comments on **the need for a more detailed**
**description of the hyperparameter tuning and model evaluation processes of the machine learning**
**architecture**. We apologize for any confusion caused by the inaccuracies and unclear descriptions in this
section of the original manuscript. According to the reviewer’s suggestions, **we have provided a**
**comprehensive description of the classification model development** in the revised manuscript to
ensure clarity regarding all processes and parameters, **especially for the hyperparameter tuning and**
**model evaluation steps implemented within the machine learning architecture**. Moreover, to clarify
the framework of the machine learning pipeline in this study, **we have illustrated a schematic workflow**
**for developing the machine learning model (Figure RL6A)**. In response to the reviewer’s inquiries,
we have outlined our clarifications as follows.

Following the feature selection, **hyperparameter optimization was performed exclusively on the 70%**
**training set** to identify the optimal parameters. **Hyperparameters were tuned using the Bayesian**

**optimization with 5-fold cross-validation**, which was implemented via the open-source “HyperOpt”
Python package (version 0.2.5). During 5-fold cross-validation, the training set was divided into five
subsets (or folds). For each iteration: the model was trained on four of the five folds and then evaluated
on the remaining fold (the test set for that iteration). This process was repeated five times, each time
using a different fold as the test set, ensuring that every data point was used for both training and testing.
This approach maximizes data utilization and reduces the risk of overfitting, thereby providing a more
reliable estimate of the model’s generalization performance. **The performance for each set of**
**hyperparameters was measured using the Receiver Operating Characteristic Area Under the**
**Curve (ROC AUC) metric averaged over the cross-validation folds.** Subsequently, the previously
constructed model with determined features was retrained with the optimal hyperparameters. **Once**
**established, the final model was locked to ensure rigorous testing and validation.** To prevent
information leakage, this procedure maintained the complete independence of the 30% hold-out test set,
which was not utilized during any stage of the model training process, to prevent information leakage.
Finally, the final locked model was then tested on the hold-out test set for internal evaluation and the
external independent, unseen datasets to evaluate its generalizability performance. **The detailed**
**descriptions of the hyperparameter tuning and model evaluation processes of the machine learning**
**architecture**, including specific parameter settings and configurations, **were provided below:**

*“Hyperparameter optimization.* Machine learning models with properly tuned hyperparameters
significantly improve the model’s accuracy, training speed, and ability to generalize to new data to
achieve state-of-the-art results. To optimize the hyperparameters of the XGBoost-based model, we
employed Bayesian optimization using the tree-structured Parzen estimator (TPE) algorithm
(*Neurocomputing*, 2020, <https://doi.org/10.1016/j.neucom.2020.07.061>) with 5-fold cross-validation on
the entire training set. The specific hyperparameters were defined as a dictionary with the following
hyperparameters:

• Number of rounds (num_boost_round): This parameter determines the number of boosting rounds or
trees to build. It is important to tune it properly to avoid overfitting. We explored values between 100
and 1000.
- • Maximum depth of trees (max_depth): The maximum depth of the tree is defined as follows: We
tested depths from 3 to 18.
- • Gamma (gamma): Specifies the minimum loss reduction needed to make a split. The search space
was between 0.1 and 1 in increments of 0.25.
- • Regularization (L1 and L2: reg_alpha and reg_lambda, respectively): These add regularization terms
to the objective function. We searched within a range of 0 to 1 in increments of 0.01 for both parameters.
- • Column Sampling by Tree (colsample_bytree): Denotes the fraction of features to be randomly
sampled for building each tree. We tested values ranging from 0.5 to 1 in increments of 0.05.
- • Minimum Child Weight (min_child_weight): This parameter determines the minimum sum of the
instance weights needed for a child. A search was performed between 0 and 10.
- • Learning rate (eta): This makes the optimization more robust by shrinking the weights in each step.
We used a range from 0.025 to 0.5 in increments of 0.05.
- • Subsample: This denotes the fraction of observations to be randomly sampled for each tree. The range
for this hyperparameter was between 0.5 and 1 in increments of 0.05.

This hyperparameter grid was used as input to a hyperparameter optimization algorithm to search for the
optimal combination of hyperparameters that maximized the area under the ROC curve (*Pattern*
*Recognit*, 2015, PMID: 25395692). We defined the objective function “objective(space)”, which took a
hyperparameter space as input and returned a dictionary containing the loss and status values. This
function was used in Bayesian optimization to evaluate the performance of the XGBoost model. For the
training process, we utilized the ‘gbtree’ booster and the exact tree method. Furthermore, since our
problem was a binary classification task, we employed the ‘binary:logistic’ objective function and used
the area under the receiver operating characteristic curve (eval_metric: 'auc') as the evaluation metric.

Hyperparameter optimization was performed using the “fmin()” function from the HyperOpt package,
which executed the “objective()” function with different hyperparameter values specified in the
“hyperparameter_grid”. The “fmin()” function employed the TPE algorithm to suggest subsequent
hyperparameters for evaluation, based on the outcomes of previous evaluations. The optimization
continued until either the maximum number of evaluations specified by “max_evals” was reached or
convergence was achieved. The “best_hyperparams” dictionary returned by “fmin()” contained the
optimal hyperparameters identified during the optimization, within the search space defined by the
“hyperparameter_grid”, aiming to minimize the loss value as defined by the objective function.

Hyperparameter optimization was conducted over 200 iterations, and the best set of hyperparameters was
selected based on the highest AUC value achieved during these iterations. After optimization, the model
was retrained on the entire training set using these optimal hyperparameters.

**Model evaluation.** After model training, we established and locked the final model to ensure rigorous
testing and validation after that. This procedure maintained the complete independence of the hold-out
test set, which was not involved in any stage of the model training process, preventing potential
information leakage. Finally, the final locked model was then tested on the hold-out test set for internal
evaluation, and the external independent, unseen datasets to evaluate its generalizability performance.
Multiple evaluation metrics were employed to comprehensively assess the robustness, generalization
performance, and prediction accuracy of the constructed classification models, including the ROC AUC,
Confusion Matrix, Accuracy, Specificity, Precision, Recall, and F1 Score. These metrics provide a
multifaceted perspective on model performance, encompassing various aspects of prediction accuracy,
robustness, and reliability.

**1) ROC AUC.** ROC (Receiver Operating Characteristic) curves illustrate the trade-off between the true
positive rate (sensitivity) and the false positive rate (1-specificity) for classification algorithms. These
curves are essential tools for evaluating a classifier’s performance, indicating the model’s ability to
increase true positive classifications while controlling for false positives. Ideally, the ROC curve should
approach the upper left corner, signifying superior performance. A curve near the diagonal suggests a
performance close to random guessing. The AUC represents the integral area beneath the ROC curve,
providing a comprehensive single-value summary of the model's performance. AUC is widely regarded
as one of the most effective metrics for encapsulating a classifier’s overall accuracy and efficacy. This
metric ranges from 0.5 to 1.0, where 0.5 represents random guessing, and 1.0 signifies perfect
classification. Models with AUC values close to 1.0 exhibit high accuracy, while those near 0.5 lack
practical utility. Importantly, the AUC is robust against class imbalance, making it a reliable metric

regardless of varying class proportions in the dataset. The AUC calculation involves ranking predictions,
normalizing by the number of positive (M) and negative (N) samples, and handling ties by assigning
average ranks.

The AUC formula is then applied:

$$1451 \quad \text{AUC} = \frac{\sum_{\text{ins}_i \in \text{positive}} \text{rank}_{\text{ins}_i} - \frac{M \times (M + 1)}{2}}{M \times N} \quad (1)$$

Where:

- • $\sum_{\text{ins}_i \in \text{positive}} \text{rank}_{\text{ins}_i}$ is the sum of the ranks for all positive instances.
- • M is the total number of positive samples.
- • N is the total number of negative samples.
- • $\frac{M \times (M + 1)}{2}$ is the sum of the first M natural numbers, which adjusts the rank sum to a baseline
where all positive samples are ranked lower than any negative sample.

**2) Confusion Matrix.** The Confusion Matrix is a tabular representation of the model's predictions,
detailing the counts of True Positives (TP), True Negatives (TN), False Positives (FP), and False
Negatives (FN) for each class. This matrix elucidates the model's performance across different categories,
facilitating the evaluation of its ability to predict different classes, as well as the IE disease stages. The
matrix compares predicted labels with actual labels, offering insights into the proportion of correct and
incorrect predictions across different categories.

**3) Accuracy.** Accuracy measures the ratio of correctly predicted samples (true positives and true
negatives) to the total number of predictions. However, it is unsuitable for unbalanced datasets.

$$1468 \quad \text{Accuracy} = \frac{TP + TN}{TP + TN + FP + FN} \quad (2)$$

**4) Specificity.** Specificity measures the ratio of correctly predicted negative samples (true negatives) to
the total number of actual negative samples.

$$1472 \quad \text{Specificity} = \frac{TN}{TN + FP} \quad (3)$$

**5) Precision and Recall.**

- • **Precision:** The proportion of true positive predictions out of all positive predictions made by the
model.

$$\text{Precision} = \frac{TP}{TP + FP} \quad (4)$$

• Recall (Sensitivity): The proportion of true positive predictions out of all actual positive cases
in the dataset.

$$\text{Recall} = \frac{TP}{TP + FN} \quad (5)$$

**6) F1 Score.** The F1 score balances precision and recall. It is the harmonic mean of precision and recall:

$$F1\text{-Score} = 2 \times \frac{\text{Precision} \times \text{Recall}}{\text{Precision} + \text{Recall}} \quad (6)$$

A high F1 score indicates a good balance between precision and recall, especially in cases of imbalanced
classes.

By employing these metrics, we ensured a comprehensive evaluation of our model’s performance,
considering various aspects of predictive accuracy, robustness, and class distribution.”

**In the revised manuscript**, these revisions are detailed under the subtitles: “(3) Hyperparameter
optimization” (lines 990-1045 on pages 34-35) and “(4) Model evaluation” (lines 1047-1058 on pages
35-36) in the “Methods” section. We hope these clarifications address your concerns. Thank you once
again for your professional comments, which have significantly enhanced the clarity and quality of the
manuscript.

**Q18. (lines 784-787) In this passage, the authors talk about the validation in a second cohort. The**
**authors say that they repeated the steps of model evaluation and hyperparameter tuning. Was the**
**model trained with Cohort 1 used to predict Cohort 2, or was there a model training using only**
**the Cohort 2 data? Also, how was the hyperparameter tuning and model evaluation in cohort 2**
**performed? Was cohort 1 used as a training dataset to select the best set of hyperparameters again**
**and then evaluated on cohort 2?**

**Response to Q18:**

Thanks for the comments. We sincerely apologize for any confusion caused by the unclear description
of model evaluation related to the use of external independent validation Cohort 2 for model evaluation
in the original manuscript. In fact, **the Cohort 1 served as the discovery cohort, utilized for model**
**training and internal testing, while Cohort 2 was an external independent validation cohort, used**
**to evaluate the model’s generalization performance on completely unseen datasets (Figure RL6A).**
We fully agree with the reviewer’s insightful comments that this process needs to be properly and clearly
clarified in the manuscript, along with the rationale behind it, to demonstrate that we have ensured there
is no data leakage throughout the model development and validation stages, which is necessary to
accurately assess the model’s reproducibility and generalizability (*Nat Rev Mol Cell Biol*, 2022, PMID:
34518686). **In the revised manuscript, we have provided a clearer and more detailed explanation**
**of the external validation process.** The detailed response was outlined as follows.

In the original manuscript, the model development was entirely conducted on the discovery cohort (Cohort 1) data. Specifically, the discovery cohort data was divided into a 70% training set and a 30% hold-out test set during model derivation. To avoid potential data leakage, all model training processes, including feature selection, hyperparameter optimization, and the final model training, were exclusively conducted on the training set. The hold-out test set was utilized to assess the final model’s performance for internal validation. Subsequently, to assess the model’s ability to generalize beyond the initial Cohort 1 data—an essential aspect of model validation—we applied the finalized model to the independent, unseen Cohort 2 data without any further training or fine-tuning. **Therefore, we did not repeat the steps of hyperparameter tuning and model evaluation on the external validation cohort (Cohort 2)**; rather, we used this external validation dataset to reassess the model’s predictive accuracy and provide an unbiased assessment of the model’s performance, thereby evaluating the generalization performance of the developed model based on the discovery cohort (Cohort 1). **Thus, we deeply apologize for the confusion caused by the inaccurate description of the original manuscript.** For more details, please refer to the detailed descriptions for model development and evaluation process presented in the “**Response to Q16**” and “**Response to Q17**” sections.

In the revision, according to the reviewer’s suggestions, to avoid potential data leakage that could introduce bias in the model performance evaluation, in addition to the original external validation cohort (Cohort 2), where samples were collected from the same hospital as the discovery cohort (Cohort 1), **we have incorporated another new independent external validation cohort (Cohort 3), where samples were recruited from an external center (a hospital in a different region), to obtain unbiased model performance metrics and to further assess the reproducibility and generalizability of our developed model.** Concurrently, we have supplemented the requested extra analyses and pertinent clarifications to address each question raised by the reviewer in the revision. **Compared to the results in the original manuscript**, the generalizability and robustness of the models were further validated in the Cohort 3 (**Figure 1 and Figure 4**). These results have further strengthened our confidence in the model’s reliability and potential for real-world application.

In the revised manuscript, we have provided a more detailed and clear description of the methodology used in the machine learning pipeline in the “**Methods**” section (**lines 933-1058 on pages 32-36**). Additionally, we have included the results of the newly added external validation cohort (Cohort 3) in the evaluation of the constructed classification models in **Figure 1 and Figure 4**. For more details, please refer to “**Results**” section (**lines 167-183 on page 7; lines 322-358 on pages 11-12**) and “**Discussion**” section (**lines 558-582 on pages 19-20; lines 629-647 on page 22**) of the revised manuscript. We would like to express our gratitude to the reviewer again for their insightful and thoughtful comments, which have helped us improve the manuscript’s clarity, accuracy, and overall quality.

Q19. (lines 280-283) The authors talk about the model results for the NYHA class prediction. In Figure 4 the legend might be switched, as the authors wrote in the legend:

“ML model establishment based on plasma proteomics (B-D) and CBMs (E-G). The top 10 CBMs (B) or the top 20 DEPs (E) for feature importance in distinguishing NYHA class IV from NYHA class II and III patients”

**But in the figure (B-D) show results for CBMs and (E-G) show results for plasma DEPs. The main**
**text matches the figure. Please verify if that is correct.**

**Response to Q19:**

We sincerely appreciate the reviewer for carefully reviewing our manuscript and pointing out the issue
in the legend in **Figure 4. We have confirmed that there was indeed an error in the legend**, where
the labels for CBMs and the plasma proteome were inadvertently switched. As suggested by the reviewer,
the correct description of the legend should be:

“ML model establishment based on CBMs (B-D) and plasma proteome (E-G). The top 10 CBMs (B) or
the top 20 DEPs (E) for feature importance in distinguishing NYHA class IV from NYHA class II and
III patients.”

We have corrected this issue to ensure the textual description accurately reflects the content of the figure
and updated the corresponding legend in the “**Figure legends**” section of the revised manuscript (**line**
**1450 on page 47**). Additionally, in the revision, we have thoroughly reviewed the entire manuscript and
systematically corrected any inaccuracies and errors to ensure no similar issues remain. Once again, we
sincerely appreciate the reviewer for the insightful comments, which have significantly improved the
quality and clarity of our manuscript.

**Q20. In Figure 4A there is a schematic of the machine learning pipeline. There are two datasets**
**represented there, one with 194 samples and another with 199 samples. Why are there two different**
**datasets and why are they used differently? According to the image, one is used to search for**
**optimal CBMs and the other is used to search for optimal DEPs, please provide more information**
**about this schematic and about the datasets used in this step.**

**Also, in the split section of the image the authors says that the datasets were split into 70% train**
**and 30% validation and further on the image there is a CV with k=6, in this case, the 30 %**
**validation dataset is actually a hold-out test set? Since k=6 for the cross-validation, the validation**
**dataset should be approx. 16.67%.**

**Response to Q20:**

We sincerely appreciate the reviewer for their careful review and for highlighting areas that might cause
potential confusion or readability challenges in the original manuscript. In response to your insightful
feedback, **we have thoroughly reviewed the sections you pointed out and have revised the schematic**
**of the machine learning pipeline and relevant descriptions** to enhance the accuracy, clarity, and
overall readability (**Figure RL7B**) in the revised manuscript. In response to the reviewer’s inquiries, we
have outlined our clarifications below:

**(1) About the Two Datasets in Figure 4A of the Original Manuscript:**

We sincerely apologize for the unclear presentation of the two datasets utilized in the machine learning
pipeline of the original manuscript. To clarify this, we have provided a schematic diagram below (**Figure**
**RL7A**). Actually, **both datasets originate from the same group of 202 IE patients in the discovery**
**cohort (Cohort 1), capturing patient characteristics across different dimensions.** To be precise, 3 of

the 202 IE patients were initially excluded from the subsequent analysis due to missing class label
information. Among the remaining 199 IE patients, Dataset 1 included clinical information primarily
derived from routine blood tests, covering 194 patients with complete and accessible clinical data.
Dataset 2 comprised plasma proteomic data, with all 199 IE patients having complete and accessible
plasma proteomic data.

As the reviewer mentioned, the two different datasets were used to search for optimal clinical blood
markers (CBMs) measured by clinical blood tests and optimal differentially expressed proteins (DEPs)
identified through plasma proteomic analysis, respectively, to predict IE patients with poor prognosis.
Specifically, we developed three prognostic models utilizing the blood-derived characteristics of the
CBMs and plasma proteome: CBM-only, proteome-only, and proteome + CBM, to distinguish between
IE patients with NYHA class IV, indicating a worse prognosis, and those with NYHA class II and III.
Dataset 1 and Dataset 2 were used to develop the CBM-only and the proteome-only prognostic models,
respectively.

**Figure RL7A.** A schematic of the clinical cohort design for developing models in the original manuscript.

**The reasons for investigating the two datasets separately primarily included the following aspects.**

The CBMs measured by clinical blood tests and the DEPs identified through plasma proteomic analysis,
as two blood-based characteristics, hold significant potential for early and convenient prediction of poor
prognosis in IE patients. By taking a two-pronged approach, we were able to: 1) independently identify
the optimal CBMs and plasma protein biomarkers; 2) compare the predictive performance of models
built using the CBMs versus the plasma protein biomarkers, to assess whether the latter offered
incremental value for risk stratification of IE patients; and 3) evaluate whether combining these two data
modalities provided complementary information that could be leveraged to enhance risk assessment in
IE. This comprehensive evaluation provides valuable insights that could inform future IE diagnostic and
risk prediction strategies.

**In the revision,** according to your insightful feedback, we have focused on the cohort of 194

overlapped patients who had both clinical blood tests and plasma proteome data, as well as the

NYHA functional class labels for each patient. To clarify the updated framework of the machine

learning pipeline for the prognostic models of IE patients, we have provided a corresponding schematic

workflow below (**Figure RL7B**). This allowed us to directly compare the predictive performance of
 models built using the blood test-derived biomarkers versus the plasma protein biomarkers within the
 same patient cohort. In this way, we can directly compare their predictive capabilities without
 confounding factors related to potential patient population differences.

 **Figure RL7B.** The schematic of the updated machine learning pipeline of the revised manuscript.

**(2) About the Methodology of Data Split in Figure 4A of the Original Manuscript:**

We sincerely apologize for the confusing and ambiguous presentation of the data split in **Figure 4A** of
 the original manuscript. Indeed, the data split methodology was as follows:

- 1. The full dataset (discovery cohort–Cohort 1) was initially split randomly into a 70% training set
 and a 30% hold-out test set for internal evaluation.
 - 2. The 70% training set was then used for 6-fold cross-validation, where in each iteration,
 approximately 83.33% (5/6) of the training set was used for model training. The remaining 16.67%
 (1/6) of the training set was used for model validation during cross-validation. This 6-fold cross-
 validation approach entailed training and testing the model six times to ensure a robust performance
 assessment, which was preferred to prevent overfitting.
 - 3. The separate 30% test set was not utilized during the model training processes with k-fold cross-
 validation; instead, it was held out for evaluating the final model during internal validation.

**In the revised manuscript**, according to your constructive suggestions, **we have implemented a**
 **stratified randomized split approach to split the entire discovery dataset and have retrained the**
 **three ML models** (CBM-based, proteome-based, and the combined “proteome + CBM”) utilizing the
 CBMs and/or DEPs, respectively. The updated schematic workflow for the machine learning pipeline
 was shown above (**Figure RL7B**). In the development of the three models, the full datasets of the clinical
 blood test dataset (n = 194), plasma proteomic dataset (n = 194), and the final dataset with initial optimal
 proteome + CBM features were first split stratified by the class label (NYHA class IV or NYHA class II
 and III), respectively, to ensure the distributions in the training and test sets were representative of the
 overall data. The detailed methodology descriptions for model development and evaluation process were
 presented in the “**Response to Q16**” and “**Response to Q17**” sections.

**Compared to the results in the original manuscript**, the revision for the development of the **three**
 **models included the following improvements:** 1) The updated model exhibited performance metrics
 that were consistent with, or even better than, those of the original version; 2) The generalizability and
 robustness of the models were further validated based on the data from the newly incorporated external

validation cohort recruited from an external center (Cohort 3). **For instance**, in the results for the
 development of proteome + CBM model (**Figure RL7C-F**), we found that the final model contained 15
 features of 4 CBMs (e.g., DBIL, BUN) and 11 DEPs (e.g., ADIPOQ, FETUB), with an overall AUC
 value of 0.873 (95% CI: 0.865-0.881) in the hold-out test set for internal validation and an overall AUC
 value of 0.829 (95% CI: 0.822-0.834) in the external validation dataset for external validation. The
 confusion matrix results revealed good model performance in distinguishing between NYHA class IV
 patients and those in NYHA class II and III.

 **Figure RL7C-F.** (C) Scatter line plot illustrating the optimal feature combinations for the proteome +
 CBM model that yield the highest accuracy, determined through RFECV. (D) Boxplots and heatmaps
 illustrating significant differences in the 15 features identified in the proteome + CBM model between
 NYHA class IV patients and those in NYHA class II and III. The annotation of pathogen identified by
 mNGS for each sample is displayed above the heatmap, with blanks indicating missing records. (E, F)
 ROC curves (E) and confusion matrices (F) illustrating the performance of the proteome + CBM model
 on the hold-out test set of the discovery cohort (Cohort 1) and the external validation cohort (Cohort 3).

**In the revised manuscript**, we have systematically updated the relevant figures and text throughout the
 manuscript to reflect the updated methods, results, and findings. Specifically, we have provided a more
 detailed and clear description of the methodology used in the machine learning pipeline in the “**Methods**”
 section (**lines 933-1058 on pages 32-36**). Additionally, we have included the results of the newly added
 external validation cohort (Cohort 3) in evaluating the constructed classification models in **Figure 4**. A
 summary of the corresponding updated results has been incorporated in the “**Results**” section (**lines 322-**
 **358 on pages 11-12**) and “**Discussion**” section (**lines 629-647 on page 22**) of the revised manuscript.
 We would like to express our gratitude to the reviewer again for their insightful and thoughtful comments,
 which have helped us improve the manuscript’s clarity, accuracy, and overall quality.

**Q21.** In Figures 1H and 4I, the authors present ROC curves for the test set. It is unclear to me why
 this plot is multiple fold. If the model was trained using cross-validation, the hold-out test set should

**be tested on the final model with the best set of hyperparameters thus you shouldn't have multiple**
**test ROC curves. Is that ROC curve related to the test set or the validation set (from the cross-**
**validation using the training set)? If there is a reason for the multiple test ROC curves, please**
**provide more detail about how the models were trained and evaluated in the methods section.**

**Response to Q21:**

Thank you for your insightful comments. We deeply apologize for the inconvenience caused by the lack
of detailed information regarding the model training and evaluation methodology in the original
manuscript. As the reviewer noted, we have confirmed that the ROC curves presented in the original
**Figures 1H and 4I all referred to the validation set (from the cross-validation using the training**
**set) rather than the hold-out test set. In the revision,** we have revised and updated the relevant figures
and text accordingly. In response to the reviewer's inquiries, we have provided our clarifications below:

In this study, we split the discovery cohort (Cohort 1) data into a 70% training set and a 30% hold-out
test set during model derivation. All model training processes, including feature selection,
hyperparameter optimization, and final model training, were exclusively conducted on the training set to
avoid potential data leakage. We employed a k-fold cross-validation strategy for model training. This
approach maximizes data utilization and reduces the risk of overfitting, thereby providing a more reliable
estimate of the model's generalization performance. **As the reviewer commented, once established,**
**the final model was locked to ensure rigorous testing and validation. It was tested on the hold-out**
**test set for internal evaluation and the external independent, unseen datasets to evaluate its**
**generalizability performance.** For more details, please refer to the detailed descriptions of the model
development and evaluation process in the **"Response to Q16"** and **"Response to Q17"** sections.

To assess model performance, we employed the AUC-ROC, accuracy, precision, F1-score, and
confusion matrix, which are standard and widely used evaluation metrics for classification models.
**Before final testing, we employed an AUC-ROC with cross-validation analysis to estimate and**
**visualize the model performance during the model training and development stage.** Specifically,
we utilized the "StratifiedKFold" function ($cv = k$) from the Python "sklearn.model_selection" module
to perform the cross-validation analysis. During k-fold cross-validation, the training set was divided into
k subsets (or folds). For each iteration, the model was trained on k-1 of the k folds, and the model was
then evaluated on the remaining fold (the test set for that iteration). This process was repeated k times,
each time using a different fold as the test set, ensuring that every data point was used for both training
and testing. For each cross-validation fold, we trained the model and generated the corresponding AUC-
ROC using the "RocCurveDisplay" and AUC functions from the Python "sklearn.metrics" module.
Taking all of these curves, it is possible to calculate the mean AUC, and see the variance of the curve
when the training set is split into different subsets. This allowed us to leverage the mean AUC and the
variance in AUC values across the cross-validation folds to quantify the model's overall classification
performance and reliability, which guided us in selecting the final model. Finally, the hold-out test set
was utilized to evaluate the final model's performance, and the results were presented with the confusion
matrix. Taken together, we sincerely appreciate you pointing out the issue with the descriptions of the
figure legends in the original **Figures 1H and 4I. The appropriate description should be: "Mean**
**ROC curve with variability for training set"** rather than "ROC of test set".

**In the revision**, to more clearly and intuitively evaluate the performance of the final model, we have
presented both the AUC-ROC and the confusion matrix results from the final model’s evaluation on the
hold-out test set (**Figure 1 and Figure 4**). We have provided a more detailed and clear description of the
model evaluation methodology in the revised manuscript, specifically under the subtitles: “(4) Model
evaluation.”. Additionally, we have incorporated these results into the “**Results**” section (**lines 167-183**
**on page 7; lines 322-358 on pages 11-12**) and “**Discussion**” section (**lines 558-582 on pages 19-20;**
**lines 629-647 on page 22**) of the revised manuscript. We would like to express our gratitude to the
reviewer again for their insightful and thoughtful comments, which have helped us improve the
manuscript’s clarity, accuracy, and overall quality.

**In summary, the model performances are good, and the methods were well thought. I believe that**
**clarifying the mentioned points would be beneficial for the overall understanding of the machine**
**learning framework used here and could increase the reliability of the interesting findings they**
**reported in this work.**

**Response:**

We sincerely appreciate the reviewer’s positive and insightful comments, which have been instrumental
in improving the quality of our work. We thoroughly studied each point raised in the comments and
accordingly revised the manuscript.

**Reviewer #4 (Remarks to the Author):**

**I co-reviewed this manuscript with one of the reviewers who provided the listed reports. This is**
**part of the Nature Communications initiative to facilitate training in peer review and to provide**
**appropriate recognition for Early Career Researchers who co-review manuscripts**

**Response:**

We truly appreciate the collaborative effort and insightful comments provided by you and your co-
reviewer. Your constructive feedback has significantly contributed to the improvement of our manuscript.
We have carefully considered each point raised in the comments and systematically revised the
manuscript accordingly. For more details, please refer to the detailed point-by-point responses above.